# STOCHASTIC GRADIENT LANGEVIN DYNAMICS BASED ON QUANTIZATION WITH INCREASING RESOLUTION

## ABSTRACT

Stochastic learning dynamics based on Langevin or Levy stochastic differential equations (SDEs) in deep neural networks control the variance of noise by varying the size of the mini-batch or directly those of injecting noise. Since the noise variance affects the approximation performance, the design of the additive noise is significant in SDE-based learning and practical implementation. In this paper, we propose an alternative stochastic descent learning equation based on quantized optimization for non-convex objective functions, adopting a stochastic analysis perspective. The proposed method employs a quantized optimization approach that utilizes Langevin SDE dynamics, allowing for controllable noise with an identical distribution without the need for additive noise or adjusting the mini-batch size. Numerical experiments demonstrate the effectiveness of the proposed algorithm on vanilla convolution neural network(CNN) models and the ResNet-50 architecture across various data sets. Furthermore, we provide a simple PyTorch implementation of the proposed algorithm.

## 1 INTRODUCTION

Stochastic analysis for a learning equation based on stochastic gradient descent (SGD) with a finite or an infinitesimal learning rate has been an essential research topic to improve machine learning performance. Particularly, the linear scaling rule(LSR) for SGD discovered by Krizhevsky (2014); Chaudhari and Soatto (2018) and Goyal et al. (2018) independently provides an essential guide to select or control the learning rate corresponding to the size of the mini-batch. More crucially, it gives a fundamental framework of stochastic analysis for the learning equation in current deep neural networks(DNN). However, the early analysis of SDE-based SGD encountered counterexamples, as demonstrated by Hoffer et al. (2017); Shallue et al. (2018); Zhang et al. (2019). Those works claim that the SGD with a momentum term or an appropriate learning rate represents superior performance to the SGD with a varying size mini-batch (Mandt et al. (2017); Kidambi et al. (2018); Liu and Belkin (2020)), even though the noise term gives improved classification performance. As related research progresses, recent studies reached the following consensus: for an objective function being close to a standard convex, the SGD with mini-batch represents better performance, while for an objective function being a non-convex or curvature dominated, the SGD with momentum is better (Ma et al. (2018); Zhang et al. (2019); Smith et al. (2019) and Smith et al. (2020)).

The other research topic of the stochastic analysis for SGD is whether or not the induced SGD noise is Gaussian. Simsekli et al. (2019); Nguyen et al. (2019) suggested that SGD noise is a heavy-tailed distribution. This argument means that if SGD noise is not Gaussian, we should analyze SGD as a Levy process-based SDE instead of the standard SDE framework. For these claims, Wu et al. (2020); Cheng et al. (2020b) and Li et al. (2022) revealed that the third or higher order moments in SGD noise have minimal effect on accuracy performance, while the second moment has a significant impact. In other words, the standard SDE is still valid for the stochastic analysis of SGD noise in the learning equation because the second moment of noise is the core component of the standard SDE. As the recent research substantiates the validation of the stochastic analysis for SGD noise based on the standard SDE, Li et al. (2019); Granziol et al. (2022); Malladi et al. (2022); Kalil Lauand and Meyn (2022) and Li et al. (2022) reinterpreted the conventional algorithm based on the standard SDE. Moreover, with the advent of novel algorithms and comprehensive analyses in noisy SGD research (e.g., the works of Fonseca and Saporito (2022); Masiha et al. (2022) and Altschuler and Talwar (2022)), standard SDE-based noisy SGD algorithm is gaining widespread popularity.

Another research is stochastic gradient Langevin dynamics(SGLD), which injects an isotropic noise, such as the Wiener process, into SGD (Welling and Teh (2011); Brosse et al. (2018); Dalalyan and Karagulyan (2019); Cheng et al. (2020a) and Zhang et al. (2022)). Unlike the noise derived from LSR, the noise introduced by SGLD, which originates from Markov Chain Monte Carlo, consists of independent and identically distributed (I.I.D.) components. As a result, we can readily apply this noise to a conventional learning algorithm to enhance its performance by robustness to non-convex optimization((Raginsky et al. (2017); Xu et al. (2018); Mou et al. (2018) and Wang et al. (2021)). However, LSR-based SGD and SGLD require additional processes such as warm-up(Goyal et al. (2018)), extra computation according to stochastic variance amplified gradient(SVAG) (Li et al. (2021) and Malladi et al. (2022)), or an identical random number generator for SGLD. Furthermore, with the advancement of research on distribution/federated learning, there is growing opposition to increasing the size of mini-batches due to practical considerations in optimization techniques. Lin et al. (2020) argued that in a distributed system with a heterogeneous hardware environment, including small computational devices, we could not expect a model learned using generalized large-batch SGD to be suitable. Therefore, they advocate for using small-sized batch SGD in such environments.

This paper introduces an alternative learning algorithm based on quantized optimization for SGLD, to address the practical issues related to LSR-based SGD and SGLD. The proposed methodology makes the following contributions.

**Optimization based on Quantization** We present the learning equation based on the quantized optimization theory, incorporating the white noise hypothesis (WNH) as suggested by Zamir and Feder (1996); Benedetto et al. (2004); Gray and Neuhoff (2006) and Jiménez et al. (2007). The WNH posits that the quantization error follows an I.I.D. white noise under regular conditions with sufficiently large sample data. While the primary goal of quantization is to reduce computational burden by simplifying data processing in conventional artificial intelligence and other signal engineering(Seide et al. (2014); De Sa et al. (2015); Han et al. (2015); Wen et al. (2016); Jung et al. (2019) and Li and Li (2019)), in our work, quantization serves as the core technology for enhancing optimization performance during the learning process. Additionally, the quantization error effectively incorporates the various noise generated by the algorithm and establishes it as additive white noise according to the WNH, thereby facilitating the SDE analysis for optimization.

**Controllable Quantization Resolution** By defining the quantization resolution function based on time and other parameters, we propose an algorithm that computes the quantization level. Controlling the noise variance induced by the quantization error is essential to apply the quantization error to an optimizer effectively. While the WNH allows us to treat the quantization error as I.I.D. white noise, it alone does not guarantee optimal results if the uncontrolled variance exists. Therefore, learning based on SGLD becomes feasible without a random number generator required by SGLD or MCMC. Furthermore, similar to increasing the mini-batch size in LSR-based SGD, we can develop a scheduler using controlled quantization resolution for optimization.

**Non-convex Optimization** The proposed optimization algorithm demonstrates robust optimization characteristics for non-convex objective functions. The quantized optimization algorithm outperforms MCMC-based optimization methods in combinatorial optimization problems, such as simulated and quantum annealing. Although further empirical evidence is needed, this result indicates that quantized optimization is a viable approach for non-convex optimization problems. We analyze the proposed algorithm's weak and local convergence to substantiate this claim.

## 2 PRELIMINARIES AND OVERVIEW

### 2.1 STANDARD ASSUMPTIONS FOR THE OBJECTIVE FUNCTION

We establish the objective function $f : \mathbb{R}^d \mapsto \mathbb{R}, \; f \in C^2$ for a given state parameter(e.g., weight vectors in DNN) $\boldsymbol{x}_{t_e} \in \mathbb{R}^d$ such that

$$f(\boldsymbol{x}_{t_e}) \triangleq \frac{1}{N_T} \sum_{k=1}^{N_T} \bar{f}_k(\boldsymbol{x}_{t_e}) = \frac{1}{B \cdot n_B} \sum_{\tau_b=1}^{B} \sum_{k=1}^{n_B} \bar{f}_{\tau_b \cdot n_B + k}(\boldsymbol{x}_{t_e}), \quad \bar{f}_k : \mathbb{R}^d \to \mathbb{R}, \; \forall \tau_b \in \mathbb{Z}^+[0, B),$$

$$\text{(1)}$$

where $t_e$ denotes a discrete time index indicating the epoch, $\bar{f}_k : \mathbb{R}^d \mapsto \mathbb{R}, \; \bar{f}_k \in C^2$ denotes a loss function for the $k$-th sample data, $B \in \mathbb{Z}^+$ denotes the number of mini-batches, $n_B$ denotes the

equivalent number of samples for each mini-batch $B_j$, and $N_T$ denotes the number of samples for an input data set such that $N_T = B \cdot n_B$.

In practical applications in DNN, the objective function is the summation of entropies with a distribution based on an exponential function such as the softmax or the log softmax, so the analytic assumption of the objective function (i.e., $f \in C^2$) is reasonable. Additionally, since a practical framework for DNN updates the parameter $\boldsymbol{x}_{t_e}$ with a unit mini-batch, we can rewrite the objective function as an average of samples in a mini-batch such that

$$f(\boldsymbol{x}_t) = \frac{1}{B} \sum_{\tau_b=0}^{B-1} \tilde{f}_{\tau_b}(\boldsymbol{x}_{t_e+\tau_b/B}),\ t \in \mathbb{R}[t_e, t_e+1) \quad \because \tilde{f}_{\tau_b}(\boldsymbol{x}_{t_e}) = \frac{1}{n_B} \sum_{k=1}^{n_B} \bar{f}_{\tau_b \cdot n_B + k}(\boldsymbol{x}_{t_e}). \quad (2)$$

Under the definition of the objective function, we establish the following assumptions:

**Assumption 1.** *For $\boldsymbol{x}_t \in B^o(\boldsymbol{x}^*, \rho)$, there exists a positive value $L_0$ with respect to $f$ such that*

$$|f(\boldsymbol{x}_t) - f(\boldsymbol{x}^*)| \leq L_0 \|\boldsymbol{x}_t - \boldsymbol{x}^*\|, \quad \forall t > t_0, \quad (3)$$

*where $B^o(\boldsymbol{x}^*, \rho)$ denotes an open ball $B^o(\boldsymbol{x}^*, \rho) = \{\boldsymbol{x} | \|\boldsymbol{x} - \boldsymbol{x}^*\| < \rho\}$ for all $\rho \in \mathbb{R}^+$, and $\boldsymbol{x}^* \in \mathbb{R}^d$ denotes the unique globally optimal point such that $f(\boldsymbol{x}^*) < f(\boldsymbol{x}_t)$. Furthermore, we define the Lipschitz constants $L_1 > 0$ for the first-order derivation of $f$, such that*

$$\|\nabla_{\boldsymbol{x}} f(\boldsymbol{x}_t) - \nabla_{\boldsymbol{x}} f(\boldsymbol{x}^*)\| \leq L_1 \|\boldsymbol{x}_t - \boldsymbol{x}^*\|. \quad (4)$$

In Assumption 1, we employ the time index $t \in \mathbb{R}^+$ instead of $t_e \in \mathbb{Z}^+$ to apply the assumption to an expanded continuous approximation. We assume that the set of the discrete epoch-unit time $\{t_e | t_e \in \mathbb{Z}^+\}$ is a subset of the set of the continuous time $\{t | t \in \mathbb{R}^+\}$.

## 2.2 Definition and Assumptions for Quantization

The conventional research relevant to signal processing defines a quantization such that $x^Q \triangleq \lfloor \frac{x}{\Delta} + \frac{1}{2} \rfloor \Delta$ for $x \in \mathbb{R}$, where $\Delta \in \mathbb{Q}^+$ denotes a fixed valued quantization step. We provide a more detailed definition of quantization to explore the impact of the quantization error, using the quantization parameter as the reciprocal of the quantization step such that $Q_p \triangleq \Delta^{-1}$.

**Definition 1.** *For $x \in \mathbb{R}$, we define the quantization of $x$ as follows:*

$$x^Q \triangleq \frac{1}{Q_p} \lfloor Q_p \cdot (x + 0.5 \cdot Q_p^{-1}) \rfloor = \frac{1}{Q_p} (Q_p \cdot x + \varepsilon^q) = x + \varepsilon^q Q_p^{-1}, \quad x^Q \in \mathbb{Q}, \quad (5)$$

*where $\lfloor x \rfloor \in \mathbb{Z}$ denotes the floor function such that $\lfloor x \rfloor \leq x$ for all $x \in \mathbb{R}$, $Q_p \in \mathbb{Q}^+$ denotes the quantization parameter, and $\varepsilon^q \in \mathbb{R}$ is the factor for quantization such that $\varepsilon^q \in \mathbb{R}[-1/2, 1/2)$.*

Furthermore, for a given normal Euclidean basis $\{\boldsymbol{e}^{(i)}\}_{i=1}^d$, we can write a vector $\boldsymbol{x} \in \mathbb{R}^d$ such that $\boldsymbol{x} \triangleq \sum_{i=1}^d (\boldsymbol{x} \cdot \boldsymbol{e}^{(i)}) \boldsymbol{e}^{(i)}$. Using these notations, we can define the quantization of a vector $\boldsymbol{x}^Q \in \mathbb{Q}^d$ and the vector-valued the quantization error $\boldsymbol{\epsilon}^q \triangleq \boldsymbol{x}^Q - \boldsymbol{x} = Q_p^{-1} \boldsymbol{\varepsilon}^q \in \mathbb{R}^d$ as follows:

$$\boldsymbol{x}^Q \triangleq \sum_{i=1}^d (\boldsymbol{x} \cdot \boldsymbol{e}^{(i)})^Q \boldsymbol{e}^{(i)} \implies \boldsymbol{\epsilon}^q = Q_p^{-1} \boldsymbol{\varepsilon}^q = \sum_{i=1}^d \left( (\boldsymbol{x}^Q - \boldsymbol{x}) \cdot \boldsymbol{e}^{(i)} \right) \boldsymbol{e}^{(i)}. \quad (6)$$

We distinguish the scalar factor for quantization $\varepsilon^q \in \mathbb{R}[-1/2, 1/2)$, the vector valued factor $\boldsymbol{\varepsilon}^q \in \mathbb{R}^d[-1/2, 1/2)$, the scalar valued quantization error $\epsilon^q = Q_p^{-1} \varepsilon^q \in \mathbb{R}[-Q_p^{-1}/2, Q_p^{-1}/2)$, and the vector valued quantization error $\boldsymbol{\epsilon}^q \in \mathbb{R}^d[-Q_p^{-1}/2, Q_p^{-1}/2)$ respectively.

**Definition 2.** *We define the quantization parameter $Q_p : \mathbb{R}^d \times \mathbb{R}^{++} \mapsto \mathbb{Q}^+$ such that*

$$Q_p(\boldsymbol{\varepsilon}^q, t) = \eta(\boldsymbol{\varepsilon}^q) \cdot b^{\bar{p}(t)}, \quad (7)$$

*where $\eta : \mathbb{R}^d \mapsto \mathbb{Q}^{++}$ denotes the auxiliary function of the quantization parameter, $b \in \mathbb{Z}^+$ is the base, and $\bar{p} : \mathbb{R}^{++} \mapsto \mathbb{Z}^+$ denotes the power function such that $\bar{p}(t) \uparrow \infty$ as $t \to \infty$. If $\eta$ is a constant value, the quantization parameter $Q_p$ is a monotone increasing function concerning $t$.*

In definition 2, we can establish an intuitive stochastic approximation based on the random variable with Gaussian distribution by an appropriate transformation through the auxiliary function $\eta$, even though the probability density function of the quantization error is a uniform distribution. We'll investigate the stochastic approximations depending on the function $\eta$ in the following chapter.

As a next step, we establish the following assumptions to define the statistical properties of the quantization error.

**Assumption 2.** *The probability density function of the quantization error $\epsilon^q$ is a uniform distribution $p_{\epsilon^q}$ on the quantization error's domain $\mathbb{R}[-Q_p^{-1}/2, Q_p^{-1}/2)$.*

Assumption 2 leads to the following scalar expectation and variance of the quantization error $\epsilon^q$ trivially as follows:

$$\forall \varepsilon^q \in \mathbb{R}, \quad \mathbb{E}_{\varepsilon^q} Q_p^{-1} \varepsilon^q = 0, \quad \mathbb{E}_{\varepsilon^q} Q_p^{-2} \varepsilon^{q2} = Q_p^{-2} \cdot \mathbb{E}_{\varepsilon^q} \varepsilon^{q2} = 1/12 \cdot Q_p^{-2} = c_0 Q_p^{-2}. \quad (8)$$

**Assumption 3** (**WNH from Jiménez et al. (2007)**). *When there is a large enough sample and the quantization parameter is sufficiently large, the quantization error is an independent and identically distributed white noise.*

**Independent condition of quantization error** Assumption 3 is reasonable when the condition is satisfied for the quantization errors as addressed in Zamir and Feder (1996); Marco and Neuhoff (2005); Gray and Neuhoff (2006) and Jiménez et al. (2007). However, the independence between the input signal for quantization and the quantization error is not always fulfilled, so we should check it. For instance, if we let a quantization value $X^Q = kQ_p^{-1} \in \mathbb{Q}$ for $k \in \mathbb{Z}$, we can evaluate the correlation of $X^Q$ and the quantization error $\epsilon^q$ such that

$$\mathbb{E}_{\varepsilon^q}[X(X^Q - X)|X^Q = kQ_p^{-1}] = \mathbb{E}_{\varepsilon^q}[X\epsilon^q|X^Q = kQ_p^{-1}] = c_0 Q_p^{-2}. \quad (9)$$

Accordingly, if the quantization parameter $Q_p$ is a monotone increasing function to a time index $t$ defined in Definition 2 such that $Q_p^{-1}(t) \downarrow 0$, $t \uparrow \infty$, we can regard the quantization error as the i.i.d. white noise as described in Assumption 3, for $t \geq t_0$. However, even though the quantization parameter is a time-dependent monotone-increasing function, we cannot ensure the independent condition between an input signal and the quantization error at the initial stage when the time index is less than $t < t_0$. For instance, since the quantization error is not independent of the input signal for quantization, the quantization error represents zero when the quantized input is zero, even though the input itself is not zero. Such a broken independent condition can cause an early paralysis of learning since there exists nothing to update the state parameter. On the contrary, when the quantization error is independent of an input signal, we can use the quantization error to optimize the objective function without early paralysis, despite the small valued norm of the input at the initial stage.

We will present the compensation function to ensure the independent condition and avoid learning paralysis at the early learning stage in the other section.

## 3 LEARNING EQUATION BASED ON QUANTIZED OPTIMIZATION

Consider that the learning equation given by:

$$\boldsymbol{X}_{\tau+1} = \boldsymbol{X}_\tau + \lambda h(\boldsymbol{X}_\tau), \quad \boldsymbol{X}_\tau \in \mathbb{R}^d, \, \forall \tau > 0, \quad (10)$$

where $h : \mathbb{R}^d \mapsto \mathbb{R}^d$ represents the search direction, and $\tau$ denotes the time index depending on the index of a mini-batches defined in *equation* 1 and *equation* 2. Most artificial intelligence frameworks provide a learning process depending on the unit of the mini-batch size, so *equation* 10 describes a real and practical learning process.

**Main Idea of the Proposed Quantized Optimization** The learning equation, represented as *equation* 10, searches for a local minimum along a line defined by a directional vector or a conjugate direction when the equation incorporates momentum. In the proposed quantized optimization, we create a grid on the objective function's domain and sample a point near the feasible point generated by the learning equation. The grid size is adjustable through the quantization parameter. By considering the quantization error as a white noise following WNH, we can reduce the variance of the quantization error by increasing the quantization parameter's size. This adjustment approximates the dynamics of Langevin SDE in stochastic approximation in the sense of the central limit theorem. From an optimization perspective, stochastic gradient-based learning can be considered stochastic sampling.

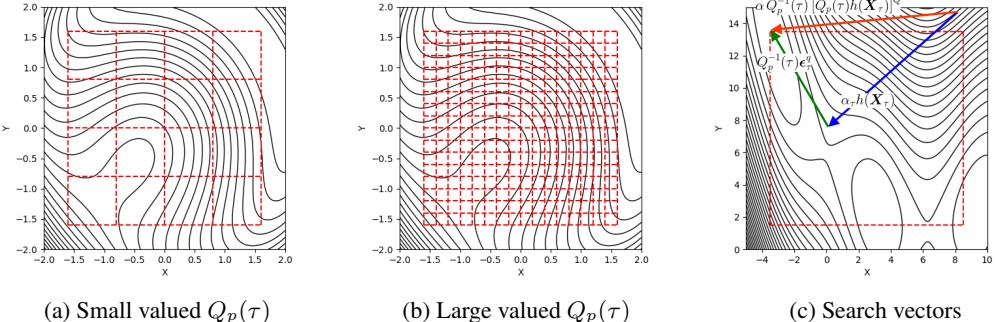

(a) Small valued $Q_p(\tau)$      (b) Large valued $Q_p(\tau)$      (c) Search vectors

Figure 1: The concept diagram of the proposed quantization: (a) In the early stage of learning, with a small value of $Q_p(\tau)$, the search points generated by quantization are widely spaced, indicating a significant quantization error or Brownian motion process affecting the learning process. (b) As the learning process progresses, the quantization parameter $Q_p(\tau)$ increases, resulting in smaller grid sizes. This process resembles an annealing-type stochastic optimization. (c) When considering a general search vector $\alpha, h(\boldsymbol{X}_\tau)$, quantization introduces an additional search vector, as shown.

## 3.1 Fundamental Learning Equation based on Quantization

Applying the quantization defined as *equation* 5 and *equation* 6 to the learning equation, *equation* 10, we can obtain the following fundamental quantization-based learning equation:

$$\boldsymbol{X}_{\tau+1}^Q = \boldsymbol{X}_\tau^Q + [\lambda h(\boldsymbol{X}_\tau^Q)]^Q = \boldsymbol{X}_\tau^Q + Q_p^{-1}(\tau) \left\lfloor Q_p(\tau) \cdot (\lambda h(\boldsymbol{X}_\tau^Q) + 0.5 Q_p^{-1}) \right\rfloor, \ \boldsymbol{X}_0^Q \in \mathbb{Q}^d. \quad (11)$$

According to the definition of quantization, we can rewrite *equation* 11 to the following stochastic equation similar to the discrete Langevin equation :

$$\boldsymbol{X}_{\tau+1}^Q = \boldsymbol{X}_\tau^Q + \lambda\, h(\boldsymbol{X}_\tau^Q) + \boldsymbol{\epsilon}_\tau^q = \boldsymbol{X}_\tau^Q + \lambda\, h(\boldsymbol{X}_\tau^Q) + Q_p^{-1}(\varepsilon_\tau^q, \tau)\varepsilon_\tau^q, \quad (12)$$

where $\boldsymbol{\epsilon}_\tau^q \in \mathbb{R}^d$ denotes the vector-valued quantization error, and $\varepsilon_\tau^q \in \mathbb{R}^d$ denotes the vector-valued factor for the quantization, defined as *equation* 6 respectively.

Substituting the search direction $h(\boldsymbol{X}_\tau^Q)$ with $-\nabla_{\boldsymbol{x}} \tilde{f}_\tau(\boldsymbol{X}_\tau^Q)$, we rewrite *equation* 12 to the following equation:

$$\boldsymbol{X}_{\tau+1}^Q = \boldsymbol{X}_\tau^Q - \lambda \nabla_{\boldsymbol{x}} \tilde{f}_\tau(\boldsymbol{X}_\tau^Q) + Q_p^{-1}(\varepsilon_\tau^q, \tau)\varepsilon_\tau^q. \quad (13)$$

While the fundamental quantized learning equations *equation* 12 and *equation* 13 are the formulae of a representative stochastic difference equation, we cannot analyze the dynamics of these equations as a conventional stochastic equation, due to the quantization error as a uniformly distributed vector-valued random variable. Therefore, to analyze the dynamics of the proposed learning equation from the perspective of stochastic analysis, we suggest the following two alternative approaches:

**Transformation to Gaussian Wiener Process** Using the fundamental form of the quantization parameter, we can transform a uniformly distributed random variable into a standard or approximated Gaussian-distributed random variable. Since the quantization parameter including the transform can generate a Gaussian distributed independent increment $\boldsymbol{z}_\tau \overset{i.i.d.}{\sim} \mathcal{N}(\boldsymbol{z}; 0, \boldsymbol{I}_d)$ such that $\eta(\varepsilon_\tau^q)\varepsilon_\tau^q = \sqrt{\lambda}\boldsymbol{z}_\tau$ under the assumption of which $\lambda = 1/B$, we can rewrite *equation* 13 as follows:

$$\boldsymbol{X}_{\tau+1}^Q = \boldsymbol{X}_\tau^Q - \lambda \nabla_{\boldsymbol{x}} \tilde{f}_\tau(\boldsymbol{X}_\tau^Q) + \sqrt{\lambda} \cdot b^{-\bar{p}(\tau)} \boldsymbol{z}_\tau. \quad (14)$$

However, even though the transformation offers theoretical benefits derived from a Gaussian-distributed process, there are no advantages to implementing the learning equation, as the property of the quantization error is equivalent to that of a uniformly distributed random variable in a conventional random number generator. Consequently, we do not treat the transformation-based algorithm.

**Analysis based on Central Limit Theorem** Another approach is based on an empirical perspective under the quantization parameter depends only on the time index such that $Q_p(\varepsilon_\tau^q, \tau) = Q_p(\tau)$. Generally, we check the performance of algorithms at the unit of epoch, not the unit of the unspecified

index of a mini-batch. Accordingly, if there are sufficient numbers of mini-batches in an epoch and the quantization parameter is constant in a unit epoch such that $Q_p^{-1}(t_e + \tau) = Q_p^{-1}(t_e), \forall \tau_b \in \mathbb{Z}[0, B)$, we can analyze the summation of the learning equation to each mini-batch index as follows:

$$\boldsymbol{X}_{t_e+1}^Q = \boldsymbol{X}_{t_e}^Q - \lambda \sum_{\tau_b=0}^{B-1} \nabla_{\boldsymbol{x}} \tilde{f}_{\tau_b}(\boldsymbol{X}_{t_e+\tau_b/B}^Q) + b^{-\bar{p}(t_e)} \lambda \sqrt{\frac{C_q}{c_0}} \sum_{\tau_b=0}^{B-1} \boldsymbol{\varepsilon}_{t_e+\tau_b/B}^q, \tag{15}$$

where $Q_p^{-1}(t_e) = \lambda \sqrt{\frac{C_q}{c_0}} b^{-\bar{p}(t_e)}$. Herein, the summation of the factor for the quantization error converges to a Gaussian-distributed random variable such that $\sqrt{\frac{\lambda}{c_0}} \sum_{\tau_b=0}^{B-1} \boldsymbol{\varepsilon}_{t_e+\tau_b/B}^q \to \boldsymbol{z}_{t_e} \sim \mathcal{N}(\boldsymbol{z}; 0, \boldsymbol{I}_d)$ as $B \uparrow \infty$, by the central limit theorem. Therefore, we can regard the stochastic difference equation, $equation$ 15, as the stochastic integrated equation concerning $t_e$ to $t_e + 1$, so we can obtain an approximated SDE to the epoch-based continuous-time index $t \in \mathbb{R}^+$. We accept this approach as a quantization-based learning equation since (16) does not require any additional operation such as the transformation to generate a Gaussian random variable.

**Application to Other Learning Algorithms** In the quantized stochastic Langevin dynamics (**QSLD**), the search direction is not fixed as the opposite direction of the gradient vector, allowing for the application of various learning methods such as ADAM (Kingma and Ba (2015)) and alternative versions of ADAM such as ADAMW(Loshchilov and Hutter (2019)), NADAM(Dozat (2016)), and RADAM(Liu et al. (2020)).

**Avoid Early Paralysis of the proposed algorithm** The initial gradient tends to vanish if the initial search point is far from optimal, especially for objective functions in deep neural networks that utilize entropy-based loss functions such as the Kullback-Leibler divergence (KL-Divergence). The small gradient in the early stage becomes zero after the quantization process, potentially causing the deep neural network (DNN) to fall into a state of paralysis, as illustrated below: Assume that $\max \|\lambda h\| < 0.5\, Q_p^{-1}(\tau) - \delta$ for $\tau < \tau_0$, where $\delta$ denotes a positive value such that $\delta Q_p < 1$, and $\tau_0$ denotes a small positive integer. Then, we have

$$1/Q_p \cdot \|\lfloor Q_p(\lambda h + 0.5\, Q_p^{-1})\rfloor\| \leq 1/Q_p \cdot \lfloor Q_p(\max\|\lambda h\| + 0.5\, Q_p)\rfloor = 1/Q_p \cdot \lfloor 1 - \delta Q_p \rfloor = 0. \tag{16}$$

To prevent the paralysis depicted in $equation$ 16, a straightforward solution is to re-establish the quantized search direction by incorporating a compensation function $r(\tau) = \lfloor r(\tau)\rfloor$ into $h$, as shown:

$$\begin{aligned}
Q_p^{-1}(\tau) \cdot \|\lfloor Q_p(\tau)(\lambda h + r(\tau) + 0.5 Q_p^{-1})\rfloor\|_{\max\|\lambda h\| < 0.5\, Q_p^{-1}(\tau) - \delta} \\
= Q_p^{-1}(\tau) \cdot \|\lfloor Q_p \cdot (\lambda h + 0.5\, Q_p^{-1} + r(\tau))\rfloor\|_{\max\|\lambda h\| < 0.5\, Q_p^{-1}(\tau) - \delta} \\
\leq Q_p^{-1}(\tau) \cdot \|\lfloor Q_p \cdot (\lambda h + 0.5\, Q_p^{-1}\rfloor\|_{\max\|\lambda h\| < 0.5\, Q_p^{-1}(\tau) - \delta} + \|\lfloor Q_p r(\tau))\rfloor\| \\
\leq 0 + Q_p^{-1}(\tau) \cdot \|\lfloor Q_p r(\tau)\rfloor\|.
\end{aligned} \tag{17}$$

In $equation$ 17, the compensation function is responsible for increasing the magnitude of the search direction during an initial finite period. To address this, we propose the compensation function $r(\tau)$ given by:

$$r(\tau, \boldsymbol{X}_\tau) = \lambda \cdot \left( \frac{\exp(-\varkappa(\tau - \tau_0))}{1 + \exp(-\varkappa(\tau - \tau_0))} \cdot \frac{h(\boldsymbol{X}_\tau^Q)}{\|h(\boldsymbol{X}_\tau^Q)\|} \right), \quad \tau_0 \in \mathbb{Z}^{++}, \tag{18}$$

where $\varkappa > 0$ denotes a determining parameter for the working period, and $\tau_0$ represents the half-time of the compensation.

Moreover, the compensation function $r(\tau)$ provides the crucial property that the proposed quantization error is uncorrelated to the quantization input, such as the directional derivatives $h$ as follows:

**Theorem 3.1.** *Let the quantized directional derivatives $h^Q : \mathbb{Z}^+ \times \mathbb{R}^d \mapsto \mathbb{Q}^d$ such that*

$$h^Q(\boldsymbol{X}_\tau^Q) \triangleq \frac{1}{Q_p} \lfloor Q_p \cdot (\lambda h(\boldsymbol{X}_\tau^Q) + r(\tau, \boldsymbol{X}_\tau^Q)) + 0.5 \rfloor, \tag{19}$$

*where $r(\tau, \boldsymbol{X}_\tau)$ denotes a compensation function such that $r : \mathbb{Z}^+ \times \mathbb{R}^d \mapsto \mathbb{R}^d\{-1, 1\}$. Then, the quantization input $h(\boldsymbol{X}_\tau^Q)$ and the quantization error $\epsilon_\tau^q$ is uncorrelated such that $\mathbb{E}_{\epsilon_\tau^q}[h(\boldsymbol{X}_\tau^Q)\epsilon_\tau^q | h^Q(\boldsymbol{X}_\tau^Q) = kQ_p^{-1}] = 0$.*

Theorem 3.1 completes the assumptions for the WNH of the quantization error referred to in Assumption 2 for stochastic analysis for the proposed learning scheme.

### 3.2 CONVERGENCE PROPERTY OF QSGLD

**Weak Convergence without Convex Assumption** Before the weak convergence analysis of QSGLD, we establish the following lemma for the approximated SDE.

**Definition 3** (**Order-1 Weak Approximation, Li et al. (2019) and Malladi et al. (2022)**). *Let* $\{\boldsymbol{X}_t : t \in [0, t_e]\}$ *and* $\{\boldsymbol{X}_\tau^Q\}_{\tau=0}^{t_e \cdot B}$ *be families of continuous and discrete stochastic processes parameterized by* $\lambda$. *We regard* $\{\boldsymbol{X}_t\}$ *and* $\boldsymbol{X}_\tau$ *are order-1 weak approximations of each other if for all test function* $g$ *with polynomial growth, there exists a constant* $C_{o1} > 0$ *independent of* $\lambda$ *such that*

$$\max_{\tau \in \mathbf{Z}[0, \lfloor t_e \cdot B \rfloor]} |\mathbb{E}g(\boldsymbol{X}_t) - \mathbb{E}g(\boldsymbol{X}_{\lfloor \tau/B \rfloor}^Q)| \leq C_{o1}\lambda^2. \tag{20}$$

We provide additional definitions required in Definition 3, such as the polynomial growth of the test function in the supplementary material.

**Lemma 3.2.** *The approximated Langevin SDE for QSGLD represented in equation 15 is as follows:*

$$d\boldsymbol{X}_t = -\nabla_{\boldsymbol{x}} f(\boldsymbol{X}_t) dt + \sqrt{C_q} \cdot \sigma(t) d\boldsymbol{B}_t, \quad \forall t > t_0 \in \mathbb{R}^+, \because \sigma(t) \triangleq b^{-\bar{p}(t)}, \tag{21}$$

*The approximation* $equation$ *21 satisfies the order-1 weak approximation described in Definition* 3.

**Sketch of proof** Li et al. (2019) introduced a rigorous analytical framework for approximating stochastic difference equations with stochastic differential equations (SDEs). Building upon this framework, Malladi et al. (2022) extended the analysis for more general search directions. We leverage the aforementioned framework to establish Lemma 3.2. We impose several bounded assumptions described in the supplementary material to derive the moment of the one-step difference in $equation$ 21. Subsequently, by utilizing the bound of the moment, we prove the order-1 weak approximation of QSGLD to the Langevin SDE.

**Theorem 3.3.** *Consider the transition probability density, denoted as* $p(t, \boldsymbol{X}_t, t + \bar{\tau}, \boldsymbol{x}^*)$, *from an arbitrary state* $\boldsymbol{X}_t \in \mathbb{R}^d$ *to the optimal point* $\boldsymbol{x}^* \in \mathbb{R}^d$, $\boldsymbol{X}_t \neq \boldsymbol{x}^*$ *after a time interval* $\bar{\tau} \in \mathbb{R}^+$, *for all* $t > t_0$. *If the quantization parameter is bounded as follows:*

$$\sup_{t \geq 0} Q_p(t) \leq \sqrt{\frac{1}{C} \cdot \log(t + 2)}, \quad C \in \mathbb{R}^{++}, \tag{22}$$

*for* $\boldsymbol{X}_t \neq \bar{\boldsymbol{X}}_t$, *QSGLD represented as* $equation$ *21 converges with distribution in the sense of Cauchy convergence such that*

$$\overline{\lim_{\bar{\tau} \to \infty}} \sup_{\boldsymbol{X}_t, \bar{\boldsymbol{X}}_t \in \mathbb{R}^n} |p(t, \bar{\boldsymbol{X}}_t, t + \bar{\tau}, \boldsymbol{x}^*) - p(t, \boldsymbol{X}_t, t + \bar{\tau}, \boldsymbol{x}^*)| \leq \tilde{C} \cdot \exp\left(-\sum_{\bar{\tau}=0}^{\infty} \delta_{t+\bar{\tau}}\right), \tag{23}$$

*where* $\delta_t$ *denotes the infimum of the transition probability density from time* $t$ *to* $t + 1$ *given by* $\delta_t = \inf_{\boldsymbol{x}, \boldsymbol{y} \in \mathbb{R}^d} p(t, \boldsymbol{x}, t + 1, \boldsymbol{y})$, *satisfying* $\sum_{\bar{\tau}=0}^{\infty} \delta_{t+\bar{\tau}} = \infty$, *and* $\tilde{C}$ *denotes a positive value.*

**Sketch of proof** First, we analyze the limit supremum of the difference between the transition probabilities expressed by the infimum $\delta_t$. Next, according to the Girsanov theorem (Øksendal (2003) and Klebaner (2012)), we calculate the Radon-Nykodym derivative of the probability measure derived by the weak solution of Langevin SDE relevant to QSGLD concerning the probability of a standard Gaussian. Using the obtained Radon-Nykodym derivative, we calculate a lower bound for $\delta_t$ and provide proof for the theorem.

**Local Convergence under Convex Assumption** For local convergence analysis, we suppose that the objective function around the optimal point is strictly convex.

**Assumption 4.** *The Hessian of the objective function* $\boldsymbol{H}(f) : \mathbb{R}^d \mapsto \mathbb{R}^d$ *around the optimal point is non-singular and positive definite,*

**Theorem 3.4.** *The expectation value of the objective function derived by the proposed QSGLD converges to a locally optimal point asymptotically under Assumption* 4, *such that*

$$\forall \varepsilon > 0, \exists \rho > 0 \text{ such that } \|\boldsymbol{X}_\tau^Q - \boldsymbol{x}^*\| < \rho \implies |\mathbb{E}_{\boldsymbol{\epsilon}_\tau^q} f(\boldsymbol{X}_{\tau+k}^Q) - f(\boldsymbol{x}^*)| < \varepsilon(\rho). \tag{24}$$

**Sketch of proof** Intuitively, the proof of the theorem follows a similar structure to the conventional proof of gradient descent under the convex assumption. However, with the existence of the Brownian motion process, we apply the stationary probability from Theorem 3.3 into the proof. Accordingly, we prove the convergence not on the point-wise vector space, but on the function space induced by a stationary expectation.

---

**Algorithm 1** QSLD/QSGLD with the proposed quantization scheme

---

 1: **Initialization** $\tau \leftarrow 0$, $\boldsymbol{X}_0 \in \mathbb{Q}^d$      ▷ Set Initialize Discrete Time Index and state
 2: **repeat**
 3:      Compute $h(\boldsymbol{X}_\tau^Q)$ at $\tau$      ▷ Compute a Search Direction Vector
 4:      Compute $Q_p(\tau)$, and $r(\tau, h(\boldsymbol{X}_\tau^Q))$      ▷ Compute Quantization Parameter and $r(\tau)$
 5:      $h_\tau^Q \leftarrow \frac{1}{Q_p} \lfloor Q_p \cdot (-\lambda h(\boldsymbol{X}_\tau^Q) + r(\tau, h(\boldsymbol{X}_\tau^Q)) + 0.5 Q_p^{-1}) \rfloor$    ▷ Quantization of Search Vector
 6:      $\boldsymbol{X}_\tau^Q \leftarrow \boldsymbol{X}_\tau + h_\tau^Q$      ▷ General Updating Rule for Learning
 7:      $\tau \leftarrow \tau + 1$      ▷ General Update Discrete Time Index
 8: **until** Stopping criterion is met

---

Table 1: Comparison of test performance among optimizers with a fixed learning rate 0.01. Evaluation is based on the Top-1 accuracy of the training and testing data.

| Data Set | FashionMNIST | | | CIFAR10 | | | CIFAR100 | | |
|---|---|---|---|---|---|---|---|---|---|
| Model | CNN with 8-Layers | | | ResNet-50 | | | | | |
| Algorithms | Training | Testing | Training Error | Training | Testing | Training Error | Training | Testing | Training Error |
| QSGD | 97.10 | 91.59 | 0.085426 | 99.90 | 73.80 | 0.009253 | 99.04 | 37.77 | 0.030104 |
| QADAM | 98.43 | 89.29 | 0.059952 | 99.99 | 85.09 | 0.011456 | 98.62 | 49.60 | 0.037855 |
| SGD | 95.59 | 91.47 | 0.132747 | 99.99 | 63.31 | 0.001042 | 98.24 | 25.90 | 0.005478 |
| ASGD | 95.60 | 91.42 | 0.130992 | 99.99 | 63.46 | 0.001166 | 98.36 | 26.43 | 0.004981 |
| ADAM | 92.45 | 87.12 | 0.176379 | 99.75 | 82.08 | 0.012421 | 98.85 | 46.32 | 0.038741 |
| ADAMW | 91.72 | 86.81 | 0.182867 | 99.57 | 82.20 | 0.012551 | 98.86 | 47.01 | 0.038002 |
| NADAM | 96.25 | 87.55 | 0.140066 | 99.56 | 82.46 | 0.014377 | 98.62 | 48.56 | 0.037409 |
| RADAM | 95.03 | 87.75 | 0.146404 | 99.65 | 82.26 | 0.010526 | 98.17 | 48.61 | 0.044193 |

# 4 EXPERIMENTAL RESULTS

## 4.1 CONFIGURATION OF EXPERIMENTS

From a practical point of view, as both the updated state $\boldsymbol{X}_{\tau+1}^Q$ and the current state $\boldsymbol{X}\tau^Q$ are quantized vectors, the quantization parameter should be a power of the base defined in $equation$ 7. Although Theorem 3.3 provides an upper bound for the quantization parameter as a real value, we constrain the quantization parameter to be a rational number within a bounded range of real values. Specifically, we set the quantization parameter $Q_p$ as follows:

$$Q_p = \left\lfloor \sqrt{\frac{1}{C} \log(t_e + 2)} \right\rfloor, \tag{25}$$

where $t_e$ represents a unit epoch defined as $t_e = \lfloor \tau/B \rfloor$, and $\tau$ denotes a unit update index such that $\tau = t_e \cdot B + k$, $k \in \mathbb{Z}^+[0, B)$. In the supplementary material, we provide detailed explanations and methods for calculating and establishing the remaining hyper-parameters.

## 4.2 BRIEF INFORMATION OF EXPERIMENTS

We conducted experiments to compare QSGLD with standard SGD and ASGD (Shamir and Zhang (2013)). Additionally, we compared the performance of QSLD to the proposed method with ADAM, ADAMW, NADAM, and RADAM in terms of convergence speed and generalization. These results provide an empirical analysis and demonstrate the effectiveness of the proposed algorithms. The network models used are ResNet-50 and a small-sized CNN network with 3-layer blocks, of which two former blocks are a bottle-neck network, and one block is a fully connected network for classification. Data sets used for the experiments are FashionMNIST(Xiao et al. (2017)), CIFAR-10, and CIFAR-100. We give detailed information about the experiments, such as hyper-parameters of each algorithm, in the supplementary material.

## 4.3 EMPIRICAL ANALYSIS OF THE PROPOSED ALGORITHM

**Test for FashionMNIST** For the experiment, we employed a small-sized CNN to evaluate the performance of the test algorithms. The FashionMNIST data set consists of black-and-white images that are similar to MNIST data. Figure 2 illustrates that SGD algorithms demonstrate superior

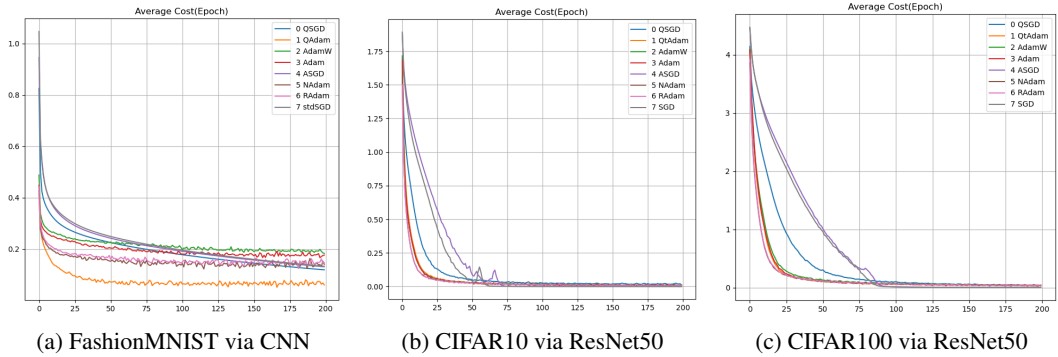

(a) FashionMNIST via CNN      (b) CIFAR10 via ResNet50      (c) CIFAR100 via ResNet50

Figure 2: The error trends of test algorithms to the data set and neural models: (a) Training error trends of the CNN model on FashionMNIST data set.(b) Training error trends of ResNet-50 on CIFAR-10 data set. (c) Training error trends of ResNet-50 on CIFAR-100 data set.

generalization performance on the test data, even though their convergence speed is slower than ADAM-based algorithms. Table 1 shows that the proposed quantization algorithms achieve a superior final accuracy error. Notably, the quantization algorithm applied to ADAM algorithms surpasses the error bound and achieves even lower accuracy error. As a result, the proposed quantization algorithm demonstrates improved classification performance for test data.

**Test for CIFAR-10 and CIFAR-100** For the CIFAR-10 and CIFAR-100 tests, we utilized ResNet-50 as the model architecture. As shown in Table 1, when classifying the CIFAR-10 data set using ResNet-50, QSGLD outperformed SGD by 8% in terms of test accuracy. As depicted in Figure 2, QSGLD exhibited significant improvements in convergence speed and error reduction compared to conventional SGD methods. On the other hand, Adam-Based QSLD showed a performance advantage of approximately 3% for test accuracy for the CIFAR-10 data set. Similar trends were observed for the CIFAR-100 data set. QSGLD demonstrated a performance advantage of around 11% over conventional SGD methods for test accuracy. In contrast, Adam-based QSLD showed an improvement of approximately 3.0% compared to Adam optimizers and 1.0% compared to NAdam and RAdam.

## 5 CONCLUSION

We introduce two stochastic descent learning equations, QSLD and QSGLD, based on quantized optimization. Empirical results demonstrate that the error trends of the quantization-based stochastic learning equations exhibit stability compared to other noise-injecting algorithms. We provide the proposed algorithm's weak and local convergence properties with the Langevin SDE perspective. In our view, exploring the potential of quantization techniques applied to the range of objective functions may offer limited performance gains. Hence, future research will investigate the quantization technique in the context of objective function domains. Furthermore, we have not analyzed a more generalized version of the proposed method, QSLD, despite the empirical results suggesting its satisfactory consistency. Analyzing QSLD requires studying the properties of the general search direction, which we reserve for future investigations.

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

## A APPENDIX : INTRODUCTION

We provide the mathematical properties and the proofs for the theorems and lemmas featured in this manuscript. Firstly, we present the statistical properties of the quantization error, such as expectation, variance, and independent properties. Moreover, we present the proof of Theorem 3.1 to provide the independence of the quantization error to the directional derivation $h$ for the early stage of learning.

Next, we provide the discrete variation of the proposed quantized optimization-based learning equation for stochastic analysis. Such discrete learning formulae show how can we transform the quantization error into a standard Wiener process with the auxiliary function in the quantization parameter.

Based on the provided discrete equation, we prove that the proposed learning equation weakly converges to the stochastic differential equation known as the Langevin equation. Using the weakly converged stochastic differential equation, we give the proof of Theorem 3.3 about the weak convergence of the proposed algorithm. Through the property of weak convergence, we posit that the proposed algorithm possesses a stationary distribution of the transition probability density towards an optimal point, even in the presence of non-convex objective functions.

Additionally, we establish the proposed algorithm's asymptotic convergence under a convex assumption. Despite the non-convexity assumptions, we provide proof of local convergence by considering the feasibility of assuming strict local convexity around an optimal point.

Furthermore, we provide detailed information regarding the experiments, including comprehensive results of simulations for each data set, network models utilized, and the corresponding hyperparameters for each algorithm. Finally, we offer further discussion and analysis of the specific empirical findings in these experiments.

### A.1 NOTATIONS

Generally, we follow the mathematical notations shown in the manuscript provided by the ICLR formatting instructions. Herein, we provide some combinations of the notations for easy reading and particular expressions used in the manuscript.

- $\mathbb{R}^n$  The n-dimensional space with real numbers.
- $\mathbb{R}$  $\mathbb{R}^n|_{n=1}$.
- $\mathbb{R}[\alpha, \beta]$  $\{x \in \mathbb{R} | \alpha \le x \le \beta, \ \alpha, \beta \in \mathbb{R}\}$.
- $\mathbb{R}(\alpha, \beta]$  $\{x \in \mathbb{R} | \alpha < x \le \beta, \ \alpha, \beta \in \mathbb{R}\}$.
- $\mathbb{R}[\alpha, \beta)$  $\{x \in \mathbb{R} | \alpha \le x < \beta, \ \alpha, \beta \in \mathbb{R}\}$.
- $\mathbb{R}(\alpha, \beta)$  $\{x \in \mathbb{R} | \alpha < x < \beta, \ \alpha, \beta \in \mathbb{R}\}$.
- $\mathbb{Q}^n$  The n-dimensional space with rational numbers.
- $\mathbb{Q}$  $\mathbb{Q}^n|_{n=1}$.
- $\mathbb{Z}$  The 1-dimensional space with integers.
- $\mathbb{N}$  The 1-dimensional space with natural numbers.
- $\mathbb{R}^+$  $\{x | x \ge 0, \ x \in \mathbb{R}\}$.
- $\mathbb{R}^{++}$  $\{x | x > 0, \ x \in \mathbb{R}\}$.
- $\mathbb{Q}^+$  $\{x | x \ge 0, \ x \in \mathbb{Q}\}$.
- $\mathbb{Q}^{++}$  $\{x | x > 0, \ x \in \mathbb{Q}\}$.
- $\mathbb{Z}^+$  $\{x | x \ge 0, \ x \in \mathbb{Z}\}$.
- $\mathbb{Z}^{++}$  $\{x | x > 0, \ x \in \mathbb{Z}\}$, $\mathbb{Z}^{++}$ is equal to $\mathbb{N}$.
- $\lfloor x \rfloor$  $\max\{y \in \mathbb{Z} | y \le x\}$, for a given number $x \in \mathbb{R}$.
- $\lceil x \rceil$  $\min\{y \in \mathbb{Z} | y \ge x\}$, for a given number $x \in \mathbb{R}$.
- $n_{B_\tau}$  The numbers of samples in the $\tau$-th mini-batch $B_\tau$.

- $B$  The total number of mini-batches. If we have totally $N_T$ numbers of samples for a dataset, we have $N_T = \sum_{\tau=1}^{B} n_{B_\tau}$.

- $k \in \mathbb{Z}^+$  Discrete index for sample data.

- $t \in \mathbb{R}^+$  Continuous time index for stochastic analysis.

- $\tau \in \mathbb{Z}^+$  Discrete time index. Typically it denotes the monotonically increasing index based on the unit of mini-batches such that $\tau \uparrow \infty$ and $\tau = t_e \cdot B$.

- $t_b \in \mathbb{Z}^+[0, B)$  Discrete index to mini-batch.

- $t_e \in \mathbb{Z}^+$  Discrete time index. Typically it denotes the epoch, corresponding to $\tau$ such that $t_e = \tau/B$.

- $P_X(x)$  Probability density function of the discrete random variable $X$ with respect to a domain $\mathcal{D}(x)$ containing $x$.

- $p_X(x)$  Probability density function of the continuous random variable $X$ with respect to a domain $\mathcal{D}(x)$ containing $x$.

- $\mathbb{E}_X(X)$  Expectation value of the random variable $X$ following the distribution $P_X(x)$ such that $\mathbb{E}_X(X) = \int_{\mathcal{D}(x)} x P_X(x) dx$, where $\mathcal{D}(x)$ is a domain containing $x$. More detailed, $\mathbb{E}_X(Y(x))$ is equivalent to $\mathbb{E}(Y(x)|X) \triangleq \int_{-\infty}^{\infty} y(x) P_X(x) dx$ Accordingly, $\mathbb{E}_{x \sim P_X}$ is an equivalent expression.

- $\langle \boldsymbol{a}, \boldsymbol{b} \rangle$  The inner product of the vector $\boldsymbol{a} \in \mathbb{R}^d$ and $\boldsymbol{b} \in \mathbb{R}^d$. This notation is equivalent to $\boldsymbol{a} \cdot \boldsymbol{b}$.

- $\mathcal{U}(\boldsymbol{x}; \mu, \sigma)$  Uniform distribution with the expectation $\mu$, the variance $\sigma$. The random variable $\boldsymbol{x}$ follows the uniform distribution. Additionally, we employ a notation without the random variable such as $\mathcal{U}(\mu, \sigma)$.

- $\mathcal{N}(\boldsymbol{x}; \mu, \sigma)$  Normal distribution with the expectation $\mu$, the variance $\sigma$. The random variable $\boldsymbol{x}$ follows the normal distribution. Additionally, we employ a notation without the random variable such as $\mathcal{N}(\mu, \sigma)$.

## A.2 Additional Definitions and Assumptions

We refer to the definitions and assumptions described in the main manuscript to all theorems and lemmas in the supplementary material. However, we didn't mention the following definition for a measure of a matrix, so that we define it as follows:

**Definition 4.** For $\boldsymbol{A} \in \mathbb{R}^{m \times m}, m \in \mathbb{N}$, we define a metric for a matrix $A$ as follows:

$$(\boldsymbol{A})_{\mathbb{R}^{m \times m}} \triangleq \langle \boldsymbol{v}, \boldsymbol{A}\boldsymbol{v} \rangle, \quad \boldsymbol{v} \in \mathbb{R}^m, \ \|\boldsymbol{v}\| = 1. \tag{26}$$

We define the matrix norm of $\boldsymbol{A}$, denoted as $\|\boldsymbol{A}\|_{\mathbb{R}^{m \times m}}$, using the absolute value of the metric in equation 26 such that $\|\boldsymbol{A}\|_{\mathbb{R}^{m \times m}} \triangleq \sup |(\boldsymbol{A})_{\mathbb{R}^{m \times m}}|$.

## B Statistical Properties of Quantization

### B.1 Proof of Basic Statistical Properties

**Fundamental Statistical Properties of the Quantization error**

From Definition 1 and Assumption 2, we note that $\epsilon^q = Q_p^{-1} \varepsilon^q$ and $p_{\epsilon^q}(x) = Q_p$ for $\mathbb{R}[-\frac{1}{2Q_p}, \frac{1}{2Q_p})$, so that we can obtain the differential $d\epsilon^q = Q_p^{-1} d\varepsilon^q$ and the integral domain for $\varepsilon^q$ such that $\mathcal{D}(\varepsilon^q) = [-\frac{1}{2}, \frac{1}{2})$. It implies that

$$\mathbb{E}_{\epsilon^q} \epsilon^q = \mathbb{E}_{\epsilon^q} Q_p^{-1} \varepsilon^q = \int_{-\frac{1}{2Q_p}}^{\frac{1}{2Q_p}} p_{\epsilon^q}(x) \epsilon^q d\epsilon^q = \int_{-\frac{1}{2}}^{\frac{1}{2}} Q_p Q_p^{-1} \varepsilon^q Q_p^{-1} d\varepsilon^q = \frac{Q_p^{-1}}{2} \varepsilon^{q2} \bigg|_{-\frac{1}{2}}^{\frac{1}{2}} = 0. \tag{27}$$

Therefore, we simply evaluate the expectation value of the quantized input $X^Q \in \mathbb{Q}$ such that

$$
\mathbb{E}_{\epsilon^q}[X^Q|X=x] = \mathbb{E}_{\epsilon^q}[X^Q|X=x]
$$

$$
= \int_{-\frac{\Delta}{2}}^{\frac{\Delta}{2}} p_{\epsilon^q}(\epsilon^q)\Delta\left\lfloor \frac{x}{\Delta} + \frac{1}{2} \right\rfloor d\epsilon^q = \int_{-\frac{1}{2Q_p}}^{\frac{1}{2Q_p}} Q_p \cdot (x + \varepsilon^q Q_p^{-1}) d\epsilon^q = \int_{-\frac{1}{2Q_p}}^{\frac{1}{2Q_p}} Q_p \cdot (x + \epsilon^q) d\epsilon^q
$$

$$
= Q_p\left( \int_{-\frac{1}{2Q_p}}^{\frac{1}{2Q_p}} x\, d\epsilon^q + \int_{-\frac{1}{2Q_p}}^{\frac{1}{2Q_p}} \epsilon^q d\epsilon^q \right) = Q_p\left( x\int_{-\frac{1}{2Q_p}}^{\frac{1}{2Q_p}} d\epsilon^q + \left. \frac{\epsilon^{q2}}{2} \right|_{-\frac{1}{2Q_p}}^{\frac{1}{2Q_p}} \right)
$$

$$
= Q_p\left( x\frac{1}{Q_p} + \frac{1}{2}\left( \frac{1}{4Q_p^2} - \frac{1}{4Q_p^2} \right) \right)
$$

$$
= x.
\tag{28}
$$

We can rewrite equation 28 to the following simplified formulae:

$$
\mathbb{E}_{\epsilon^q}[X^Q|X=x] = \mathbb{E}_{\epsilon^q}[X + \epsilon^q|X=x] = \mathbb{E}_{\epsilon^q}[x + \epsilon^q] = x + \mathbb{E}_{\epsilon^q}\epsilon^q = x,
\tag{29}
$$

where $x \in \mathbb{R}$ is a scalar value and $X \in \mathbb{R}$ is a scalar random variable as an input of quantization. Further, we can obtain the variance of the quantization error as follows:

$$
Var_{\epsilon^q}\epsilon^q = \mathbb{E}_{\epsilon^q}(\epsilon^q)^2 - (\mathbb{E}_{\epsilon^q}\epsilon^q)^2 = \mathbb{E}_{\epsilon^q}(\epsilon^q)^2 = \mathbb{E}_{\epsilon^q}Q_p^{-2}\varepsilon^{q2}
$$

$$
= \int_{-\frac{1}{2}}^{\frac{1}{2}} Q_p Q_p^{-2}\varepsilon^{q2}Q_p^{-1}d\varepsilon^q = \left. \frac{Q_p^{-2}}{3}\varepsilon^{q3} \right|_{-\frac{1}{2}}^{\frac{1}{2}} = \frac{1}{12Q_p^2} = c_0 Q_p^{-2}.
\tag{30}
$$

**Independent Condition of the Quantization error** The independent condition mentioned herein is relevant whether the correlation between the input of the quantization and the quantization error is zero or not. If we let a quantization value $X^Q = kQ_p^{-1} \in \mathbb{Q}$ for $k \in \mathbb{Z}$, we can evaluate the correlation of $X^Q$ and the quantization error $\bar{\epsilon}^q$ such that

$$
\mathbb{E}_{\bar{\varepsilon}^q}[X(X - X^Q)|X^Q = kQ_p^{-1}] = \mathbb{E}_{\bar{\varepsilon}^q}[X\bar{\epsilon}^q|X^Q = kQ_p^{-1}]
$$

$$
= \int_{-\frac{1}{2}Q_p^{-1}}^{\frac{1}{2}Q_p^{-1}} Q_p x\epsilon^q d\bar{\epsilon}^q, \quad \because x = x^Q|_{x^Q = kQ_p^{-1}} + \bar{\epsilon}^q\Big|_{\bar{\epsilon}^q = -Q_p^{-1}/2}^{\bar{\epsilon}^q = Q_p^{-1}/2}, \ dx = d\bar{\epsilon}^q
$$

$$
= \int_{(-\frac{1}{2}+k)Q_p^{-1}}^{(\frac{1}{2}+k)Q_p^{-1}} Q_p x(x - x^Q) dx
$$

$$
= Q_p\left[ \left. \frac{x^3}{3} \right|_{(-\frac{1}{2}+k)Q_p^{-1}}^{(\frac{1}{2}+k)Q_p^{-1}} - \frac{k}{Q_p}\left. \frac{x^2}{2} \right|_{(-\frac{1}{2}+k)Q_p^{-1}}^{(\frac{1}{2}+k)Q_p^{-1}} \right]
\tag{31}
$$

$$
= Q_p\left[ \frac{1}{3Q_p^3}\left( \left(\frac{1}{2}+k\right)^3 - \left(-\frac{1}{2}+k\right)^3 \right) - \frac{k}{2Q_p^3}\left( \left(\frac{1}{2}+k\right)^2 - \left(-\frac{1}{2}+k\right)^2 \right) \right]
$$

$$
= Q_p^{-2}\left[ k^2 + \frac{1}{12} - k^2 \right] = \frac{1}{12}Q_p^{-2} = c_0 Q_p^{-2}.
$$

In equation 31, to get a positive correlation value, we establish the quantization value $X^Q$ and quantization input $X$ as the following equation:

$$
X = X^Q + \bar{\epsilon}^q, \ \bar{\epsilon}^q = Q_P^{-1}\bar{\varepsilon}^q \in \mathbb{R}[-Q_p^{-1}/2, Q_p^{-1}/2],
\tag{32}
$$

where the scalar quantization error $\bar{\epsilon}^q$ has negative sign to originally defined quantization error $\epsilon^q$ such that $\bar{\epsilon}^q = -\epsilon^q$.

Likewise as previously mentioned, we can rewrite equation 31 as follows

$$
\mathbb{E}_{\bar{\varepsilon}^q}[X\bar{\epsilon}^q|X^Q = kQ_p^{-1}] = \mathbb{E}_{\bar{\varepsilon}^q}[(X^Q + \bar{\epsilon}^q)\bar{\epsilon}^q|X^Q = kQ_p^{-1}] = \mathbb{E}_{\bar{\varepsilon}^q}[(kQ_p^{-1} + \bar{\epsilon}^q)\bar{\epsilon}^q]
$$

$$
= kQ_p^{-1}\mathbb{E}_{\bar{\varepsilon}^q}\bar{\epsilon}^q + \mathbb{E}_{\bar{\varepsilon}^q}(\bar{\epsilon}^q)^2 = 0 + c_0 Q_p^{-2}
\tag{33}
$$

The result of equation 31 represents that the quantization input and quantization error are correlated with the equal variance of the quantization error, so those are not generally independent. This correlation enables to bring about early paralysis in the learning process under quantized directional derivation. Considering equation 31, the quantized directional derivative $h^Q \in \mathbb{R}$ can be zero, even if there exists a quantization error such that

$$\mathbb{E}_{\bar{\epsilon}^q}[h\epsilon^q|h^Q = kQ_p^{-1}]\Big|_{k=0} = \int_{-\frac{1}{2Q_p}}^{\frac{1}{2Q_p}} Q_p h(h - h^Q)dx\Big|_{h^Q=0} = c_0 Q_p^{-2}. \tag{34}$$

If the quantization parameter is sufficiently large and leads $Q_p^{-1} \downarrow 0$, we can treat the quantization error as an independent white noise to the quantization input. However, the quantization parameter is not large enough in the early stage of the learning process, so such a relatively small quantization parameter raises the early paralysis of the learning algorithm. Considering the learning equation based on the quantized directional derivative such that $X_{\tau+1}^Q = X_\tau + h^Q$, we can note that the dependent property between the quantization error and the quantization input leads to the diminishing of the quantized directional derivative such that

$$|h| \downarrow 0 \implies |\epsilon^q| \downarrow 0 \implies |h + \epsilon^q| = |h^Q| \downarrow 0. \tag{35}$$

Accordingly, the learning process does not work when we devise a learning equation based on the quantized directional derivation. However, if there exists a compensated quantization $\breve{\epsilon}^q$ that is independent of the input $X$ such that $\mathbb{E}_{\breve{\epsilon}^q}[X\breve{\epsilon}^q|X^Q = kQ_p^{-1}] = 0$, the independent quantization error provides

$$|h| \downarrow 0 \text{ and } |\breve{\epsilon}^q| > 0 \implies |h + \breve{\epsilon}^q|_{|h|\downarrow 0} \leq |h|_{|h|\downarrow 0} + |\breve{\epsilon}| \triangleq |h^Q|_{|h|\downarrow 0} > 0 \tag{36}$$

This intuitive consideration states that the independent quantization error to the quantization input enables to avoidance of early learning paralysis raised by $|X| \downarrow 0$. Consequently, we should establish a compensation function that enables it to be independent between the quantization input and the quantization error.

### B.2 Avoid Early Paralysis of the proposed algorithm

**Dithering for Independent Condition of the Quantization error** We assume that there exists a random variable $\boldsymbol{z} \in \mathbb{R}$ defined on $\mathbb{R}[-\frac{1}{2Q_p}, \frac{1}{2Q_p})$ with a uniform distribution $p_{\boldsymbol{z}}(x) = Q_p$. Since the probability density function of the random variable $\boldsymbol{z}$ and $\epsilon^q$ is equal, we calculate the expectation such that

$$\mathbb{E}_{\boldsymbol{z}}[(X + \boldsymbol{z})^Q|X = x] = \int_{\frac{-1}{2Q_p}}^{\frac{1}{2Q_p}} p_{\boldsymbol{z}}(z) \cdot (x + z + \epsilon^q) \, dz$$

$$= \int_{x-\frac{1}{2Q_p}}^{x+\frac{1}{2Q_p}} Q_p(y + \epsilon^q)dy, \; \because y = x + z \implies dy = dz, \; y \in \mathbb{R}\left[x - \frac{1}{2Q_p}, x + \frac{1}{2Q_p}\right]$$

$$= Q_p\left(\int_{x-\frac{1}{2Q_p}}^{x+\frac{1}{2Q_p}} ydy + \int_{x-\frac{1}{2Q_p}}^{x+\frac{1}{2Q_p}} \epsilon^q dy\right) \; \because y^Q + \epsilon = y \implies d\epsilon = dy$$

$$= Q_p\left(\frac{y^2}{2}\Big|_{x-\frac{1}{2Q_p}}^{x+\frac{1}{2Q_p}} ydy + \int_{-\frac{1}{Q_p}}^{\frac{1}{Q_p}} \epsilon^q d\epsilon^q\right) \; \because \epsilon^q = x \pm \frac{1}{2Q_p} - y^Q$$

$$= Q_p\left(\frac{1}{2}\left[\left(x + \frac{1}{2Q_p}\right)^2 - \left(x + \frac{1}{2Q_p}\right)^2\right] + \frac{\epsilon^{q2}}{2}\Big|_{-\frac{1}{Q_p}}^{\frac{1}{Q_p}}\right)$$

$$\because \epsilon^q \in \{\epsilon^q \in \mathbb{R}|x - (x \pm 1/2Q_p) \pm 1/2Q_p\} = \mathbb{R}[-1/Q_p, 1/Q_p]$$

$$= Q_p\left(\frac{1}{2}\frac{2x}{Q_p} + \frac{\epsilon^{q2}}{2}\Big|_{-\frac{1}{2Q_p}}^{\frac{1}{2Q_p}}\right) = Q_p \cdot \frac{x}{Q_p} + Q_p \cdot 0 = x$$

$$\tag{37}$$

From the result of equation 37, we can obtain the correlation between the quantization input and the quantization error with an additional uniformly distributed noise as follows:

$$
\begin{aligned}
&\mathbb{E}_{X,\boldsymbol{z}}[X(X-(X+\boldsymbol{z})^Q)|X^Q = kQ_p^{-1}] \\
&= \int_{-(\frac{1}{2}+k)Q_p^{-1}}^{(\frac{1}{2}+k)Q_p^{-1}} p_X(x) \int_{-\frac{Q_p^{-1}}{2}}^{\frac{Q_p^{-1}}{2}} p_{\boldsymbol{z}}(z)x(x-(x+z)^Q)dzdx \\
&= \int_{-(\frac{1}{2}+k)Q_p^{-1}}^{(\frac{1}{2}+k)Q_p^{-1}} p_X(x) \left( x^2 \int_{-\frac{Q_p^{-1}}{2}}^{\frac{Q_p^{-1}}{2}} p_{\boldsymbol{z}}(z)dz - x \int_{-\frac{Q_p^{-1}}{2}}^{\frac{Q_p^{-1}}{2}} p_{\boldsymbol{z}}(z)(x+z)^Q dz \right) dx \\
&= \int_{-(\frac{1}{2}+k)Q_p^{-1}}^{(\frac{1}{2}+k)Q_p^{-1}} p_X(x)(x^2 - x^2)dx = 0.
\end{aligned}
\tag{38}
$$

By the result of equation 38, we are aware that the additional noise $\boldsymbol{z}$ enables the quantization error and the quantization input to be independent. The technique described in equation 38 is known as the dithering to fulfill the independent property among the quantization error and the quantization input(provided by Marco and Neuhoff (2005), and Gray and Neuhoff (2006)).

Whereas the dithering employs an additive noise with a uniform distribution herein, equation 12 presents that even an additive noise with an appropriate symmetrical distribution can satisfy the independent condition. Based on such a conjecture, we present the compensation function to satisfy the independent condition for the quantization error in the following section.

When we elaborate the above equations with the formulas incorporating the expectation symbol, we can rewrite equation 37 such that

$$
\mathbb{E}_{\boldsymbol{z}}[(X+\boldsymbol{z})^Q|X=x] = \mathbb{E}_{\boldsymbol{z}}[x+\boldsymbol{z}-\bar{\epsilon}^q] = x + \mathbb{E}_{\boldsymbol{z}}\boldsymbol{z} - \mathbb{E}_{\boldsymbol{z}}\bar{\epsilon}^q = x, \; \therefore \mathbb{E}_{\boldsymbol{z}}\boldsymbol{z} = \mathbb{E}_{\boldsymbol{z}}\bar{\epsilon}^q
\tag{39}
$$

The result of equation 39 presents that we can measure the quantization error $\bar{\epsilon}^q$ or $\epsilon^q$ with the probability density of $\boldsymbol{z}$ due to the equal uniform distribution. Hence we can regard the additional noise for the dithering $\boldsymbol{z}$ as a transformation of the quantization error. Additionally, we note that the additional noise as the transformation for the dithering requires the changed sign of the quantization error.

Finally, holding the dithering condition for additional noise, we rewrite equation 38 such that

$$
\begin{aligned}
\mathbb{E}_{X,\boldsymbol{z}}[X(X-(X+\boldsymbol{z})^Q)|X^Q = kQ_p^{-1}] &= \mathbb{E}_{X,\boldsymbol{z}}[X(X-(X+\boldsymbol{z}+\epsilon^q))|X^Q = kQ_p^{-1}] \\
&= \mathbb{E}_{X,\boldsymbol{z}}[X^2 - X^2 - X(\boldsymbol{z}+\epsilon^q)|X^Q = kQ_p^{-1}] \\
&= \mathbb{E}_{X,\boldsymbol{z}}[-X(\boldsymbol{z}+\epsilon^q)|X = kQ_p^{-1}+\bar{\epsilon}^q] \\
&= -\mathbb{E}_{\boldsymbol{z}}[(kQ_p^{-1}+\bar{\epsilon}^q)(\boldsymbol{z}+\epsilon^q)] \\
&= -kQ_p^{-1}\left(\mathbb{E}_{\boldsymbol{z}}\boldsymbol{z}+\mathbb{E}_{\boldsymbol{z}}\epsilon^q\right) - \mathbb{E}_{\boldsymbol{z}}\bar{\epsilon}^q\boldsymbol{z} - \mathbb{E}_{\boldsymbol{z}}\bar{\epsilon}^q\epsilon^q \\
&= 0 - \mathbb{E}_{\boldsymbol{z}}(\boldsymbol{z})^2 + \mathbb{E}_{\epsilon^q}(\epsilon^q)^2 = 0.
\end{aligned}
\tag{40}
$$

In equation 40, we note that $\mathbb{E}_{\boldsymbol{z}}\bar{\epsilon}^q\boldsymbol{z} = \mathbb{E}_{\boldsymbol{z}}(\boldsymbol{z})^2$ and $\mathbb{E}_{\boldsymbol{z}}\bar{\epsilon}^q\epsilon^q = -\mathbb{E}_{\boldsymbol{z}}(\epsilon^q)^2 = -\mathbb{E}_{\epsilon^q}(\epsilon^q)^2$.

**The Compensation Function for Early Paralysis** We establish the compensation function to avoid early paralysis in the quantized directional derivative-based learning process as follows:

$$
r(\tau, \boldsymbol{X}_\tau) = \lambda \cdot \left( \frac{\exp(-\varkappa(\tau-\tau_0))}{1+\exp(-\varkappa(\tau-\tau_0))} \cdot \frac{h(\boldsymbol{X}_\tau)}{\|h(\boldsymbol{X}_\tau)\|} \right), \quad \tau_0 \in \mathbb{Z}^{++}
\tag{41}
$$

We consider a directional derivative with one dimension, $h \in \mathbb{R}$, for convenience of discussing. Since the compensation function defined in equation 41 contains a normalized directional derivative $h/\|h\|$, for $\tau \ll \tau_0$, we assume that the range of the compensation function such that

$$
r(\tau, \boldsymbol{X}_\tau) \triangleq \lambda \cdot sgn(\boldsymbol{X}_\tau), \quad \forall \tau \ll \tau_0, \text{ and } \tau \gg \tau_0,
\tag{42}
$$

where $sgn$ denotes the sign function of each element in $\boldsymbol{X}_\tau$ such that $sgn(\boldsymbol{X}) = \sum_{i=1}^d sgn(\boldsymbol{X} \cdot \boldsymbol{e}^{(i)})\boldsymbol{e}^{(i)}$.

To analyze the effectiveness of the compensation function, we suppose the scalar input such that $X_\tau \in \mathbb{R}$. Intuitively, since we assume that the probability density function of the compensation function is an equal Bernoulli distribution $P_{\boldsymbol{r}}(x)$ with the value of $\{-\lambda, \lambda\}$ for the domain $\{x|x < 0\}$ and $\{x|x \geq 0\}$, respectively, we can straightforwardly obtain the expectation and variance of the compensation function for a $\tau$ such that $|\tau - \tau_0| \gg 0$ as follows:

$$
\mathbb{E}_r r(\tau, X) = \sum_{r=-\lambda}^{r=\lambda} P_{\boldsymbol{r}}(x) r(\cdot, X) = 0,
$$

$$
Var_r r(\tau, X) = \mathbb{E}_r r^2(\tau, X) = \lambda^2 \int_{\mathbb{R}} P_{\boldsymbol{r}}(x) dx = \lambda^2 \tag{43}
$$

$$
\mathbb{E}_r r(\tau, X) r(s, X) = 0, \forall t \neq s, \ |s - \tau_0| \gg 0.
$$

Moreover, the statistical quantity of the compensation function exhibits that the summation of $r(\tau, X)$, i.e., $Y_\tau \triangleq \sum_{k=0}^{\tau} r(\tau, X), \forall |\tau - \tau_0| \gg 0$, yields a standard Wiener process in the sense of the following moments:

$$
\mathbb{E}_r Y_\tau = \sum_{k=0}^{\tau-1} \mathbb{E}_r r(k, X) = 0, \ \ \mathbb{E}_r Y_\tau^2 = \sum_{k=0}^{\tau-1} \mathbb{E}_r r^2(k, X) + 2 \sum_{k=0}^{\tau-1} \sum_{l, l \neq k}^{\tau-1} r(k, X) r(s, X) = \lambda^2 \tau. \tag{44}
$$

Holding the definition of the quantization, we can write the quantization of the summation of the directional derivative and the compensation such that

$$
\begin{aligned}
(\lambda h(X) + r(\tau, X))^Q &= Q_p^{-1} \left[ Q_p \cdot (\lambda h(X) + r(\tau, X)) + \varepsilon^q \right] \\
&= \lambda h(X) + \varepsilon^q Q_p^{-1} + r(\tau, X) \\
&= (\lambda h(X))^Q + r(\tau, X) = (\lambda h(X))^Q + \lambda \cdot sgn(h(x))
\end{aligned} \tag{45}
$$

Finally, we can expand the statistical quantities to the vector-valued compensation $r : \mathbb{R} \times \mathbb{R}^d \mapsto \mathbb{R}^d$ with the following equation:

$$
r(\tau, \boldsymbol{X}) = \sum_{i=1}^{d} (r(\tau, \boldsymbol{X}) \cdot \boldsymbol{e}^i) \boldsymbol{e}^i, \quad \forall \boldsymbol{X} \in \mathbb{R}^d \tag{46}
$$

Thus, we can obtain the statistical moments of the vector-valued compensation for all $\boldsymbol{X} \in \mathbb{R}^d$ and $|\tau - \tau_0| \gg 0$ as follows:

$$
\begin{aligned}
\mathbb{E}_r r(\tau, \boldsymbol{X}) &= \sum_{i=1}^{d} \left( \mathbb{E}_r [r(\tau, \boldsymbol{X}) \cdot \boldsymbol{e}^i] \right) \boldsymbol{e}^i = \sum_{i=1}^{d} 0 \cdot \boldsymbol{e}^i = 0 \\
\mathbb{E}_r r^2(\tau, \boldsymbol{X}) &= \mathbb{E}_r \left( \sum_{i=1}^{d} (r(\tau, \boldsymbol{X}) \cdot \boldsymbol{e}^i) \boldsymbol{e}^i \right) \cdot \left( \sum_{j=1}^{d} (r(\tau, \boldsymbol{X}) \cdot \boldsymbol{e}^j) \boldsymbol{e}^j \right) \\
&= \mathbb{E}_r \left( \sum_{i=1}^{d} \sum_{j=1}^{d} (r(\tau, \boldsymbol{X}) \cdot \boldsymbol{e}^i)(r(\tau, \boldsymbol{X}) \cdot \boldsymbol{e}^j) \boldsymbol{e}^i \cdot \boldsymbol{e}^j \right) \\
&= \mathbb{E}_r \sum_{i=1}^{d} (r(\tau, \boldsymbol{X}) \cdot \boldsymbol{e}^i)^2 = \lambda^2 d
\end{aligned} \tag{47}
$$

where $\boldsymbol{I}_d$ denotes an identity matrix such that $\boldsymbol{I}_d \in \mathbb{R}^{d \times d}$. Henceforth, we prove the following theorem using the derived statistical properties of the proposed compensation function.

**Theorem 3.1** Let the quantized directional derivatives $h^Q : \mathbb{Z}^+ \times \mathbb{R}^d \mapsto \mathbb{Q}^d$ such that

$$
h^Q(\boldsymbol{X}_\tau^Q) \triangleq \frac{1}{Q_p} \lfloor Q_p \cdot (\lambda h(\boldsymbol{X}_\tau^Q) + r(\tau, \boldsymbol{X}_\tau^Q)) + 0.5 \rfloor, \tag{48}
$$

where $r(\tau, \boldsymbol{X}_\tau)$ denotes a compensation function such that $r : \mathbb{Z}^+ \times \mathbb{R}^d \mapsto \mathbb{R}^d\{-1, 1\}$. Then, the quantization input $h(\boldsymbol{X}_\tau^Q)$ and the quantization error $\boldsymbol{\epsilon}_\tau^q$ is uncorrelated when the quantized directional derivative $h(\boldsymbol{X}_\tau^Q)$ is 0 such that $\mathbb{E}_{\boldsymbol{\epsilon}_\tau^q}[h(\boldsymbol{X}_\tau^Q)\boldsymbol{\epsilon}_\tau^q | h^Q(\boldsymbol{X}_\tau^Q) = kQ_p^{-1}] = 0$.

*Proof.* From the definition of the quantization error $\bar{\boldsymbol{\epsilon}}_\tau^q = -\boldsymbol{\epsilon}_\tau^q = \boldsymbol{X}_\tau - \boldsymbol{X}_\tau^Q$ and substituting $h(\boldsymbol{X}_\tau^Q)$, $(h(\boldsymbol{X}_\tau) + r(\tau, h(\boldsymbol{X}_\tau)))^Q$ into $X_\tau$, $X_\tau^Q$ respectively, we can rewrite the expectation $\mathbb{E}_{\boldsymbol{X}, r, \boldsymbol{\epsilon}_\tau^q}[h(\boldsymbol{X}_\tau^Q)\boldsymbol{\epsilon}_\tau^q | h^Q(\boldsymbol{X}_\tau^Q) = kQ_p^{-1}]$ such that

$$
\begin{aligned}
&\mathbb{E}_{\boldsymbol{X}, r, \boldsymbol{\epsilon}_\tau^q}[h(\boldsymbol{X}_\tau^Q)\boldsymbol{\epsilon}_\tau^q | h^Q(\boldsymbol{X}_\tau^Q) = kQ_p^{-1}] \\
&= \mathbb{E}_{\boldsymbol{X}, r, \boldsymbol{\epsilon}_\tau^q}[h(\boldsymbol{X}_\tau^Q)\left((\lambda h(\boldsymbol{X}_\tau) + r(\tau, \boldsymbol{X}_\tau))^Q - \lambda h(\boldsymbol{X}_\tau)\right) | h^Q(\boldsymbol{X}_\tau^Q) = kQ_p^{-1}] \\
&= \mathbb{E}_{\boldsymbol{X}, r, \boldsymbol{\epsilon}_\tau^q}[h(\boldsymbol{X}_\tau^Q)\left((\lambda h(\boldsymbol{X}_\tau) + r(\tau, \boldsymbol{X}_\tau) + \boldsymbol{\epsilon}_\tau^q) - \lambda h(\boldsymbol{X}_\tau)\right) | h^Q(\boldsymbol{X}_\tau^Q) = kQ_p^{-1}] \\
&= \mathbb{E}_{\boldsymbol{X}, r, \boldsymbol{\epsilon}_\tau^q}[\lambda(h^2(\boldsymbol{X}_\tau^Q) - h^2(\boldsymbol{X}_\tau^Q)) + r(\tau, \boldsymbol{X}_\tau) + \boldsymbol{\epsilon}_\tau^q) | h^Q(\boldsymbol{X}_\tau^Q) = kQ_p^{-1}] \\
&= \mathbb{E}_{r, \boldsymbol{\epsilon}_\tau^q}[r(\tau, \boldsymbol{X}_\tau) + \boldsymbol{\epsilon}_\tau^q)] \\
&= \mathbb{E}_r[r(\tau, \boldsymbol{X}_\tau)] + \mathbb{E}_{\boldsymbol{\epsilon}_\tau^q}[\boldsymbol{\epsilon}_\tau^q] = 0 + 0 = 0
\end{aligned}
\tag{49}
$$

$\square$

We can model the early paralysis of learning such that $h^Q(\boldsymbol{X}_\tau) = k \cdot Q_p|_{k=0} = 0$ for a quantization parameter $Q_p > 0$ for a $\tau > 0$. Theorem 3.1 describes that even though the quantized directional derivative $h^Q(\boldsymbol{X}_\tau) = 0$, the proposed compensation function yields an independent noise to $h^Q$, and the noise can work the learning process avoiding early paralysis as follows:

$$
\begin{aligned}
\boldsymbol{X}_{\tau+1}^Q &= \boldsymbol{X}_\tau^Q + \frac{1}{Q_p}\left\lfloor Q_p \cdot (\lambda h(\boldsymbol{X}_\tau^Q) + r(\tau, \boldsymbol{X}_\tau^Q)) + 0.5 \right\rfloor \\
&= \boldsymbol{X}_\tau^Q + \frac{1}{Q_p}\left\lfloor Q_p \cdot (\lambda h(\boldsymbol{X}_\tau^Q) + 0.5) + Q_p r(\tau, h(\boldsymbol{X}_\tau^Q)) \right\rfloor \\
&= \boldsymbol{X}_\tau^Q + \frac{1}{Q_p}\left\lfloor Q_p \cdot (\lambda h(\boldsymbol{X}_\tau^Q) + 0.5) + Q_p \lambda sgn(h(\boldsymbol{X}_\tau^Q)) \right\rfloor \Big|_{\lambda = k \cdot Q_p, \, k \in \mathbb{Z}^+} \\
&= \boldsymbol{X}_\tau^Q + \frac{1}{Q_p}\left\lfloor Q_p \cdot (\lambda h(\boldsymbol{X}_\tau^Q) + 0.5) \right\rfloor + \frac{1}{Q_p} \cdot Q_p \lambda \, sgn(h(\boldsymbol{X}_\tau^Q)) \\
&= \boldsymbol{X}_\tau^Q + 0 + \lambda \, sgn(h(\boldsymbol{X}_\tau^Q)).
\end{aligned}
\tag{50}
$$

In equation 50, we assume that $\lambda = k \cdot Q_p$, $k \in \mathbb{Z}^+$ for convenience.

While the proposed compensation function allows for avoiding early paralysis, we can't verify the convergence of the learning equation based on the proposed method due to the variance of the process $Y_t$ represented in equation 44. In the proposed compensation function, we designed the critical time index $\tau_0$ where the function denotes 0.5 and drops to zero after the critical time. While the proposed time-dependent compensation approach facilitates the convergence of the learning algorithm to a stable point beyond the critical time, our design does not provide an absolute guarantee of complete convergence for the learning algorithm.

Consequently, we will design an improved compensation to secure convergence by adding a time-dependent variance. One possible intuitive design we suggest is as follows:

$$
r(\tau, h(\boldsymbol{X}_\tau^Q)) \triangleq \lambda \sqrt{\frac{1}{\log\log(\kappa\tau + c_0)}} \frac{h(\boldsymbol{X}_\tau)}{\|h(\boldsymbol{X}_\tau)\|}.
\tag{51}
$$

## C  Fundamental Learning Equation based on Quantization

For convenience of discussion, we analyze the Quantized Stochastic Gradient Langevin Dynamics(QSGLD) instead of the Quantized Stochastic Langevin Dynamics(QSLD). We provide the learning equation of QSGLD is as follows :

$$
\boldsymbol{X}_{\tau+1}^Q = \boldsymbol{X}_\tau^Q - \lambda \nabla_{\boldsymbol{x}} \tilde{f}_\tau(\boldsymbol{X}_\tau) + Q_p^{-1}(\tau)\varepsilon_\tau^q.
\tag{52}
$$

---

**Algorithm 2** Virtual algorithm for transformation from the quantization error to a Gaussian random variable

---

1: $Q_p \leftarrow Q_p(-1, \tau)$        ▷ Set a Pure Quantization Parameter
2: Compute $h(\boldsymbol{X}_\tau)$ at $\tau$        ▷ Compute a Search Direction Vector
3: $h_\tau^Q \leftarrow \frac{1}{Q_p} \lfloor Q_p \cdot (-\lambda h(\boldsymbol{X}_\tau) + 0.5 Q_p^{-1}) \rfloor$        ▷ Quantization of the Search Direction Vector
4: $\boldsymbol{\varepsilon}_\tau \leftarrow Q_p(h_\tau^Q - h(\boldsymbol{X}_\tau))$        ▷ General Updating Rule for Learning
5: Compute $\boldsymbol{z}_\tau \leftarrow Q_p^{-1}(\boldsymbol{\varepsilon}_\tau, \tau) \boldsymbol{\varepsilon}_\tau$        ▷ Compute Gaussian random variable

---

For the analysis of the proposed learning equation, we establish the two systems of time indices. One is a single-sided time system based on a mini-batch index $\tau$, the other is a double-sided time system based on $\tau$ and an epoch $t_e$ as the unit summation of mini-batches.

In the single-sided time system, we define the time index with a time difference $\delta_\tau$ intuitively such that

$$\tau + 1 \triangleq \tau + \delta_\tau, \quad \delta_\tau \triangleq \frac{1}{B}, \tag{53}$$

where $B$ is the number of mini-batches for a unit epoch. We assume that the learning rate $\lambda$ is the reciprocal of the number of mini-batches $B$, so we can rewrite the time index as follows:

$$\tau + 1 \cdot \lambda \triangleq \tau + \lambda, \quad \lambda \triangleq \frac{1}{B} = \delta_\tau. \tag{54}$$

Consequently, we treat the increased time index $\tau + 1$ as equivalent to $\tau + \lambda$, so we abbreviate the learning rate behind the time index number when we use the number-based single-sided time system such that $\tau + 1 \cdot (\lambda)$.

In the double-sided time system, we define the epoch-based time index $t_e$ with the mini-batch-based time index $\tau$ such that

$$t_e + 1 = t_e + \sum_{\tau_b = 0}^{B-1} \delta_{\tau_b} = t_e + \lambda B, \quad \delta_{\tau_b} = \frac{1}{B}, \ \forall \tau_b \in \mathbb{Z}[0, B). \tag{55}$$

Therefore, we establish a connection with $\tau$ and $t_e$ such that

$$\tau = t_e \cdot B + k, \ k \in \mathbb{Z}[0, B), \quad t_e = \left\lfloor \frac{\tau}{B} \right\rfloor. \tag{56}$$

Using these time indexing systems, we derive a stochastic difference equation based on the uniform distributed quantization error according to WNH in the following section.

Before wrapping up the section, we discuss the sample-based time system. Each mini-batch contains $n_B$ samples equally, so we recognize there are $N_T = B \cdot n_B$ samples as the total numbers of data. Accordingly, we can establish the time index with such a unit sample, and the unit sample-based time system is natural in signal processing. However, the artificial intelligence framework such as the PyTorch works on a unit of mini-batch, not a unit sample. When we define the size of a mini-batch to be one, in other words, each mini-batch contains one sample, we can work the learning process to a unit sample. Even though such a process for a unit sample is possible, the process for a unit sample is not practically valuable for heavy computation time.

### C.1 TRANSFORMATION TO GAUSSIAN WIENER PROCESS

Firstly, we derive an intuitive stochastic difference equation of the quantization learning based on the single-side time system. Since the single-sided time system is based on a unit mini-batch index, we should directly transform the quantization error into a standard Wiener process. For the transformation, we exploit the quantization parameter with two parameters $Q_p(\varepsilon^q, t)$ as the transform function from the uniformly distributed random variable to a Gaussian random variable. The random number generation of a Gaussian random variable based on a uniformly distributed random variable is a standard and widely used method.

Box-Muller algorithm (by Box and Muller (1958)), the Ziggurat algorithm (by Marsaglia (1963) and Marsaglia and Tsang (2000)) and the inverse transform sampling (by Thomas et al. (2007)) are representative transforms.

Algorithm 2 describes the calculation of the quantization error $\epsilon_\tau^q$, the factor for the quantization $\varepsilon_\tau^q$, and the generation of a Gaussian random variable using the quantization parameter $Q_p(\varepsilon^q, t)$.

If we establish the $\bar{\eta}(\varepsilon_\tau^q) \triangleq z_\tau \in \mathbb{R}^d$ to be a Gaussian random generator as the factorizing function of the quantization parameter, we can obtain the Gaussian random variable $z_\tau \overset{i.i.d.}{\sim} \mathcal{N}(z; 0, I_d)$ as follows:

$$Q_p^{-1}(\varepsilon^q, \tau)\varepsilon_\tau^q = b^{-\bar{p}(\tau)}\eta^{-1}(\varepsilon_\tau^q) \cdot \varepsilon_\tau^q = \frac{\sqrt{\lambda}b^{-\bar{p}(\tau)}}{\|\varepsilon_\tau^q\|^2}\bar{\eta}(\varepsilon_\tau^q)\varepsilon_\tau^q \cdot \varepsilon_\tau^q = \sqrt{\lambda}b^{-\bar{p}(\tau)}z_\tau, \ b^{-\bar{p}(\tau)} \in \mathbb{Q}, \ (57)$$

where $\eta^{-1}(\varepsilon_\tau^q)$ is a mapping such that $\eta^{-1} : \mathbb{R}^d \mapsto \mathbb{R}^d$.

Additionally, if the input of $\eta^{-1}$ is $-1$, we define the quantization parameter as not a mapping but a scalar function such that $Q_p^{-1}(\tau) = \eta_0^{-1}b^{-\bar{p}(t)} \in \mathbb{Q}$.

Using equation 56, we can rewrite equation 13 as QSGLD such that

$$X_{\tau+1}^Q = X_\tau^Q - \lambda\nabla_x\tilde{f}_\tau(X_\tau) + \sqrt{\lambda} \cdot b^{-\bar{p}(\tau)}z_\tau. \tag{58}$$

The stochastic difference equation 58 formulates the representative Euler-Maruyama approximation of the following SDE:

$$dX_\tau^Q = -\nabla_x\tilde{f}_\tau(X_\tau)d\tau + b^{-\bar{p}(\tau)}dB_\tau, \tag{59}$$

where $B_\tau \in \mathbb{R}^d$ is a standard Wiener process such that $B_\tau \sim \mathcal{N}(z; 0, I_d)$.

Although we can define the transformation to the Gaussian Wiener process to obtain the SDE on the mini-batch indexed time system, the proposed algorithm does not show an advantage in comparison to the conventional SGLD. While we utilize the quantization error as a seed to generate the Gaussian random variable in the proposed algorithm, the conventional SGLD deploys the Gaussian random generator for the learning system. Hence, there is not any practical difference in the implementation of the algorithm, and the complexity of the implementation only increases.

## C.2 Analysis based on Central Limit Theorem

The key idea is the simplest method to transform the uniformly distributed quantization error into a standard Wiener process. For this purpose, we establish the sum of the learning equation within a unit epoch regarding the mini-batch-based time index based on the double-sided time index system as follows:

$$\begin{aligned}
X_{t_e+B/B}^Q &= X_{t_e+(B-1)/B}^Q - \lambda\nabla_x\tilde{f}_{B-1}(X_{t_e+(B-1)/B}) + Q_p^{-1}(t_e)\varepsilon_{t_e+(B-1)/B}^q \\
&\cdots \\
X_{t_e+2/B}^Q &= X_{t_e+1/B}^Q - \lambda\nabla_x\tilde{f}_1(X_{t_e+1/B}) + Q_p^{-1}(t_e)\varepsilon_{t_e+1/B}^q \\
X_{t_e+1/B}^Q &= X_{t_e}^Q - \lambda\nabla_x\tilde{f}_0(X_{t_e}) + Q_p^{-1}(t_e)\varepsilon_{t_e}^q.
\end{aligned} \tag{60}$$

Adding up each term in equation 60 gives

$$X_{t_e+1}^Q = X_{t_e}^Q - \lambda\sum_{\tau_b=0}^{B-1}\nabla_x\tilde{f}_{\tau_b}(X_{t_e+\tau_b/B}) + Q_p^{-1}(t_e)\sum_{\tau_b=0}^{B-1}\varepsilon_{t_e+\tau_b/B}^q, \tag{61}$$

where we set the initial time index to be $t_e = \tau$ and maintain the quantization parameter to be an equal value such that $Q_p^{-1}(t_e + \tau_b) = Q_p^{-1}(t_e), \forall\tau_b \in \mathbb{Z}[0, B)$.

According to Assumption 2 and Assumption 3, the quantization error $\{\epsilon_{t_e+\tau_b/B}^q\}_{\tau=0}^{B-1} = \{Q_p^{-1}(t_e)\varepsilon_{t_e+\tau_b/B}^q\}_{\tau_b=0}^{B-1}$ are independent random variables of the equal uniform distributions $\mathcal{U}(\varepsilon_{t_e+\tau_b/B}^q; 0, c_0Q_p^{-2}(t_e)I_d)$. Therefore, we can apply the Lindeberg–Lévy central limit theorem(CLT) to equation 61, and we obtain a stochastic difference equation based on a standard Wiener process.

Let the average factor of quantization $S_B \in \mathbb{R}^d$ such that $S_B = 1/B \cdot \sum_{\tau_b=0}^{B-1} \varepsilon_{t_e+\tau_b/B}^q = \lambda \cdot \sum_{\tau_b=0}^{B-1} \varepsilon_{t_e+\tau_b/B}^q$. Based on the expectation and the variance of $\varepsilon_{t_e+\tau_b/B}^q$, we can obtain the expectation and variance straightforwardly as follows:

$$\mathbb{E}S_B = \frac{1}{B} \sum_{\tau_b=0}^{B-1} \mathbb{E}\varepsilon_{t_e+\tau_b/B}^q = 0,$$

$$cov\, S_B = \mathbb{E}S_B \otimes S_B = \frac{1}{B^2}\mathbb{E} \sum_{\tau_b=0}^{B-1} \left( \varepsilon_{t_e+\tau_b/B}^q \otimes \sum_{\tau_b=0}^{B-1} \varepsilon_{t_e+\tau_b/B}^q \right) = \frac{B\,c_0}{B^2} \cdot \boldsymbol{I}_d = \lambda\,c_0\,\boldsymbol{I}_d \tag{62}$$

Let the i-th component of $S_B$ denote $S_B^{(i)}$. According to the Lindeberg–Lévy CLT, we can obtain the random variable $S_B^{(i)}$ following a normal distribution such that

$$\frac{1}{\sqrt{\lambda\,c_0}}S_B^{(i)} = \frac{\sqrt{B}}{\sqrt{c_0}}S_B^{(i)} \sim \mathcal{N}(0,1) \tag{63}$$

Therefore, if we define the quantization parameter appropriately, we can establish a standard wiener process as the summation of the quantization error for a unit epoch.

Since the i-th component of $1/\sqrt{\lambda\,c_0}\,S_B^{(i)}$ follows a normal distribution, we establish the quantization parameter to extract a vector-valued Gaussian random variable from the summation of the quantization error as follows:

$$\frac{1}{\sqrt{\lambda\,c_0}}S_B^{(i)} = \frac{1}{\sqrt{\lambda\,c_0}\,B} \sum_{\tau_b=0}^{B-1} \varepsilon_{t_e+\tau_b/B}^{(i)} = \sqrt{\frac{\lambda}{c_0}} \sum_{\tau_b=0}^{B-1} \varepsilon_{t_e+\tau_b/B}^{(i)} \triangleq z_{t_e}^{(i)} \in \mathbb{R}, \quad z_{t_e}^{(i)} \sim \mathcal{N}(0,1)$$

$$\implies \sum_{\tau_b=0}^{B-1} \varepsilon_{t_e+\tau_b/B}^{(i)} = \sqrt{\frac{c_0}{\lambda}}z_{t_e}^{(i)} \tag{64}$$

$$\implies Q_p^{-1}(t_e) \sum_{\tau_b=0}^{B-1} \varepsilon_{t_e+\tau_b/B}^{(i)} = Q_p^{-1}(t_e)\sqrt{\frac{c_0}{\lambda}}z_{t_e}^{(i)} = \eta^{-1}b^{-\bar{p}(t_e)}\sqrt{\frac{c_0}{\lambda}}z_{t_e}^{(i)},$$

where $z^{(i)}$ denotes the i-th component of $z \in \mathbb{R}^d$, which denotes a vector-valued Gaussian random variable.

Consequently, if we let $\eta \triangleq \frac{1}{\lambda}\sqrt{c_0/C_q}$, where $C_q \in \mathbb{R}^+$ is a positive constant value. Applying the auxiliary parameter of the quantization parameter $\eta$ to equation 61, we can obtain the following stochastic difference equation:

$$\boldsymbol{X}_{t_e+1}^Q = \boldsymbol{X}_{t_e}^Q - \lambda \sum_{\tau_b=0}^{B-1} \nabla_{\boldsymbol{x}}\tilde{f}_{\tau_b}(\boldsymbol{X}_{t_e+\tau_b/B}) + b^{-\bar{p}(t_e)}\lambda\sqrt{\frac{C_q}{c_0}} \sum_{\tau_b=0}^{B-1} \varepsilon_{t_e+\tau_b/B}^q$$

$$= \boldsymbol{X}_{t_e}^Q - \lambda \sum_{\tau_b=0}^{B-1} \nabla_{\boldsymbol{x}}\tilde{f}_{\tau_b}(\boldsymbol{X}_{t_e+\tau_b/B}) + \sqrt{C_q}b^{-\bar{p}(t_e)}\sqrt{\lambda}z_{t_e}, \tag{65}$$

where $\{z_{t_e}\}_{t_e=0}^{\infty}$ is a vector valued standard Wiener process such that $z_{t_e} \sim \mathcal{N}(0,\boldsymbol{I}_d),\ \forall t_e \in \mathbb{Z}^+$.

The proposed CLT-based QSGLD reveals the practical advantage that it does not require a random number generator. The learning process itself generates a Gaussian random number based on the uniformly distributed quantization error. From the practical perspective, at least after 50 units of mini-batch iterations, we can regard the proposed learning algorithm as operating in cooperation with a standard Wiener process.

## D    CONVERGENCE PROPERTY OF QSGLD

We use the following lemma to prove the theorems.

**Lemma : Auxiliary 1.** *For all $x \in \mathbf{R}$,*

$$(1-x) \leq \exp(-x). \tag{66}$$

*Proof.* The above equation is explicit, so we employ the lemma without proof.     □

### D.1 WEAK CONVERGENCE WITHOUT CONVEX ASSUMPTION

We rewrite the QSGLD represented in equation 65 with an increment of a vector-valued standard Wiener process $\Delta \boldsymbol{B}_\tau \in \mathbb{R}^d$ as follows:

$$\boldsymbol{X}_{t_e+1}^Q = \boldsymbol{X}_{t_e}^Q - \lambda \sum_{\tau_b=0}^{B-1} \nabla_{\boldsymbol{x}} \tilde{f}_{\tau_b}(\boldsymbol{X}_{t_e+\tau_b/B}) + \sqrt{C_q} b^{-\bar{p}(t_e)} \sum_{\tau_b=0}^{B-1} \Delta \boldsymbol{B}_{t_e+\tau_b/B}, \tag{67}$$

where we substitute $\sqrt{\lambda} \boldsymbol{z}_{t_e}$ into $\sum_{\tau_b=0}^{B-1} \Delta \boldsymbol{B}_{t_e+\tau_b/B}$, since the variance of the increments is the time increment $\lambda$ such that $\mathbb{E}(\Delta B_{\tau_b})^2 = \lambda$, $\forall \Delta B_{\tau_b} \in \mathbb{R}$ whereas $\boldsymbol{z} \sim \mathcal{N}(0, 1)$.

In equation 67, we consider the learning rate $\lambda$ as an increment of time $t_e$ which is the numbers of a mini-batch. Thus, if the learning rate $\lambda$ monotonically decreases to 0 such that $\lambda \downarrow 0$, we can obtain the following stochastic integration intuitively.

$$\boldsymbol{X}_{t_e+1}^Q = \boldsymbol{X}_{t_e}^Q - \int_{t_e}^{t_e+1} \nabla_{\boldsymbol{x}} \tilde{f}_s(\boldsymbol{X}_s) ds + \sqrt{C_q} b^{-\bar{p}(t_e)} \int_{t_e}^{t_e+1} d\boldsymbol{B}_s \tag{68}$$

Differentiate both terms with respect to $t = t_e + 1$, we get

$$d\boldsymbol{X}_t^Q = -\nabla_{\boldsymbol{x}} \tilde{f}_t(\boldsymbol{X}_t) dt + \sqrt{C_q} b^{-\bar{p}(t)} d\boldsymbol{B}_t. \tag{69}$$

This intuitive deduction looks like a verification of the SDE approximation of the discrete stochastic difference equation from equation 65 for QSGLD.

However, such a deduction does not provide any evidence of tightness regarding the state $\boldsymbol{X}_t^Q$, so we should provide more rigorous evidence of the SDE approximation. For the rigorous proof of the SDE approximation, we investigated two approaches. The one is weak convergence of a stochastic difference equation to corresponding SDE. Unfortunately, the weak convergence criterion provided by Kushner (1974) requires the monotonically decreasing of the learning rate $\lambda \downarrow 0$ to time index increases, i.e., $t \uparrow \infty$ Some of the weak convergence criteria cannot be satisfied without such a monotonically decreasing learning rate, despite the limited drift and diffusion terms.

The other approach is a weak approximation of SDE. This approach is based on the numerical analysis of SDE approximation. The fundamental concept of the approach is that if the statistical quantities between a discrete stochastic difference equation and approximated continuous SDE are equivalent, we consider the approximation to be well-defined.

In this paper, we provide the SDE approximation regarding the proposed quantization-based learning algorithm as following lemma.

**Lemma 3.2** The approximated Langvin SDE for QSGLD represented in equation 15 is as follows:

$$d\boldsymbol{X}_t = -\nabla_{\boldsymbol{x}} f(\boldsymbol{X}_t) dt + \sqrt{C_q} \cdot \sigma(t) d\boldsymbol{B}_t, \quad \forall t > t_0 \in \mathbb{R}^+, \because \sigma(t) \triangleq b^{-\bar{p}(t)}, \tag{70}$$

The approximation $equation$ 70 satisfies the order-1 weak approximation described in Definition 3.

*Proof.* **Preparation** Let the transition probability density from $t = \tau$ to $t + 1 = \tau + \lambda$ such that $p(t, \boldsymbol{X}_t^Q, t+1, \boldsymbol{X}_{t+1}^Q)$. We define the following one-step changes of QSLD/QSLGD from equation 12 and equation 13 in the main manuscript such that

$$\Delta(\boldsymbol{X}_t) \triangleq \lambda\, h(\boldsymbol{X}_\tau^Q), \quad \tilde{\Delta}(\boldsymbol{X}_t) \triangleq \boldsymbol{X}_{t+1} - \boldsymbol{X}_\tau^Q, \quad \Delta, \tilde{\Delta} \in \mathbf{R}^d, \tag{71}$$

where $\boldsymbol{X}_{t+1}^Q \sim p(t, \boldsymbol{X}_t^Q, t+1, \boldsymbol{X}_{t+1}^Q)$, and $\boldsymbol{X}_t^Q = \boldsymbol{X}_\tau^Q$.

By proving the following conditions presented in the theorems of SDE approximation to the SGD equation as proposed by Li et al. (2019) and Malladi et al. (2022), we validate the applicability of the SDE approximation to the proposed equation.

**Lipschitz continuity and continuous differentiability of the drift and diffusion functions** Assumption 1 in the manuscript provides that the drift $\nabla f(\boldsymbol{X}_t)$ is Lipschitz continuity. As the diffusion function in QSLGD is the quantization parameter, it is Lipschitz continuity by Definition 2 in the manuscript.

**Bounded moments condition** The first order bounded moments condition is given by:

$$\left|\mathbb{E}(\Delta_i(\boldsymbol{X}_t) - \tilde{\Delta}_i(\boldsymbol{X}_t)\right| \le K(\boldsymbol{X}_t)\lambda^2 \tag{72}$$

From the definition of $\Delta$, $\tilde{\Delta}$, for the initial time $t = \tau/B$, we get

$$\Delta(\boldsymbol{X}_t) - \tilde{\Delta}(\boldsymbol{X}_t) = \lambda h(\boldsymbol{X}_t) - \lambda h(\boldsymbol{X}_t) - Q_p^{-1}(t)\varepsilon_t^q = -Q_p^{-1}(t)\varepsilon_t^q \Rightarrow \mathbb{E}Q_p^{-1}(t)\varepsilon_t^q = Q_p^{-1}(t)\mathbb{E}\varepsilon_t^q = 0. \tag{73}$$

The equation 73 represents that the proposed algorithm fulfills the first-order bounded moment.

For the bound of the second moment, we calculate the moment as follows:

$$\Delta_i(\boldsymbol{X}_t)\Delta_j(\boldsymbol{X}_t) = \lambda^2 h_i(\boldsymbol{X}_t)h_j(\boldsymbol{X}_t)$$
$$\tilde{\Delta}_i(\boldsymbol{X}_t)\tilde{\Delta}_j(\boldsymbol{X}_t) = (\lambda h_i(\boldsymbol{X}_t) + Q_p^{-1}(t)\varepsilon_{it}^q)(\lambda h_j(\boldsymbol{X}_t) + Q_p^{-1}(t)\varepsilon_{jt}^q)$$
$$= \lambda^2 h_i(\boldsymbol{X}_t)h_j(\boldsymbol{X}_t) + \lambda(h_j(\boldsymbol{X}_t)Q_p^{-1}(t)\varepsilon_{jt}^q + h_i(\boldsymbol{X}_t)Q_p^{-1}(t)\varepsilon_{it}^q) + Q_p^{-2}(t)\varepsilon_{it}^q\varepsilon_{jt}^q. \tag{74}$$

The equation 74 implies that

$$\Delta_i(\boldsymbol{X}_t)\Delta_j(\boldsymbol{X}_t) - \tilde{\Delta}_i(\boldsymbol{X}_t)\tilde{\Delta}_j(\boldsymbol{X}_t)$$
$$= -\lambda(h_j(\boldsymbol{X}_t)Q_p^{-1}(t)\varepsilon_{jt}^q + h_i(\boldsymbol{X}_t)Q_p^{-1}(t)\varepsilon_{it}^q) - Q_p^{-2}(t)\varepsilon_{it}^q\varepsilon_{jt}^q.$$
$$\Rightarrow \mathbb{E}(\Delta_i(\boldsymbol{X}_t)\Delta_j(\boldsymbol{X}_t) - \tilde{\Delta}_i(\boldsymbol{X}_t)\tilde{\Delta}_j(\boldsymbol{X}_t))$$
$$= \lambda(h_j(\boldsymbol{X}_t)Q_p^{-1}(t)\mathbb{E}\varepsilon_{jt}^q + h_i(\boldsymbol{X}_t)Q_p^{-1}(t)\mathbb{E}\varepsilon_{it}^q) - Q_p^{-2}(t)\mathbb{E}\varepsilon_{it}^q\varepsilon_{jt}^q = -\frac{1}{12}Q_p^{-2}(t)\delta(i-j), \tag{75}$$

where $i, j \in \mathbf{N}[1, d]$ denotes the index of the vector components.

Thus, we can choose $t > t_0$ such that

$$\left|\Delta_i(\boldsymbol{X}_t)\Delta_j(\boldsymbol{X}_t) - \tilde{\Delta}_i(\boldsymbol{X}_t)\tilde{\Delta}_j(\boldsymbol{X}_t)\right| = \frac{1}{12}Q_p^{-2}(t)\delta(i-j) \le K_1(\boldsymbol{X}_t)\lambda^2, \tag{76}$$

where $K_1$ denotes a secondary differentiable function such that $K_1 : \mathbf{R}^d \to \mathbf{R}, \ K_1 \in C^2$.

Similarly, we calculate the third moment as follows:

$$\Delta_i(\boldsymbol{X}_t)\Delta_j(\boldsymbol{X}_t)\Delta_k(\boldsymbol{X}_t) = \lambda^3 h_i(\boldsymbol{X}_t)h_j(\boldsymbol{X}_t)h_k(\boldsymbol{X}_t) \tag{77}$$

, and

$$\tilde{\Delta}_i(\boldsymbol{X}_t)\tilde{\Delta}_j(\boldsymbol{X}_t)\tilde{\Delta}_k(\boldsymbol{X}_t) = \lambda^3 h_i(\boldsymbol{X}_t)h_j(\boldsymbol{X}_t)h_k(\boldsymbol{X}_t)$$
$$+ \lambda^2 Q_p^{-1}(t)\left(h_i(\boldsymbol{X}_t)h_j(\boldsymbol{X}_t)\varepsilon_{kt}^q + h_j(\boldsymbol{X}_t)h_k(\boldsymbol{X}_t)\varepsilon_{it}^q + h_k(\boldsymbol{X}_t)h_i(\boldsymbol{X}_t)\varepsilon_{jt}^q\right) \tag{78}$$
$$+ \lambda Q_p^{-2}(t)\left(h_i(\boldsymbol{X}_t)\varepsilon_{jt}^q\varepsilon_{kt}^q + h_j(\boldsymbol{X}_t)\varepsilon_{kt}^q\varepsilon_{it}^q + h_k(\boldsymbol{X}_t)\varepsilon_{it}^q\varepsilon_{jt}^q\right) + Q_p^{-3}(t)\varepsilon_{it}^q\varepsilon_{jt}^q\varepsilon_{kt}^q.$$

Based on the white noise hypothesis(WNH) of the quantization error, we can determine the expectation value of equation 78 such that

$$\mathbb{E}\tilde{\Delta}_i(\boldsymbol{X}_t)\tilde{\Delta}_j(\boldsymbol{X}_t)\tilde{\Delta}_k(\boldsymbol{X}_t) = \lambda^3 h_i(\boldsymbol{X}_t)h_j(\boldsymbol{X}_t)h_k(\boldsymbol{X}_t)$$
$$+ \frac{1}{12}\lambda Q_p^{-2}(t)\left(h_i(\boldsymbol{X}_t)\delta(j-k) + h_j(\boldsymbol{X}_t)\delta(k-i) + h_k(\boldsymbol{X}_t)\delta(i-j)\right) \tag{79}$$
$$= \lambda^3 h_i(\boldsymbol{X}_t)h_j(\boldsymbol{X}_t)h_k(\boldsymbol{X}_t) + \frac{1}{4}\lambda Q_p^{-2}(t)h_i(\boldsymbol{X}_t)|_{i=j=k}.$$

By equation 79 and equation 77, we get

$$\mathbb{E}\tilde{\Delta}_i(\boldsymbol{X}_t)\tilde{\Delta}_j(\boldsymbol{X}_t)\tilde{\Delta}_k(\boldsymbol{X}_t) - \mathbb{E}\Delta_i(\boldsymbol{X}_t)\Delta_j(\boldsymbol{X}_t)\Delta_k(\boldsymbol{X}_t)$$
$$= \lambda^3 h_i(\boldsymbol{X}_t)h_j(\boldsymbol{X}_t)h_k(\boldsymbol{X}_t) + \frac{1}{4}\lambda Q_p^{-2}(t)h_i(\boldsymbol{X}_t)|_{i=j=k} - \lambda^3 h_i(\boldsymbol{X}_t)h_j(\boldsymbol{X}_t)h_k(\boldsymbol{X}_t) \tag{80}$$
$$= \frac{1}{4}\lambda Q_p^{-2}(t)h_i(\boldsymbol{X}_t)|_{i=j=k}$$

According to Assumption 1, there exists an optimal point $\boldsymbol{X}^* \in \mathcal{D} \subset B^o(\boldsymbol{X}^*, \rho)$ such that $\nabla f(\boldsymbol{X}^*) = 0$ allowing us to obtain

$$
\begin{aligned}
&|\mathbb{E}\tilde{\Delta}_i(\boldsymbol{X}_t)\tilde{\Delta}_j(\boldsymbol{X}_t)\tilde{\Delta}_k(\boldsymbol{X}_t) - \mathbb{E}\Delta_i(\boldsymbol{X}_t)\Delta_j(\boldsymbol{X}_t)\Delta_k(\boldsymbol{X}_t)| \\
&= \frac{1}{4}\lambda Q_p^{-2}(t)|h_i(\boldsymbol{X}_t)|_{i=j=k} = \frac{1}{4}\lambda Q_p^{-2}(t)|h_i(\boldsymbol{X}_t) + \nabla f_i(\boldsymbol{X}^*)|_{i=j=k}.
\end{aligned}
\tag{81}
$$

In QSLGD, since $h(\boldsymbol{X}_t)$ is equal to $-\nabla f(\boldsymbol{X}_t)$, we apply the $L_1$ Lipschitz condition to calculate the bound as follows:

$$
\begin{aligned}
&|\mathbb{E}\tilde{\Delta}_i(\boldsymbol{X}_t)\tilde{\Delta}_j(\boldsymbol{X}_t)\tilde{\Delta}_k(\boldsymbol{X}_t) - \mathbb{E}\Delta_i(\boldsymbol{X}_t)\Delta_j(\boldsymbol{X}_t)\Delta_k(\boldsymbol{X}_t)| \\
&= \frac{1}{4}\lambda Q_p^{-2}(t)|h_i(\boldsymbol{X}_t) + \nabla f_i(\boldsymbol{X}^*)|_{i=j=k} = \frac{1}{4}\lambda Q_p^{-2}(t)|\nabla f_i(\boldsymbol{X}^*) - \nabla f_i(\boldsymbol{X}_t)|_{i=j=k} \\
&\leq \frac{1}{4}\lambda Q_p^{-2}(t)L_1|\boldsymbol{X}^* - \boldsymbol{X}_t|_{i=j=k} \leq \frac{1}{4}\lambda Q_p^{-2}(t)L_1\rho
\end{aligned}
\tag{82}
$$

By selecting $t > t_0$ such that $Q_p(t)^{-2} < \lambda$, the bound condition of the third-order moment is satisfied, resulting in

$$
|\mathbb{E}\tilde{\Delta}_i(\boldsymbol{X}_t)\tilde{\Delta}_j(\boldsymbol{X}_t)\tilde{\Delta}_k(\boldsymbol{X}_t) - \mathbb{E}\Delta_i(\boldsymbol{X}_t)\Delta_j(\boldsymbol{X}_t)\Delta_k(\boldsymbol{X}_t)| \leq \frac{1}{4}\lambda Q_p^{-2}(t)L_1\rho < \frac{1}{4}\lambda^2 L_1 = K_1(\boldsymbol{X}_t)\lambda^2
\tag{83}
$$

**Adaptation Lemma by Li et al. (2019) and Malladi et al. (2022)** Under the verification of the bounded momentum condition equation 73, equation 75, equation 81, we can establish the following polynomial growth condition: There exists a subset $P$ of the index set $\triangleq \{1, 2, \cdots, d\}$ such that the following holds. Below we use the notations $\|\boldsymbol{x}_P\| \triangleq \sqrt{\sum_{i \in P} x_i^2}$ and $\|\boldsymbol{x}_R\| \triangleq \sqrt{\sum_{i \notin P} x_i^2}$.

1. There is a constant $C_1 > 0$, which is independent of $\lambda$, so that for all $\tau \leq t \cdot B$

$$
\begin{aligned}
\|\mathbb{E}\Delta(\boldsymbol{X}_\tau)\|_P &\leq C_1\lambda(1 + \|\boldsymbol{X}_\tau\|_P) \\
\|\mathbb{E}\Delta(\boldsymbol{X}_\tau)\|_R &\leq C_1\lambda(1 + \|\boldsymbol{X}_\tau\|_P^{w_1})(1 + \|\boldsymbol{X}_\tau\|_R)
\end{aligned}
\tag{84}
$$

2. For all $m \geq 1$, there are constant $C_{2m}, w_{2m} > 0$ (independent of $\lambda$) such that for all $\tau \leq t \cdot B$

$$
\begin{aligned}
\|\mathbb{E}\Delta(\boldsymbol{X}_\tau)\|_P^{2m} &\leq C_1\lambda(1 + \|\boldsymbol{X}_\tau\|_P^{2m}) \\
\|\mathbb{E}\Delta(\boldsymbol{X}_\tau)\|_R &\leq C_1\lambda(1 + \|\boldsymbol{X}_\tau\|_P^{w_{2m}})(1 + \|\boldsymbol{X}_\tau\|_R^{2m})
\end{aligned}
\tag{85}
$$

The analysis shows that the function $K_1$ is differentiable to the first order, allowing for the bound on the third moment. Consequently, a function $g$ twice differentiable ($g \in C^2$) is enough to achieve a first-order weak approximation for QSLGD.

**Adaptation Theorem 3 in by Li et al. (2019) Malladi et al. (2022)** Since QSLGD satisfies the Lipschitz continuity, the bounded moment conditions, and the adaptation lemma, QSLGD fulfills a first-order weak approximation defined as Definition 3 in the manuscript, by the theorem provided by Theorem 3 in by Li et al. (2022), as follows: For each function $g \in C^2$, there exists a constant $C_{o1} > 0$ (independent of $\lambda$) such that

$$
\max_{0 \leq t \leq T}|\mathbb{E}g(\boldsymbol{X}_{\tau=t \cdot B}^Q) - \mathbb{E}g(\boldsymbol{X}_t)| \leq C_{o1}\lambda
\tag{86}
$$

$\square$

**Theorem 3.3** Consider the transition probability density, denoted as $p(t, \boldsymbol{X}_t, t + \bar{\tau}, \boldsymbol{x}^*)$, from an arbitrary state $\boldsymbol{X}_t \in \mathbb{R}^d$ to the optimal point $\boldsymbol{x}^* \in \mathbb{R}^d$, $\boldsymbol{X}_t \neq \boldsymbol{x}^*$ after a time interval $\bar{\tau} \in \mathbb{R}^+$, for all $t > t_0$. If the quantization parameter is bounded as follows:

$$
\sup_{t \geq 0} Q_p(t) = \frac{1}{C} \cdot \log(t + 2), \quad C \in \mathbb{R}^{++},
\tag{87}
$$

for $\boldsymbol{X}_t \neq \bar{\boldsymbol{X}}_t$, QSGLD represented as equation 70 converges with distribution in the sense of Cauchy convergence such that

$$\overline{\lim_{\bar{\tau} \to \infty}} \sup_{\boldsymbol{X}_t, \bar{\boldsymbol{X}}_t \in \mathbb{R}^n} |p(t, \bar{\boldsymbol{X}}_t, t + \bar{\tau}, \boldsymbol{x}^*) - p(t, \boldsymbol{X}_t, t + \bar{\tau}, \boldsymbol{x}^*)| \leq \tilde{C} \cdot \exp\left( -\sum_{\bar{\tau}=0}^{\infty} \delta_{t+\bar{\tau}} \right), \qquad (88)$$

where $\delta_t$ denotes the infimum of the transition probability density from time $t$ to $t + 1$ given by $\delta_t = \inf_{x,y \in \mathbb{R}^d} p(t, x, t + 1, y)$, satisfying $\sum_{\bar{\tau}=0}^{\infty} \delta_{t+\bar{\tau}} = \infty$, and $\tilde{C}$ denotes a positive value.

*Proof.* We depend on the lemmas in works of Geman and Hwang (1986) to prove the theorem. Herein, we prove the following convergence of the transition probability density:

$$\overline{\lim_{\bar{\tau} \to \infty}} \sup_{\boldsymbol{X}_t, \bar{\boldsymbol{X}}_t \in \mathbb{R}^n} |p(t, \bar{\boldsymbol{X}}_t, t + \bar{\tau}, \boldsymbol{x}^*) - p(t, \boldsymbol{X}_t, t + \bar{\tau}, \boldsymbol{x}^*)| = 0, \qquad (89)$$

where $t$ and $\tau$ denote the current time index and the process time index, respectively. $\boldsymbol{x}^* \in \mathbb{R}^d$ denotes an global optimum for the objective function $f(\boldsymbol{X}_t)$ such that $f(\boldsymbol{x}^*) < f(\boldsymbol{X}_t)$, $\forall t \geq 0$.

Let the infimum of the transition probability density from $t$ to $t + 1$ such that

$$\delta_t = \inf_{\boldsymbol{x}, \boldsymbol{y} \in \mathbf{R}^d} p(t, \boldsymbol{x}, t + 1, \boldsymbol{y}) \qquad (90)$$

According to the lemma in Geman and Hwang (1986), we can evaluate the upper bound of equation 89 as follows: For state vectors $\boldsymbol{v}, \boldsymbol{w}, \boldsymbol{z}, \boldsymbol{f} \in \mathbf{R}^d$ and time indexes $s, t \in \mathbf{R}^+$,

$$\overline{\lim_{t \to \infty}} \sup_{\boldsymbol{v}, \boldsymbol{w} \in \mathbb{R}^d} |p(s, \boldsymbol{v}, t, \boldsymbol{f}) - p(s, \boldsymbol{w}, t, \boldsymbol{f})|$$

$$= \overline{\lim_{t \to \infty}} \sup_{\boldsymbol{v}, \boldsymbol{w} \in \mathbb{R}^d} \left| \int p(s, \boldsymbol{v}, s + 1, \boldsymbol{z}) p(s + 1, \boldsymbol{z}, t, \boldsymbol{f}) dz - \int p(s, \boldsymbol{w}, s + 1, \boldsymbol{z}) p(s + 1, \boldsymbol{z}, t, \boldsymbol{f}) dz \right|$$

$$= \overline{\lim_{t \to \infty}} \sup_{\boldsymbol{v}, \boldsymbol{w} \in \mathbb{R}^d} \left| \int p(s, \boldsymbol{v}, s + 1, \boldsymbol{z}) p(s + 1, \boldsymbol{z}, t, \boldsymbol{f}) dz - \int p(s, \boldsymbol{w}, s + 1, \boldsymbol{z}) p(s + 1, \boldsymbol{z}, t, \boldsymbol{f}) dz \right.$$
$$\left. -(\delta_s - \delta_s) p(s + 1, \boldsymbol{z}; t, \boldsymbol{f}) \right|$$

$$= \overline{\lim_{t \to \infty}} \sup_{\boldsymbol{v}, \boldsymbol{w} \in \mathbb{R}^d} \left| \int (p(s, \boldsymbol{v}; s + 1, \boldsymbol{z}) - \delta_s) p(s + 1, \boldsymbol{z}; t, \boldsymbol{f}) dz - \int (p(s, \boldsymbol{w}; s + 1, \boldsymbol{z}) - \delta_s) p(s + 1, \boldsymbol{z}; t, \boldsymbol{f}) dz \right|$$

$$\leq \overline{\lim_{t \to \infty}} \sup_{\boldsymbol{v}, \boldsymbol{w} \in \mathbb{R}^d} \left| \int (p(s, \boldsymbol{v}; s + 1, \boldsymbol{z}) - \delta_s) \sup_{\boldsymbol{z} \in \mathbb{R}^d} p(s + 1, \boldsymbol{z}; t, \boldsymbol{f}) dz - \int (p(s, \boldsymbol{w}; , s + 1, \boldsymbol{z}) - \delta_s) \inf_{\boldsymbol{z} \in \mathbb{R}^d} p(s + 1, \boldsymbol{z}; t, \boldsymbol{f}) dz \right|$$

$$= \overline{\lim_{t \to \infty}} \sup_{\boldsymbol{v}, \boldsymbol{w} \in \mathbb{R}^d} \left| \sup_{\boldsymbol{z} \in \mathbb{R}^d} p(s + 1, \boldsymbol{z}; t, \boldsymbol{f}) \int (p(s, \boldsymbol{v}; s + 1, \boldsymbol{z}) - \delta_s) dz - \inf_{\boldsymbol{z} \in \mathbb{R}^d} p(s + 1, \boldsymbol{z}; t, \boldsymbol{f}) \int (p(s, \boldsymbol{w}; s + 1, \boldsymbol{z}) - \delta_s) dz \right|$$

$$\leq \overline{\lim_{t \to \infty}} \sup_{\boldsymbol{v}, \boldsymbol{w} \in \mathbb{R}^d} \left| (1 - \delta_s) \sup_{\boldsymbol{z} \in \mathbb{R}^d} p(s + 1, \boldsymbol{z}; t, \boldsymbol{f}) - (1 - \delta_s) \inf_{\boldsymbol{z} \in \mathbb{R}^d} p(s + 1, \boldsymbol{z}; t, \boldsymbol{f}) \right|$$

$$= \overline{\lim_{t \to \infty}} \sup_{\boldsymbol{v}, \boldsymbol{w} \in \mathbb{R}^d} (1 - \delta_s) \left| \sup_{\boldsymbol{z} \in \mathbb{R}^d} p(s + 1, \boldsymbol{z}; t, \boldsymbol{f}) - \inf_{\boldsymbol{z} \in \mathbb{R}^d} p(s + 1, \boldsymbol{z}; t, \boldsymbol{f}) \right|$$

$$\cdots$$

$$\leq \overline{\lim_{t \to \infty}} \left( \prod_{k=0}^{(t-s)-1} (1 - \delta_{s+k}) \right) \cdot \sup_{\boldsymbol{v}, \boldsymbol{w} \in \mathbb{R}^d} |p(s + (t - s), \boldsymbol{v}; t, \boldsymbol{f}) - p(s + (t - s), \boldsymbol{w}; t, \boldsymbol{f})|$$

$$\leq \overline{\lim_{t \to \infty}} \left( \prod_{k=0}^{(t-s)-1} (1 - \delta_{s+k}) \right) = \prod_{k=0}^{\infty} (1 - \delta_{s+k}).$$

$$(91)$$

Thus, we obtain

$$\overline{\lim_{\tau \to \infty}} \sup_{\boldsymbol{X}_t, \bar{\boldsymbol{X}}_t \in \mathbb{R}^n} |p(t, \bar{\boldsymbol{X}}_t, t + \bar{\tau}, \boldsymbol{x}^*) - p(t, \boldsymbol{X}_t, t + \bar{\tau}, \boldsymbol{x}^*)| \leq \prod_{k=0}^{\infty} (1 - \delta_{t+k}). \qquad (92)$$

From the exponential approximation equation 1 in **Lemma:Auxiliary**, we rewrite equation 92 as follows:

$$\overline{\lim_{\tau \to \infty}} \sup_{\boldsymbol{X}_t, \bar{\boldsymbol{X}}_t \in \mathbf{R}^n} |p(t, \bar{\boldsymbol{X}}_t, t + \bar{\tau}, \boldsymbol{x}^*) - p(t, \boldsymbol{X}_t, t + \bar{\tau}, \boldsymbol{x}^*)| \leq \exp(-\sum_{k=0}^{\infty} \delta_{t+k})). \tag{93}$$

To verify the existence of an upper bound of the right-hand side in equation 93, we rephrase the approximation SDE governing the dynamics of the proposed algorithm, as stated in Lemma 3.2:

$$d\boldsymbol{X}_s = -\nabla f(\boldsymbol{X}_s)ds + \sigma(s)\sqrt{C_q}d\boldsymbol{B}_s, \quad s \in \mathbf{R}[t, t+1). \tag{94}$$

Let $P_x$ be the probability measures on $\mathcal{F}$ induced by equation 94 and the probability distribution $Q_x$ given by the following equation:

$$d\bar{\boldsymbol{X}}_s = \sigma(s)\sqrt{C_q}d\boldsymbol{B}_s, \quad s \in \mathbf{R}[t, t+1). \tag{95}$$

According to the Girsanov theorem (Øksendal (2003); Klebaner (2012)), we obtain

$$\frac{dP_{\boldsymbol{X}}}{dQ_{\bar{\boldsymbol{X}}}} = \exp\left\{ -\int_t^{t+1} \frac{C_q^{-1}}{\sigma^2(s)} \nabla f(\boldsymbol{X}_s)d\bar{\boldsymbol{X}}_s - \frac{1}{2}\int_t^{t+1} \frac{C_q^{-1}}{\sigma^2(s)} \|\nabla f(\boldsymbol{X}_s)\|^2 ds \right\}. \tag{96}$$

To compute the upper bound of equation 96, we will check the upper bound of $\|\nabla f\|$. Considering Assumption 2, the gradient of $f(\boldsymbol{X}_t) \in C^2$ fulfills the Lipschitz continuous condition as well. Thereby, there exists a positive value $L_1 \in \mathbf{R}^+$ such that

$$\|\nabla f(\boldsymbol{X}_s) - \nabla f(\boldsymbol{x}^*)\| \leq L_1 \|\boldsymbol{X}_s - \boldsymbol{x}^*\|, \quad \forall s > 0. \tag{97}$$

Successively, since $\nabla f(\boldsymbol{x}^*) = 0$, the Lipschitz condition forms simply as follows :

$$\|\nabla f(\boldsymbol{X}_t)\| \leq L_1 \rho = C_0, \tag{98}$$

where $\rho = \|\boldsymbol{X}_t - \boldsymbol{x}^*\|$ from Assumption 1.

Additionally, holding the $L_1$ Lipschitz continuity and the assumption of which $f \in C^2$, it implies that there exists a positive value $L_2 \in \mathbf{R}^+$ such that

$$\|\boldsymbol{H}_{\boldsymbol{x}}(f)(\boldsymbol{x})\|_{\mathbb{R}^{d \times d}} < L_2 \text{ and } |\Delta_{\boldsymbol{x}}^2 f(\boldsymbol{x})| < L_2 d, \quad \forall \boldsymbol{x} \in B^o(\boldsymbol{x}, \rho), \tag{99}$$

where $\boldsymbol{H}_{\boldsymbol{x}}(f) \in \mathbb{R}^{d \times d}$ denotes the Hessian of $f : \mathbb{R}^d \mapsto \mathbb{R}$, and $\Delta_{\boldsymbol{x}}^2 f \in \mathbb{R}$ denotes the Laplacian of $f$.

To evaluate the first term of the right-hand side in equation 96, we derive the stochastic differential of $f(\boldsymbol{X}_s)$ with respect to $d\boldsymbol{X}_t$ described in equation 70 as follows:

$$df(\boldsymbol{X}_s) = \nabla f(\boldsymbol{X}_s) \cdot d\bar{\boldsymbol{X}}_s + \frac{1}{2}C_q\sigma^2(s)\Delta f(\boldsymbol{X}_s)ds$$

$$\implies \frac{C_q^{-1}}{\sigma^2(s)}\nabla f(\boldsymbol{X}_s) \cdot d\bar{\boldsymbol{X}}_s = \frac{C_q^{-1}}{\sigma^2(s)}df(\boldsymbol{X}_s) - \frac{1}{2}\Delta f(\boldsymbol{X}_s)ds \tag{100}$$

$$\implies \int_t^{t+1} \frac{C_q^{-1}}{\sigma^2(s)}\nabla f(\boldsymbol{X}_s) \cdot d\bar{\boldsymbol{X}}_s = \int_t^{t+1} \frac{C_q^{-1}}{\sigma^2(s)}df(\boldsymbol{X}_s) - \frac{1}{2}\int_t^{t+1}\Delta f(\boldsymbol{X}_s)ds.$$

To get a feasible result for the stochastic integration, We integrate the first term of the right-hand side in equation 100 partially such that

$$\int_t^{t+1} \frac{C_q^{-1}}{\sigma^2(s)}df(\boldsymbol{X}_s) = C_q^{-1}\left( \frac{f(\boldsymbol{X}_s)}{\sigma^2(s)}\Big|_t^{t+1} - \int_t^{t+1} f(\boldsymbol{X}_s)d\left(\frac{1}{\sigma^2(s)}\right) \right). \tag{101}$$

Since equation 101 relies on the random variable $\boldsymbol{X}_t$, we should the upper bound of equation 101 for the positive result such that $\int_t^{t+1} \frac{C_q^{-1}}{\sigma^2(s)}df(\boldsymbol{X}_s) \geq 0$, and the negative result $\int_t^{t+1} \frac{C_q^{-1}}{\sigma^2(s)}df(\boldsymbol{X}_s) < 0$.

For $\int_t^{t+1} \frac{C_q^{-1}}{\sigma^2(s)} df(\boldsymbol{X}_s) \geq 0$, we can obtain

$$
\begin{aligned}
\left| \int_t^{t+1} \frac{C_q^{-1}}{\sigma^2(s)} df(\boldsymbol{X}_s) \right| &\leq C_q^{-1} \left| \frac{f(\boldsymbol{X}_{t+1})}{\sigma^2(t+1)} - \frac{f(\boldsymbol{X}_t)}{\sigma^2(t)} - \int_t^{t+1} f(\boldsymbol{X}_s) d\left( \frac{1}{\sigma^2(s)} \right) \right| \\
&\leq C_q^{-1} \left| \frac{\sup_{x\in\mathbf{R}^d} f}{\sigma^2(t+1)} - \frac{\inf_{x\in\mathbf{R}^d} f}{\sigma^2(t)} - \int_t^{t+1} (\inf_{x\in\mathbf{R}^d} f) d\left( \frac{1}{\sigma^2(s)} \right) \right| \\
&= C_q^{-1} \left| \frac{\sup_{x\in\mathbf{R}^d} f}{\sigma^2(t+1)} - \inf_{x\in\mathbf{R}^d} f \left( \frac{1}{\sigma^2(t)} + \int_t^{t+1} d\left( \frac{1}{\sigma^2(s)} \right) \right) \right| \qquad (102) \\
&= C_q^{-1} \left| \frac{\sup_{x\in\mathbf{R}^d} f}{\sigma^2(t+1)} - \inf_{x\in\mathbf{R}^d} f \left( \frac{1}{\sigma^2(t)} + \frac{1}{\sigma^2(t+1)} - \frac{1}{\sigma^2(t)} \right) \right| \\
&= \frac{C_q^{-1}}{\sigma^2(t+1)} \left| \sup_{x\in\mathbf{R}^d} f - \inf_{x\in\mathbf{R}^d} f \right| \leq \frac{C_q^{-1} L_0 \rho}{\sigma^2(t+1)}.
\end{aligned}
$$

For $\int_t^{t+1} \frac{C_q^{-1}}{\sigma^2(s)} df(\boldsymbol{X}_s) < 0$, in the same manner, we get

$$
\begin{aligned}
\left| \int_t^{t+1} \frac{C_q^{-1}}{\sigma^2(s)} df(\boldsymbol{X}_s) \right| &\leq C_q^{-1} \left| \frac{f(\boldsymbol{X}_{t+1})}{\sigma^2(t+1)} - \frac{f(\boldsymbol{X}_t)}{\sigma^2(t)} - \int_t^{t+1} f(\boldsymbol{X}_s) d\left( \frac{1}{\sigma^2(s)} \right) \right| \\
&\leq C_q^{-1} \left| \frac{\inf_{x\in\mathbf{R}^d} f}{\sigma^2(t+1)} - \frac{\sup_{x\in\mathbf{R}^d} f}{\sigma^2(t)} - \int_t^{t+1} (\sup_{x\in\mathbf{R}^d} f) d\left( \frac{1}{\sigma^2(s)} \right) \right| \\
&= C_q^{-1} \left| \frac{\inf_{x\in\mathbf{R}^d} f}{\sigma^2(t+1)} - \sup_{x\in\mathbf{R}^d} f \left( \frac{1}{\sigma^2(t)} + \int_t^{t+1} d\left( \frac{1}{\sigma^2(s)} \right) \right) \right| \qquad (103) \\
&= C_q^{-1} \left| \frac{\inf_{x\in\mathbf{R}^d} f}{\sigma^2(t+1)} - \sup_{x\in\mathbf{R}^d} f \left( \frac{1}{\sigma^2(t)} + \frac{1}{\sigma^2(t+1)} - \frac{1}{\sigma^2(t)} \right) \right| \\
&= \frac{C_q^{-1}}{\sigma^2(t+1)} \left| \inf_{x\in\mathbf{R}^d} f - \sup_{x\in\mathbf{R}^d} f \right| \leq \frac{C_q^{-1} L_0 \rho}{\sigma^2(t+1)}.
\end{aligned}
$$

Hence, we get the upper bound of the first term in equation 100 as follows:

$$
\left| \int_t^{t+1} \frac{C_q^{-1}}{\sigma^2(s)} df(\boldsymbol{X}_s) \right| \leq \frac{C_q^{-1} L_0 \rho}{\sigma^2(t+1)} \qquad (104)
$$

The second term in equation 100 is the integration to deterministic time index $s$, so we can obtain the upper-bound conveniently holding equation 99 such that

$$
\left| \frac{1}{2} \int_t^{t+1} \Delta_{\boldsymbol{x}} f(\boldsymbol{X}_s) ds \right| \leq \frac{1}{2} \sup_{\forall x \in \mathcal{D}} |\Delta_{\boldsymbol{x}} f(x)| = \frac{1}{2} L_2 d. \qquad (105)
$$

The result of equation 104 and equation 105 implies the first term of the right-hand side in equation 96 as follows:

$$
\begin{aligned}
\left| \int_t^{t+1} \frac{C_q^{-1}}{\sigma^2(s)} \nabla f(\boldsymbol{X}_s) \cdot d\bar{\boldsymbol{X}}_s \right| &\leq \left| \int_t^{t+1} \frac{C_q^{-1}}{\sigma^2(s)} df(\boldsymbol{X}_s) \right| + \left| \frac{1}{2} \int_t^{t+1} \Delta f(\boldsymbol{X}_s) ds \right| \\
&\leq \frac{C_q^{-1} L_0 \rho}{\sigma^2(t+1)} + \frac{1}{2} L_2 d < \frac{C_q^{-1} L_0 \rho + 0.5 L_2 d \cdot \sigma^2(0)}{\sigma^2(t+1)}.
\end{aligned} \qquad (106)
$$

Since $\sigma(t)$ is a monotonic decreasing function, there exists a positive value $\bar{s} > 0$ such that $\sigma(t) \leq \bar{s}^{-1} \sigma(t+1)$. It implies that

$$
\left| - \int_t^{t+1} \frac{C_q^{-1}}{\sigma^2(s)} \nabla f(\boldsymbol{X}_s) \cdot d\bar{\boldsymbol{X}}_s \right| \leq \frac{C_q^{-1} L_0 \rho + 0.5 L_D d \sigma^2(0)}{\bar{s}} \frac{1}{\sigma^2(t)} = \frac{C_1}{\sigma^2(t)} \qquad (107)
$$

where $C_1$ denotes a positive value such that $C_1 > \frac{C_q^{-1} L_0 \rho + 0.5 L_D d \sigma^2(0)}{\bar{s}}$.

Furthermore, We can straightforwardly obtain the upper bound of the second term in the right-hand side of equation 96 as follows:

$$\frac{1}{2}\left| \int_t^{t+1} \frac{C_q^{-1}}{\sigma^2(s)} \|\nabla f(\boldsymbol{X}_s)\|^2 ds \right| \leq \frac{1}{2} \frac{C_q^{-1}}{\sigma^2(t+1)} \sup\|\nabla_x f(\boldsymbol{X}_s)\|^2 \int_t^{t+1} ds$$

$$\leq \frac{1}{2\sigma^2(t+1)} C_q^{-1} \cdot C_0^2 \leq \frac{C_2}{\sigma^2(t)}, \quad \because C_2 > \frac{C_0^2}{2C_q \bar{s}}. \tag{108}$$

Since $\sigma(s) \triangleq b^{-\bar{p}(t)}$ is monotone decreasing function, the supremum of $\sigma(s)$ is $\sigma(0)$ for all $s \in \mathbf{R}[0, \infty)$, i.e. $\sup_{s \in \mathbf{R}[0,\infty]} \sigma(s) = \sigma(0) \triangleq \sigma$. With the supremum of each term in equation 96, we can obtain the lower bound of the Radon-Nykodym derivative equation 96 such that

$$\frac{dP_{\boldsymbol{X}}}{dQ_{\bar{\boldsymbol{X}}}} \geq \exp\left(-\frac{C_1 + C_2}{\sigma^2(t)}\right) \geq \exp\left(-\frac{C_3}{\sigma^2(t)}\right), \quad \because C_3 > C_2 + C_1. \tag{109}$$

Accordingly, for any $\varepsilon > 0$ and $\boldsymbol{X}_t, \ \boldsymbol{x}^* \in \mathbf{R}^d$, the infimum of $P_x(|X_{t+1} - \boldsymbol{x}^*| < \varepsilon)$ is

$$P_{\boldsymbol{X}}(\|\boldsymbol{X}_t - \boldsymbol{x}^*\| < \varepsilon) \geq \exp\left(-\frac{C_3}{\sigma^2(t)}\right) Q_{\bar{\boldsymbol{X}}}(\|\boldsymbol{X}_t - \boldsymbol{x}^*\| < \varepsilon). \tag{110}$$

As $Q_w$ is a normal distribution based on equation 95, we have

$$P_{\boldsymbol{X}}(|\boldsymbol{X}_{t+1} - \boldsymbol{x}^*| < \varepsilon) \geq \exp\left(-\frac{C_3}{\sigma^2(t)}\right) \int_{\|\boldsymbol{X}-\boldsymbol{x}^*\|<\varepsilon} \frac{1}{\sigma(t)\sqrt{2\pi \int_t^{t+1} C_q d\tau}} \exp\left(-\frac{(\boldsymbol{X} - \boldsymbol{x}^*)^2}{2 \int_t^{t+1} C_q \sigma^2(\tau) d\tau}\right) dx$$

$$\geq \exp\left(-\frac{C_3}{\sigma^2(t)}\right) \int_{\|\boldsymbol{X}-\boldsymbol{x}^*\|<\varepsilon} \frac{1}{\sigma(t)\sqrt{2\pi C_q \int_t^{t+1} d\tau}} \exp\left(-\frac{(\sqrt{\rho} + \varepsilon)^2}{2\sigma^2(0) C_q \int_t^{t+1} d\tau}\right) dx$$

$$\geq \exp\left(-\frac{C_3}{\sigma^2(t)}\right) \frac{1}{\sigma(0)\sqrt{2\pi C_q}} \exp\left(-\frac{(\sqrt{\rho} + \varepsilon)^2}{2\sigma(0) C_q}\right) \int_{\|\boldsymbol{X}-\boldsymbol{x}^*\|<\varepsilon} dx$$

$$= \exp\left(-\frac{C_3}{\sigma^2(t)}\right) \frac{1}{\sqrt{2\pi C_q}} \exp\left(-\frac{(\sqrt{\rho} + \varepsilon)^2}{2C_q}\right) \frac{2\pi^{d/2} \varepsilon^d}{\Gamma(d/2 + 1)} \quad \because \sigma(0) = 1$$

$$\geq \exp\left(-\frac{C_3}{\sigma^2(t)}\right) \frac{1}{\sqrt{2\pi C_q}} \left(1 + \frac{(\sqrt{\rho} + \varepsilon)^2}{2C_q}\right) \frac{2\pi^{d/2} \varepsilon^d}{\Gamma(d/2 + 1)}$$

$$\geq \exp\left(-\frac{C_3}{\sigma^2(t)}\right) \frac{1}{\sqrt{2\pi C_q}} \left(\frac{2C_q + (\sqrt{\rho} + \varepsilon)^2}{C_q}\right)\Bigg|_{\rho=0, \varepsilon=0} \cdot \frac{\pi^{d/2} \varepsilon^d}{\Gamma(d/2 + 1)}$$

$$\geq \exp\left(-\frac{C_3}{\sigma^2(t)}\right) \cdot C_4 \cdot \varepsilon, \quad \because C_4 = \frac{\sqrt{2}}{\sqrt{\pi C_q}} \cdot \frac{\pi^{d/2} \varepsilon^{d-1}}{\Gamma(d/2 + 1)}. \tag{111}$$

Finally, we obtain the lower bound of the transition probability density such that

$$\delta_t = \inf_{\boldsymbol{x}, \boldsymbol{y} \in \mathbf{R}^d} p(t, \boldsymbol{x}, t+1, \boldsymbol{y})\Bigg|_{\boldsymbol{x}=\boldsymbol{X}_t, \ y=\boldsymbol{x}^*} = \inf_{\boldsymbol{x}, \boldsymbol{y} \in \mathbf{R}^d} \lim_{\varepsilon \to 0} \frac{1}{\varepsilon} P_{\boldsymbol{X}}(\|\boldsymbol{X}_{t+1} - \boldsymbol{x}^*| < \varepsilon)$$

$$\geq \inf_{\boldsymbol{x}, \boldsymbol{y} \in \mathbf{R}^d} \lim_{\varepsilon \to 0} \frac{1}{\varepsilon} \cdot C_4 \cdot \exp\left(-\frac{C_3}{\sigma^2(t)}\right) \cdot \varepsilon \geq \exp\left(-\frac{C_5}{\sigma^2(t)}\right), \quad \because C_5 > C_3 + \cdot|\ln C_4|$$

The above inequality implies that if there exists a monotone decreasing function such that $\sigma^2(s) \geq \frac{C_5}{\log(t+2)}$, it satisfies that the convergence condition given by equation 93 such that

$$\sum_{k=0}^{\infty} \delta_{t+k} \geq \sum_{k=0}^{\infty} \exp\left(-\frac{C_5}{C_5} \log(t+2+k)\right) = \sum_{k=0}^{\infty} \frac{1}{t+2+k} = \infty, \quad \forall k \geq 0. \tag{112}$$

Substitute equation 112 into equation 93, we obtain

$$\overline{\lim_{\tau \to \infty}} \sup_{\boldsymbol{X}_t, x_{t+\tau} \in \mathbf{R}^d} |p(t, \boldsymbol{X}_t, t+\tau, \boldsymbol{x}^*) - p(t, \boldsymbol{X}_t, t+\tau, \boldsymbol{x}^*)| \le \exp(-\sum_{k=0}^{\infty} \delta_{t+k})) = 0. \quad (113)$$

$\square$

## D.2 LOCAL CONVERGENCE UNDER CONVEX ASSUMPTION

**Assumption 4.** The Hessian of the objective function $\boldsymbol{H}(f) : \mathbb{R}^d \mapsto \mathbb{R}^d$ around the optimal point is non-singular and positive definite,

**Theorem 3.4** The expectation value of the objective function derived by the proposed QSGLD converges to a locally optimal point asymptotically under Assumption 4.

*Proof.* Given the learning equation derived by QSGLD as equation 52, we can calculate the one-step difference of the objective function as follows:

$$f(\boldsymbol{X}_{\tau+1}^Q) - f(\boldsymbol{X}_\tau^Q) = \langle \nabla_{\boldsymbol{x}} f(\boldsymbol{X}_\tau^Q), -\lambda \nabla_{\boldsymbol{x}} f(\boldsymbol{X}_\tau^Q) + Q_p^{-1}(\tau)\varepsilon_\tau^q \rangle$$
$$+ \lambda^2 \int_0^1 (1-s)\langle \nabla_{\boldsymbol{x}} f(\boldsymbol{X}_\tau^Q), \boldsymbol{H}_{\boldsymbol{x}}(f)(\boldsymbol{X}_\tau^Q + s(\boldsymbol{X}_{\tau+1}^Q - \boldsymbol{X}_\tau^Q))\nabla_{\boldsymbol{x}} f(\boldsymbol{X}_\tau^Q)\rangle ds, \quad (114)$$

where $\boldsymbol{H}_{\boldsymbol{x}}(f)(\cdot) : \mathbf{R}^d \to \mathbf{R}^{d \times d}$ denotes Hessian of the objective function $f$.

Assumption 4 and Definition 4 indicates that there exists the eigenvalue $M_{\max} \in \mathbf{R}^{+}+$ of the Hessian such that $(\boldsymbol{H}_{\boldsymbol{x}}(f))_{\mathbf{R}^{d \times d}} \le M_{\max}$. It implies

$$f(\boldsymbol{X}_{\tau+1}^Q) - f(\boldsymbol{X}_\tau^Q) \le \langle \nabla_{\boldsymbol{x}} f(\boldsymbol{X}_\tau^Q), -\lambda \nabla_{\boldsymbol{x}} f(\boldsymbol{X}_\tau^Q)\rangle + Q_p^{-1}(\tau)\langle \nabla_{\boldsymbol{x}} f(\boldsymbol{X}_\tau^Q), \varepsilon_\tau^q\rangle + \frac{1}{2}\lambda^2 M_{\max}\|\nabla_{\boldsymbol{x}} f(\boldsymbol{X}_\tau^Q)\|^2$$
$$= -\lambda\|\nabla_{\boldsymbol{x}} f(\boldsymbol{X}_\tau^Q)\|^2 + \frac{1}{2}\lambda^2 M_{\max}\|\nabla_{\boldsymbol{x}} f(\boldsymbol{X}_\tau^Q)\|^2 + Q_p^{-1}(\tau)\langle \nabla_{\boldsymbol{x}} f(\boldsymbol{X}_\tau^Q), \varepsilon_\tau^q\rangle$$
$$= \lambda\|\nabla_{\boldsymbol{x}} f(\boldsymbol{X}_\tau^Q)\|^2 \left(\frac{1}{2}\lambda M_{\max} - 1\right) + Q_p^{-1}(\tau)\langle \nabla_{\boldsymbol{x}} f(\boldsymbol{X}_\tau^Q), \varepsilon_\tau^q\rangle. \quad (115)$$

Furthermore, assuming the existence of the minimum eigenvalue $m_{\min} \in \mathbf{R}$ of the Hessian matrix $H$ such that $(H)_{\mathbf{R}^{d \times d}} \ge m_{\min}$, we can compute the difference between the optimal point $\boldsymbol{x}^* \in \mathbb{R}^d$ and $\boldsymbol{X}_\tau$ as follows:

$$f(\boldsymbol{x}^*) - f(\boldsymbol{X}_\tau^Q) \ge -\lambda\|\nabla_{\boldsymbol{x}} f(\boldsymbol{X}_\tau^Q)\|^2 + \frac{1}{2}\lambda^2 m_{\min}\|\nabla_{\boldsymbol{x}} f(\boldsymbol{X}_\tau^Q)\|^2$$
$$= \frac{1}{2}m_{\min}\|\nabla_{\boldsymbol{x}} f(\boldsymbol{X}_\tau^Q)\|^2 \left(\lambda^2 - \frac{2}{m_{\min}}\lambda\right)$$
$$= \frac{1}{2}m_{\min}\|\nabla_{\boldsymbol{x}} f(\boldsymbol{X}_\tau^Q)\|^2 \left(\left(\lambda - \frac{1}{m_{\min}}\right)^2 - \frac{1}{m_{\min}^2}\right) \ge -\frac{1}{2m_{min}}\|\nabla_{\boldsymbol{x}} f(\boldsymbol{X}_\tau)\|^2. \quad (116)$$

For convenience, we abbreviate $m_{\min}$ as $m$ and $M_{\max}$ as $M$. equation 116 implies that

$$\|\nabla_{\boldsymbol{x}} f(\boldsymbol{X}_\tau)\|^2 \ge 2m \left(f(\boldsymbol{x}^*) - f(\boldsymbol{X}_\tau)\right). \quad (117)$$

Let us assume that the learning rate is sufficiently small, such that $\lambda < \min\{1, \frac{2}{M}\}$. By substituting the inequality equation 117 into equation 116, we can derive the following equation:

$$f(\boldsymbol{X}_{\tau+1}^Q) - f(\boldsymbol{x}^*) + f(\boldsymbol{x}^*) - f(\boldsymbol{X}_\tau^Q)$$
$$\le \frac{1}{2}M \cdot 2m \left(f(\boldsymbol{X}_\tau^Q) - f(\boldsymbol{x}^*)\right)\left(\lambda^2 - \frac{2}{M}\lambda\right) + Q_p^{-1}(\tau)\langle \nabla_{\boldsymbol{x}} f(\boldsymbol{X}_\tau^Q), \varepsilon_\tau^q\rangle$$
$$\Rightarrow f(\boldsymbol{X}_{\tau+1}^Q) - f(\boldsymbol{x}^*) \le \left(1 + M \cdot m \left(\lambda^2 - \frac{2}{M}\lambda\right)\right)\left(f(\boldsymbol{X}_\tau^Q) - f(\boldsymbol{x}^*)\right) + Q_p^{-1}(\tau)\langle \nabla_{\boldsymbol{x}} f(\boldsymbol{X}_\tau^Q), \varepsilon_\tau^q\rangle. \quad (118)$$

Applying the expectation of the quantization to both terms, we obtain

$$
\begin{aligned}
&\mathbb{E}_{\boldsymbol{\varepsilon}_\tau^q} f(\boldsymbol{X}_{\tau+1}^Q) - \mathbb{E}_{\boldsymbol{\varepsilon}_\tau^q} f(\boldsymbol{x}^*) \\
&\leq \left(1 + M \cdot m\left(\lambda^2 - \frac{2}{M}\lambda\right)\right)\left(\mathbb{E}_{\boldsymbol{\varepsilon}_\tau^q} f(\boldsymbol{X}_\tau^Q) - \mathbb{E}_{\boldsymbol{\varepsilon}_\tau^q} f(\boldsymbol{x}^*)\right) + Q_p^{-1}(\tau)\langle \nabla_{\boldsymbol{x}} f(\boldsymbol{X}_\tau^Q), \mathbb{E}_{\boldsymbol{\varepsilon}_\tau^q} \boldsymbol{\varepsilon}_\tau^q\rangle \\
&= \left(1 + M \cdot m\left(\lambda^2 - \frac{2}{M}\lambda\right)\right)\left(\mathbb{E}_{\boldsymbol{\varepsilon}_\tau^q} f(\boldsymbol{X}_\tau^Q) - f(\boldsymbol{x}^*)\right),
\end{aligned}
\tag{119}
$$

where $\mathbb{E}_{\boldsymbol{\varepsilon}_\tau^q} f(\boldsymbol{x}^*) = f(\boldsymbol{x}^*)$.

To assess the convergence, we extend the inequality equation 119 to $t + k$ for $k > 0$.

$$
\mathbb{E}_{\boldsymbol{\varepsilon}_\tau^q} f(\boldsymbol{X}_{\tau+k}^Q) - f(\boldsymbol{x}^*) \leq \prod_{j=0}^{k-1}\left(1 + M \cdot m\left(\lambda^2 - \frac{2}{M}\lambda\right)\right)\left(\mathbb{E}_{\boldsymbol{\varepsilon}_\tau^q} f(\boldsymbol{X}_\tau^Q) - f(\boldsymbol{x}^*)\right).
\tag{120}
$$

The exponential lemma equation 66 to the equation 120 yields

$$
\begin{aligned}
&\mathbb{E}_{\boldsymbol{\varepsilon}_\tau^q} f(\boldsymbol{X}_{\tau+k}^Q) - f(\boldsymbol{x}^*) \\
&\leq \exp\left(M \cdot m \cdot \lambda \cdot \sum_{j=0}^{k-1}\left(\lambda - \frac{2}{M}\right)\right)\left(\mathbb{E}_{\boldsymbol{\varepsilon}_\tau^q} f(\boldsymbol{X}_\tau^Q) - f(\boldsymbol{x}^*)\right).
\end{aligned}
\tag{121}
$$

The assumption on $\lambda$ such that $\lambda < \min\{1, \frac{2}{M}\}$ implies the existence of a negative value, denoted as $\bar{h}^-(k)$, which depends on $k$ and is defined as

$$
\bar{h}^-(k) \triangleq \frac{1}{\lambda}\sum_{j=0}^{k-1}\left(\lambda - \frac{2}{M}\right) = \sum_{j=0}^{k-1}\left(1 - \frac{2}{\lambda M}\right) = -c_1 k, \quad \because c_1 = \left|1 - \frac{2}{\lambda M}\right| > 0.
\tag{122}
$$

Thus, we obtain

$$
\begin{aligned}
\mathbb{E}_{\boldsymbol{\varepsilon}_\tau^q} f(\boldsymbol{X}_{\tau+k}^Q) - f(\boldsymbol{x}^*) &\leq \exp\left(Mm\lambda^2 \bar{h}^-(k)\right)\left(\mathbb{E}_{\boldsymbol{\varepsilon}_\tau^q} f(\boldsymbol{X}_\tau^Q) - f(\boldsymbol{x}^*)\right) \\
&\leq \exp\left(-C_0 \cdot k\right)\left(\mathbb{E}_{\boldsymbol{\varepsilon}_\tau^q} f(\boldsymbol{X}_\tau^Q) - f(\boldsymbol{x}^*)\right),
\end{aligned}
\tag{123}
$$

where $C_0$ denotes $M \cdot m \cdot \lambda^2 \cdot c_1$.

The Lipschitz continuity assumption implies that

$$
\mathbb{E}_{\boldsymbol{\varepsilon}_\tau^q} f(\boldsymbol{X}_\tau^Q) - f(\boldsymbol{x}^*) \leq \sup_{\boldsymbol{X}_\tau^Q \in B^o(\boldsymbol{x}^*, \rho)} \|f(\boldsymbol{X}_\tau^Q) - f(\boldsymbol{x}^*)\| \leq L_0\|\boldsymbol{X}_\tau^Q - \boldsymbol{x}^*\| \leq L_0\rho
\tag{124}
$$

Consequently, applying an absolute value to both terms, we get

$$
\mathbb{E}_{\boldsymbol{\varepsilon}_\tau^q} f(\boldsymbol{X}_{\tau+k}^Q) - f(\boldsymbol{x}^*) \leq \exp\left(-C_0 \cdot k\right) L_0\rho = \exp\left(-C_0 \cdot k + \ln L_0\right)\rho
\tag{125}
$$

The result of equation 125 describes that for all $\rho > 0$, we can find an appropriate positive value $\delta > 0$, which implies $\delta = \exp(-c_0 k + a)\rho$ so that $\mathbb{E}_{\boldsymbol{\varepsilon}_\tau^q} f(\boldsymbol{X}_{\tau+k}^Q) \to f(\boldsymbol{x}^*)$. Therefore, we can pick a $k > k_0 = \left\lceil \frac{1}{C_0}\ln L_0\right\rceil$ satisfying the following proposition of convergence:

$$
\forall \varepsilon > 0, \ \exists \rho > 0 \text{ such that } \|\boldsymbol{X}_\tau^Q - \boldsymbol{x}^*\| < \rho \implies |\mathbb{E}_{\boldsymbol{\varepsilon}_\tau^q} f(\boldsymbol{X}_{\tau+k}^Q) - f(\boldsymbol{x}^*)| < \varepsilon(\rho).
\tag{126}
$$

$\square$

# E    DETAILED INFORMATION OF EXPERIMENTAL RESULTS

We conducted the experiments using a Python program based on the PyTorch framework version 1.13.1. For the experiments, we utilized three computers, and the detailed specifications of each computer are provided in Table 2. The Python version used was 3.10.0, and the Anaconda version was 23.10. We conducted the experiments for the FashionMNIST dataset using a vanilla CNN with three-layer blocks. For the CIFAR-10 and CIFAR-100 datasets, we used the ResNet-50 model with 56 layer blocks. The representative experimental results are presented in Table 3. A fixed learning rate of 0.01 is utilized in all experiments, and 200 epochs are conducted for all datasets. The batch sizes for FashionMNIST, CIFAR-10, and CIFAR-100 are 100 samples, 128 samples, and 100 samples, respectively.

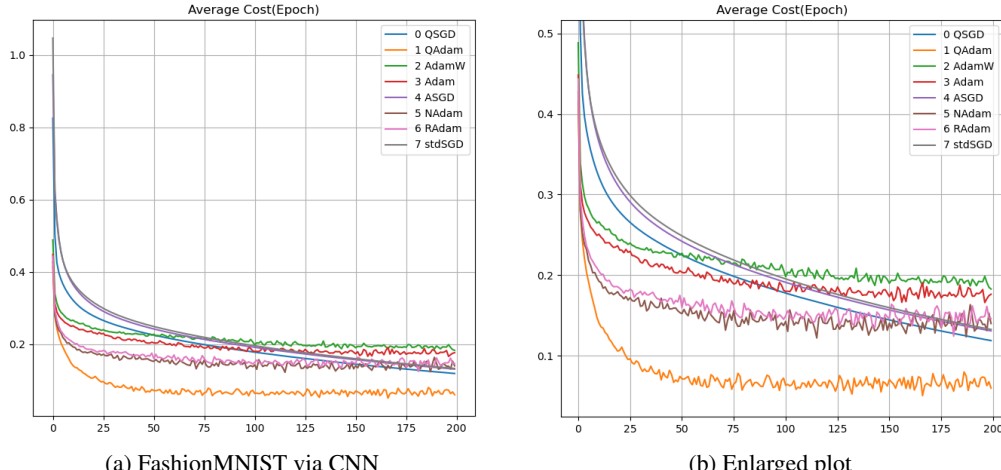

(a) FashionMNIST via CNN  (b) Enlarged plot

Figure 3: The error trends of test algorithms to the dataset and neural models: (a) Training error trends of the CNN model on FashionMNIST dataset.(b) Enlarged Training error trends

Table 2: Experimental Environment

| PC Name | OS | GPU | CPU |
|---|---|---|---|
| PC-1 | Linux Ubuntu 22.0 | NVIDIA GeForce GTX 1080Ti | Intel i9 7900 |
| PC-2 | Windows 11 | NVIDIA GeForce RTX 3050 | Intel i9 7900 |
| PC-3 | Windows 11 | NVIDIA GeForce GTX Ti | Intel i7 6700 |

## E.1 LEARNING EQUATIONS USED IN THE EXPERIMENT

**QSGLD and QSLD-ADAM**

The fundamental learning equation of the proposed algorithm is as follows:

$$\boldsymbol{X}_{\tau+1}^{Q} = \boldsymbol{X}_{\tau}^{Q} + Q_p^{-1}(\tau)\left[Q_p(\tau) \cdot \lambda h(\boldsymbol{X}_{\tau}^{Q})\right]^{Q}, \tag{127}$$

where $h(\boldsymbol{X}_{\tau}) = -\nabla_{\boldsymbol{x}} f((\boldsymbol{X}_{\tau})$ for QSGLD, and $h(\boldsymbol{X}_{\tau}) = -\frac{\hat{\boldsymbol{m}}_{\tau}}{\sqrt{\hat{\boldsymbol{v}}_{\tau}+\epsilon}}$ for Adam-based QSLD, respectively.

The quantization parameter $Q_p$ is defined as follows:

$$Q_p = \eta \cdot b^{\bar{p}(t_e)}. \tag{128}$$

Table 3: Comparison of test performance among optimizers with a fixed learning rate 0.01. Evaluation is based on the Top-1 accuracy of the training and testing data.

| Data Set | FashionMNIST | | | CIFAR10 | | | CIFAR100 | | |
|---|---|---|---|---|---|---|---|---|---|
| Model | CNN with 3-Layer Blocks | | | ResNet-50 (56 Layer Blocks) | | | | | |
| Algorithms | Training | Testing | Training Error | Training | Testing | Training Error | Training | Testing | Training Error |
| QSGD | 97.10 | 91.59 | 0.085426 | 99.90 | 73.80 | 0.009253 | 99.04 | 37.77 | 0.030104 |
| QADAM | 98.43 | 89.29 | 0.059952 | 99.99 | 85.09 | 0.011456 | 98.62 | 49.60 | 0.037855 |
| SGD | 95.59 | 91.47 | 0.132747 | 99.99 | 63.31 | 0.001042 | 98.24 | 25.90 | 0.005478 |
| ASGD | 95.60 | 91.42 | 0.130992 | 99.99 | 63.46 | 0.001166 | 98.36 | 26.43 | 0.004981 |
| ADAM | 92.45 | 87.12 | 0.176379 | 99.75 | 82.08 | 0.012421 | 98.85 | 46.32 | 0.038741 |
| ADAMW | 91.72 | 86.81 | 0.182867 | 99.57 | 82.20 | 0.012551 | 98.86 | 47.01 | 0.038002 |
| NADAM | 96.25 | 87.55 | 0.140066 | 99.56 | 82.46 | 0.014377 | 98.62 | 48.56 | 0.037409 |
| RADAM | 95.03 | 87.75 | 0.146404 | 99.65 | 82.26 | 0.010526 | 98.17 | 48.61 | 0.044193 |

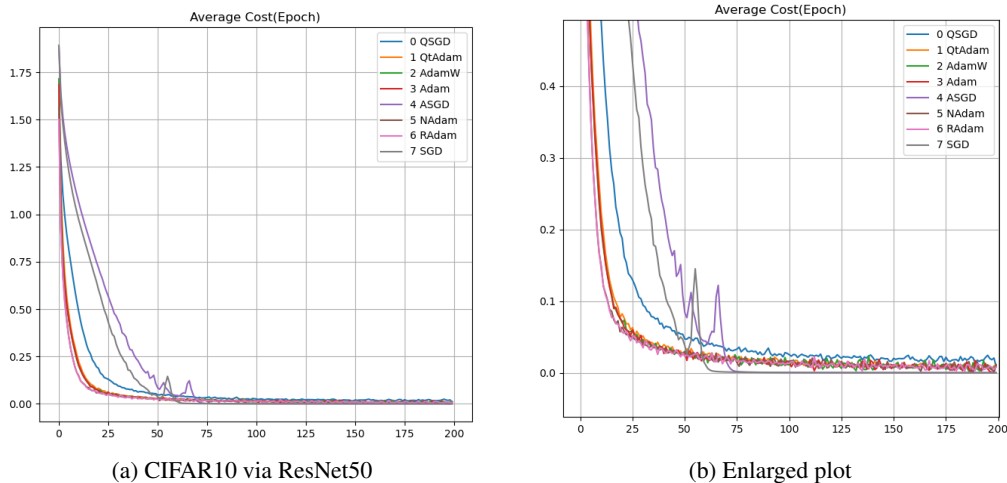

(a) CIFAR10 via ResNet50          (b) Enlarged plot

Figure 4: The error trends of test algorithms to the dataset and neural models: (a) Training error trends of ResNet-50 on CIFAR-10 dataset.(b) Enlarged Training error trends

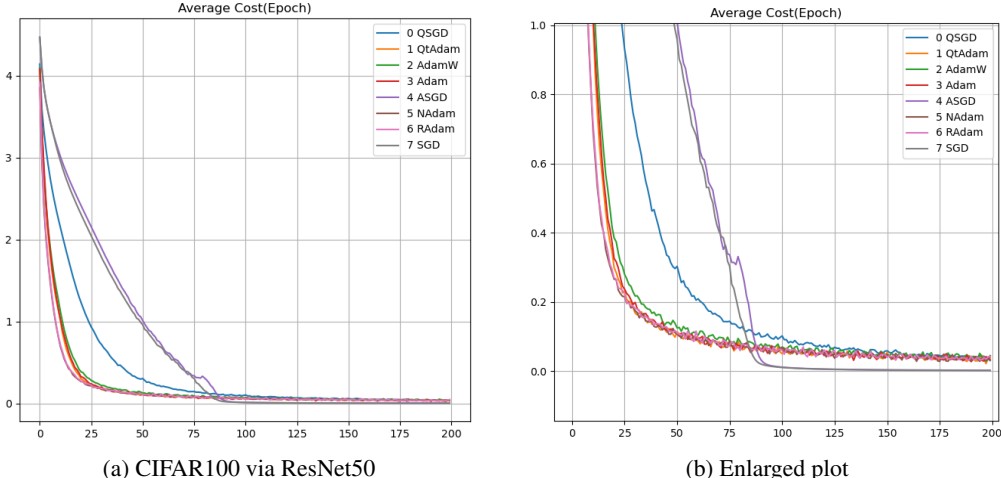

(a) CIFAR100 via ResNet50          (b) Enlarged plot

Figure 5: The error trends of test algorithms to the dataset and neural models: (a) Training error trends of ResNet-50 on CIFAR-100 dataset.(b) Enlarged Training error trends

where $t_e$ represents the time index of each epoch, given by $t_e = \frac{\tau}{B}$ for the number of mini-batches per epoch $B$, and $\bar{p}(t_e)$ denotes the power function, derived as follows:"

$$Q_p = \eta \cdot b^{\bar{p}(t_e)}\big|_{t_e=\tau/B} \leq \sqrt{\frac{1}{C}\log(\tau+2)}$$

$$b^{2\bar{p}(t_e)}\big|_{t_e=\tau/B} \leq \frac{1}{\eta^2 C}\log(\tau+2) \qquad (129)$$

$$\bar{p}(t_e)\big|_{t_e=\tau/B} \leq \frac{1}{2}\log_b\left(\frac{1}{\eta^2 C}\log(\tau+2)\right).$$

For convenience, we define the constant $C$ in equation 129 as the reciprocal value of $\eta^2$. Additionally, Considering the upper bound of the quantization parameter, which needs to be a rational number, we apply the floor function to the upper bound of the quantization parameter. Therefore, the power function for the quantization parameter is as follows:

$$\bar{p}(t_e)\big|_{t_e=\tau/B} \triangleq \lfloor 0.5 \cdot \log_b \log(\tau+2) \rfloor. \qquad (130)$$

Finally, we introduce an enforcement function to prevent early paralysis, defined as follows:

$$r(\tau, \boldsymbol{X}_\tau) = \lambda \cdot \left( \frac{\exp(-\varkappa(\tau - \tau_0))}{1 + \exp(-\varkappa(\tau - \tau_0))} \cdot \frac{h(\boldsymbol{X}_\tau)}{\|h(\boldsymbol{X}_\tau)\|} \right), \quad \tau_0 \in \mathbf{Z}^d, \tag{131}$$

where $\tau_0$ is a parameter, measured in mini-batches, that determines the interval for applying the enforcement function during the learning process. The parameter $\varkappa$ represents the shape of the enforcement function, with larger values causing a rapid decrease toward zero.

The following is the summary of all the equations for the proposed algorithm:

$$
\begin{aligned}
t_e &= \tau/B \\
\bar{p}(t_e) &= \lfloor 0.5 \cdot \log_b \log(\tau + 2) \rfloor \\
Q_p &= \eta \cdot b^{\bar{p}(t_e)} \\
r(\tau) &= \lambda \cdot \left( \frac{\exp(-\varkappa(\tau - \tau_0))}{1 + \exp(-\varkappa(\tau - \tau_0))} \cdot \frac{h(\boldsymbol{X}_\tau^Q)}{\|h(\boldsymbol{X}_\tau^Q)\|} \right) \\
\boldsymbol{X}_{\tau+1}^Q &= \boldsymbol{X}_\tau^Q + Q_p^{-1}(\tau) \left[ Q_p(\tau) \cdot \left( \lambda h(\boldsymbol{X}_\tau^Q) + r(\tau, \boldsymbol{X}_\tau^Q) \right) \right]^Q.
\end{aligned}
\tag{132}
$$

We recommend the hyper-parameters represented as follows:

$$\eta^2 \in 2^{19} \approx 0.5 \times 10^6, \ C = 1/\eta^2, \ b = 2 \ \kappa = 2.0 \text{ or } 4.0, \ t_0 = 5 \quad 20\% \text{ of all epochs.} \tag{133}$$

In the following section, we present an empirical analysis of the impact of changing hyperparameters.

**SGD** We set the SGD for the experiments using standard gradient descent form, as follows:

$$\boldsymbol{X}_{\tau+1} = \boldsymbol{X}_\tau - \lambda \nabla_{\boldsymbol{x}} f(\boldsymbol{X}_\tau). \tag{134}$$

**ASGD** (Average SGD by Shamir and Zhang (2013)) optimizer updates the parameters using the following equation:

$$\boldsymbol{X}_{\tau+1} = \frac{1}{t} \sum_{i=0}^{t-1} \nabla_{\boldsymbol{x}} f(\boldsymbol{X}_{\tau-i}), \tag{135}$$

where $\boldsymbol{X}_\tau$ represents the updated parameter at time step $\tau$.

**ADAM** (Adaptive Moment Estimation by Kingma and Ba (2015)) optimizer updates the parameters using the following equations:

$$
\begin{aligned}
\boldsymbol{m}_\tau &= \beta_1 \cdot \boldsymbol{m}_{\tau-1} + (1 - \beta_1) \cdot \boldsymbol{g}_\tau, \\
\boldsymbol{v}_\tau &= \beta_2 \cdot \boldsymbol{v}_{\tau-1} + (1 - \beta_2) \cdot \boldsymbol{g}_\tau^2, \\
\hat{\boldsymbol{m}}_\tau &= \frac{\boldsymbol{m}_\tau}{1 - \beta_1^\tau}, \\
\hat{\boldsymbol{v}}_\tau &= \frac{\boldsymbol{v}_\tau}{1 - \beta_2^\tau}, \\
\boldsymbol{X}_\tau &= \boldsymbol{X}_{\tau-1} - \frac{\eta}{\sqrt{\hat{\boldsymbol{v}}_\tau} + \epsilon} \cdot \hat{\boldsymbol{m}}_\tau,
\end{aligned}
\tag{136}
$$

where $\boldsymbol{m}_\tau \in \mathbf{R}^d$ and $\boldsymbol{v}_\tau \in \mathbf{R}^d$ are the first and second moment estimates respectively, $\boldsymbol{g}_\tau \in \mathbf{R}^d$ is the gradient at time step $t$ i.e. $-\nabla_{\boldsymbol{x}} f(\boldsymbol{X}_t)$, $\beta_1$ and $\beta_2$ are the decay rates for the moments, $\hat{\boldsymbol{m}}_\tau \in \mathbf{R}^d$ and $\hat{\boldsymbol{v}}_\tau \in \mathbf{R}^d$ are the bias-corrected moment estimates, $\boldsymbol{X}_\tau$ represents the updated parameter at time step $t$, $\eta$ is the learning rate, and $\epsilon$ is a small constant to avoid division by zero. We set the hyperparameters for ADAM such that

$$\beta_1 = 0.9, \ \beta_2 = 0.999, \ \epsilon = 10^{-8}. \tag{137}$$

We utilize the ADAMW optimizer implemented in PyTorch.

**NADAM** (Nestrov momentum incooperated ADAM by Dozat (2016))

$$\mu_\tau = \beta_1 \left(1 - \frac{1}{2}0.96^{\tau\psi)}\right)$$

$$\mu_{\tau+1} = \beta_1 \left(1 - \frac{1}{2}0.96^{(\tau+1)\psi}\right)$$

$$\boldsymbol{m}_\tau = \beta_1 \cdot \boldsymbol{m}_{\tau-1} + (1 - \beta_1) \cdot \boldsymbol{g}_\tau,$$

$$\boldsymbol{v}_\tau = \beta_2 \cdot \boldsymbol{v}_{\tau-1} + (1 - \beta_2) \cdot \boldsymbol{g}_\tau^2, \tag{138}$$

$$\hat{\boldsymbol{m}}_\tau = \frac{\mu_{\tau+1}}{1 - \prod_{i=1}^{t+1}} \boldsymbol{m}_\tau + \frac{1 - \mu_\tau}{1 - \prod_{i=1}^{t}} \boldsymbol{g}_\tau,$$

$$\hat{\boldsymbol{v}}_\tau = \frac{\boldsymbol{v}_\tau}{1 - \beta_2^\tau},$$

$$\boldsymbol{X}_\tau = \boldsymbol{X}_{\tau-1} - \frac{\eta}{\sqrt{\hat{\boldsymbol{v}}_\tau} + \epsilon} \cdot \hat{\boldsymbol{m}}_\tau,$$

**RADAM** (Rectified Adam by Liu et al. (2020)) optimizer updates the parameters using the following equations:

$$\boldsymbol{m}_\tau = \beta_1 \cdot \boldsymbol{m}_{\tau-1} + (1 - \beta_1) \cdot \boldsymbol{g}_\tau,$$

$$\boldsymbol{v}_\tau = \beta_2 \cdot \boldsymbol{v}_{\tau-1} + (1 - \beta_2) \cdot \boldsymbol{g}_\tau^2,$$

$$\hat{\boldsymbol{m}}_\tau = \frac{\boldsymbol{m}_\tau}{1 - \beta_1^\tau},$$

$$\hat{\boldsymbol{v}}_\tau = \frac{\boldsymbol{v}_\tau}{1 - \beta_2^\tau},$$

$$\rho_\tau = \rho_\infty - \frac{2t\beta_2^t}{1 - \beta_2^t} \tag{139}$$

$$\boldsymbol{r}_\tau = \sqrt{\frac{(\rho_\tau - 4)(\rho_\tau - 2)\rho_\infty}{(\rho_\infty - 4)(\rho_\infty - 2)\rho_\tau}},$$

$$\boldsymbol{X}_\tau = \boldsymbol{X}_{\tau-1} - \eta \, \boldsymbol{m}_\tau \cdot \begin{cases} \boldsymbol{r}_\tau \frac{\sqrt{1-\beta_2^\tau}}{\sqrt{\boldsymbol{v}_\tau}+\epsilon} & \rho_t > 5 \\ 1 & \text{else} \end{cases},$$

where $\boldsymbol{r}_\tau \in \mathbf{R}^d$ is the "leaky" update term, and $\rho_\tau, \rho_\infty$ is an additional hyperparameter introduced in RADAM. $\rho_\infty$ is initialized with $\rho_\infty = \frac{2}{1-\beta_2} - 1$. We utilize the RADAM optimizer implemented in PyTorch.

**ADAMW** optimizer updates the parameters using the following equations:

$$\boldsymbol{m}_\tau = \beta_1 \cdot \boldsymbol{m}_{\tau-1} + (1 - \beta_1) \cdot \boldsymbol{g}_\tau,$$

$$\boldsymbol{v}_\tau = \beta_2 \cdot \boldsymbol{v}_{\tau-1} + (1 - \beta_2) \cdot \boldsymbol{g}_\tau^2,$$

$$\hat{\boldsymbol{m}}_\tau = \frac{\boldsymbol{m}_\tau}{1 - \beta_1^\tau}, \tag{140}$$

$$\hat{\boldsymbol{v}}_\tau = \frac{\boldsymbol{v}_\tau}{1 - \beta_2^\tau},$$

$$\boldsymbol{X}_\tau = \boldsymbol{X}_{\tau-1} - \frac{\eta}{\sqrt{\hat{\boldsymbol{v}}_\tau} + \epsilon} \cdot (\hat{\boldsymbol{m}}_\tau + \lambda \boldsymbol{X}_{\tau-1}),$$

where $\lambda$ is a weight decay coefficient or regularization term added in ADAMW. Loshchilov and Hutter (2019) provided the algorithm. We utilize the ADAMW optimizer implemented in PyTorch.

### E.2 EXPERIMENTAL RESULTS ACCORDING TO DATASETS

**FashionMNIST** The FashionMNIST dataset is a drop-in replacement for the original MNIST dataset, which consists of handwritten digits. The FashionMNIST dataset contains 60,000 grayscale images

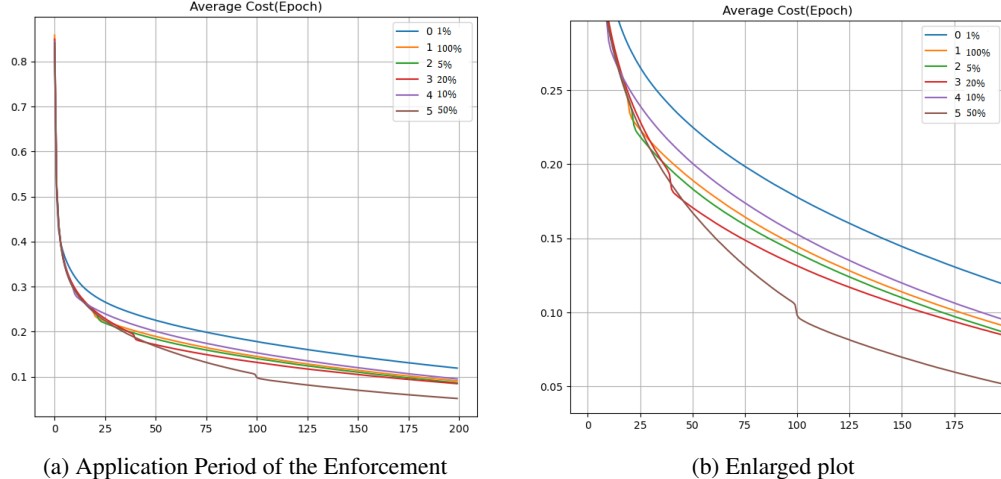

(a) Application Period of the Enforcement        (b) Enlarged plot

Figure 6: The error trends of performance Variation of Adam-Based QSLD based on the application period of the enforcement function: (a) Training error trends of Vanilla CNN on FashionMNIST dataset.(b) Enlarged Training error trends

Table 4: Performance Variation of Adam-Based QSLD Based on the Application Period of the Enforcement Function

| Period Ratio (%) | FashionMNIST | | | CIFAR-10 | | | CIFAR-100 | | |
|---|---|---|---|---|---|---|---|---|---|
| | Training | Testing | Error | Training | Testing | Error | Training | Testing | Error |
| 0.5 | 96.05 | 91.04 | 0.92853 | 99.74 | 83.61 | 0.007327 | 98.66 | 44.51 | 0.036558 |
| 1.0 | 96.11 | 91.59 | 0.118914 | 99.84 | 83.29 | 0.006766 | 98.62 | 44.38 | 0.032726 |
| 5.0 | 96.75 | 91.38 | 0.094954 | 99.87 | 83.39 | 0.006549 | 98.61 | 49.60 | 0.037855 |
| 10.0 | 96.95 | 91.59 | 0.090756 | 99.84 | 84.18 | 0.0 | 98.84 | 44.59 | 0.035554 |
| 20.0 | 97.05 | 90.84 | 0.084489 | 100.0 | 85.08 | 0.0 | 99.13 | 46.78 | 0.035354 |
| 50.0 | 97.82 | 90.64 | 0.051491 | 99.91 | 84.04 | 0.005456 | 98.94 | 48.49 | 0.029179 |
| 100.0 | 99.46 | 90.01 | 0.104510 | 99.87 | 83.39 | 0.010629 | 98.81 | 46.76 | 0.044370 |

of 10 different fashion categories, each with a 28x28 pixel representation. The ten fashion categories in FashionMNIST include T-shirts/tops, trousers, pullovers, dresses, coats, sandals, shirts, sneakers, bags, and ankle boots. Each image in the dataset is associated with a corresponding label indicating the category of the depicted fashion item.

Simple vanilla multilayer networks, equipped with well-tuned optimizers and moderately wide hidden layers, can achieve high accuracy in classifying each category of the MNIST dataset, resulting in minimal accuracy errors. This poses challenges in meaningfully evaluating the performance of different optimizers.

However, while the classification test scores for FashionMNIST are higher than CIFAR-10 and CIFAR-100, standard SGD performs superior to the ADAM optimizer family in evaluation tests on the FashionMNIST dataset. This result suggests that the objective function of FashionMNIST exhibits a more convex property around the optimal point, and previous research (Xie et al. (2021)) has revealed that the ADAM optimizer may struggle to select flat minima.

**Experiments to FashionMNIST** The experiments on the FashionMNIST dataset yielded interesting results. Firstly, similar to the MNIST dataset, the experimental results showed that the training classification performance exceeded 90% for all models. However, there was a noticeable performance difference between the SGD and ADAM optimizers. The SGD optimizer exhibited better classification performance compared to the ADAM optimizer. As shown in Figure 3, the error trend indicated that ADAM converged faster initially, but as the number of epochs increased, SGD gradually reduced the error more effectively than the ADAM optimizer. In general, once the number of epochs exceeded 400, the SGD optimizer consistently achieved a training classification accuracy of 100% regardless of the learning rate. However, the test classification accuracy reached a limit of around 91.25%.

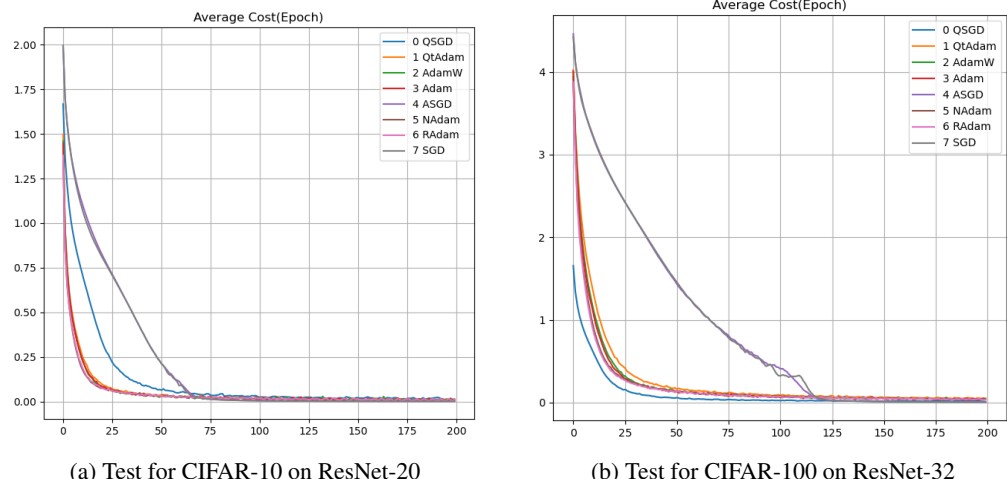

(a) Test for CIFAR-10 on ResNet-20      (b) Test for CIFAR-100 on ResNet-32

Figure 7: Performance Comparison of the Proposed Algorithm on Networks with Different Depths: (a) When testing on the CIFAR-10 dataset, the algorithm was evaluated on ResNet-20.(b) When testing on the CIFAR-100 dataset, the algorithm was evaluated on ResNet-30.

The proposed QSGLD and ADAM-based QSLD perform better than the conventional SGD and ADAM optimizer, but they do not exhibit significant improvements on the FashionMNIST dataset. QSGLD achieves a slight improvement of approximately 0.2% in classification accuracy for both training and testing, while Adam-Based QSLD achieves an improvement of around 2%. In Figure 3a, it can be observed that QSGLD converges faster than SGD due to the similar learning rate, although the convergence speed is similar. On the other hand, Adam-based QSLD exhibits a convergence trend similar to the Adam optimizer. Nevertheless, Adam-based QSLD shows lower error trends than conventional ADAM optimizers.

**CIFAR10 and CIFAR100** We don't provide any specific explanation for the CIFAR-10 and CIFAR-100 datasets. For the CIFAR datasets, separate experiments were conducted based on the depth of ResNet. This was done to verify that the proposed algorithm works effectively regardless of the depth of the neural network model. The experimental results demonstrated the superiority of the proposed algorithm for all depths of ResNet. As shown in Table 3, when classifying the CIFAR-10 dataset using ResNet-50, QSGD outperformed SGD by 8% in terms of test accuracy.

As depicted in Figure 4, QSGLD exhibited significant improvements in both convergence speed and error reduction compared to the conventional SGD methods. On the other hand, Adam-Based QSLD showed a performance advantage of approximately 1.5% for test accuracy.

Similar trends were observed for the CIFAR-100 dataset. QSGD demonstrated a performance advantage of around 11% over conventional SGD methods for test accuracy. In contrast, Adam-based QSLD showed an improvement of approximately 1.0% compared to conventional Adam-based optimizers.

The experiments on the CIFAR datasets revealed interesting characteristics of QSGLD and Adam-based QSLD.

### E.3 EXPERIMENTAL RESULTS OF CHANGING OF HYPER-PARAMETERS

**Period Parameter for Enforcement Function** We conduct experiments to analyze the effect of the period parameter $\tau_0$ for the enforcement function. Regarding the quantization parameter, there were variations depending on the algorithm, but generally, the size of the search vector was around $10^{-6}$, so the eta value was set to half of it, which is 524288. Firstly, the optimal application period of the enforcement function varied depending on the dataset. For FashionMNIST and CIFAR-10, the best performance improvement was observed when the enforcement function was applied for about 10-20% of the epochs. Applying it for longer or shorter periods did not result in significant performance improvement. On the other hand, for CIFAR-100, the best performance improvement was achieved

when the enforcement function was applied throughout all the epochs. Since CIFAR-100 has a higher classification complexity than FashionMNIST or CIFAR-10, it is considered optimal to integrate the enforcement function with a measure that can assess problem complexity, such as the Fisher Information Matrix, rather than using it as a function of time. Regarding another hyperparameter, $\kappa$, no significant performance variations were observed with changes in its value.

