# OpenReview forum: "Stochastic Gradient Langevin Dynamics Based on Quantization with Increasing Resolution"
_ICLR.cc/2024/Conference — Submitted to ICLR 2024_

### Official Review · Reviewer_JTTw · 2023-10-27

**Soundness:** 2 fair
**Presentation:** 2 fair
**Contribution:** 2 fair
**Rating:** 3
**Confidence:** 3

**Summary:**

This paper uses the quantized optimization to propose a stochastic descent learning equation, and combines it with SLD/SGLD to propose two alternative algorithms. The methods utilize Langvin SDE dynamics and aloow for controllable noise with an identical distribution without need for additive noise or adjusting minibatch size. Numerical experiments are carried out on CNN and ResNet-50 Models.

**Strengths:**

1. The paper applies the quantized optimization theory, which is primarily used to reduce computational burden, to optimization algorithms and uses Langevin SDEs for analysis. The originality is high.
2. Both theoretical analysis and experiments are presented, so this work is self-contained and easy to follow.

**Weaknesses:**

1. The uniform distribution assumption (Assumption 2) seems to be too stringent and unverifiable, since the quantization error $\epsilon^q$ depends highly on iterates. Throughout the anaysis in paper this assumption is crucial, so it is better to try explaining its validity in detail. For example, the authors could experiment on some simulation/real datasets to see if this assumption is satisfied.
2. The advantage of applying quantized optimization is not clearly stated. It would be better if clear motivation of using quantized method, or its computational or analytical benefits are claimed to convince readers.
3. The key point of this paper is somewhat ambiguous. If the major contribution lies in theoretical analysis, the authors should emphasize it and conduct simulation experiments to validate the stochastic approximation and convergence results, rather than merely performing real data experiments; if the contribution lies in experimental results, models like CNN and ResNet may seem to be too simple. Larger datasets and model structures should be tested to verify the robustness and efficiency of the proposed method.
4. Some typos:
 - lines after an equation should not start with comma, e.g. line after (12) (15) (18)...
 - 'Langvine' should be 'Langevin' in line before (12);
 - 'evalutae' should be 'evaluate' in line before (9);
 - 'lsyers' should be 'layers' in Table 1.

**Questions:**

1.Can you explain the claim in 'Appliance to Other Learning Algorithms'? It seems ADAM/ADAMW type methods require information of past iterates, so these may result in past-dependent SDEs which is different from Langevin dynamics.

---

> ### Author Response · Authors · 2023-11-17
> **Answer for the reviewer  JTTw (1)**
>
> We appreciate the thoroughness and thoughtfulness of your comments.
> Particularly, we'd like to answer for the 1 and 2 in the weakness part as a unified response.
>
> ### Opinion to Weakness
>
> The uniform distribution assumption (Assumption 2) is a widely accepted postulation(almost regarded as an axiom) for the distribution of the quantization error(or noise) for tens of years in signal processing.
> Therefore, the distribution of the quantization error at a specific time concerning a particular point being a uniform distribution is a valid assumption even though the system contains a recursive loop.
> Meanwhile, as the reviewer states, the quantization error for a fixed state during a time interval in the recursive system, such as the learning equation in DNN, does not follow the uniform distribution.
> Thus, we should analyze the distribution of such quantization error by the summation or integration for the time interval, and we provided a detailed analysis based on the classical (or Lindberg-L'evy) central limit theorem in Appendix C.
>
> Heretofore, the analysis of the stochastic learning equation, such as SGD and ADAM, depends on what kind of CLT applies to the summation of the gradient noise.
> Since the calculation of the distribution for the gradient noise through a rigorous analysis is so hard work, researchers have intuitively assumed the distribution of the gradient error from the noise trends as follows:
>
> 1. The assumption of the identical independent distributed gradient noise with the finite second moment implies that the gradient noise is the Gaussian noise random variable by the classical CLT.  This assumption yields the standard Langvine SDE for the learning equation as follows:
> $$
> \boldsymbol{x}\_{\tau+1} = \boldsymbol{x}\_{\tau} - \lambda \nabla f(\boldsymbol{x}\_{\tau}) + \sqrt{\lambda} \boldsymbol{\xi}\_{\tau}
> \implies d\boldsymbol{X}\_t = -\nabla f(\boldsymbol{X}\_t) + \sigma \sqrt{\lambda/B} d\boldsymbol{B}\_t
> $$
> , where $\boldsymbol{\xi} \in \mathbb{R}^d$ denotes the Gradient noise defined as $\boldsymbol{\xi} \triangleq \nabla f(\boldsymbol{x}\_{t}) - \nabla \tilde{f}\_{\tau}(\boldsymbol{x}\_{t})$ with the assumption of $\mathbb{E} \boldsymbol{\xi} = 0$ and $\mathbb{E} \boldsymbol{\xi}^2 = \sigma/B$, and other notations are defined in the manuscript.
>
> 2. According to the data analysis of the gradient noise representing a heavy-tailed or long-tailed noise, the distribution of the gradient noise is a symmetric $\alpha$-stable (S$\alpha$S) distribution derived from the general central limit theorem(GCLT).
> This assumption yields a Le'vy SDE for the stochastic learning equation as follows:
> $$
> \boldsymbol{x}\_{\tau+1}
> = \boldsymbol{x}\_{\tau} - \lambda \nabla f(\boldsymbol{x}\_{\tau}) + \lambda^{1/\alpha} \left( \lambda^{\frac{\alpha-1}{\alpha}} \sigma \right) \boldsymbol{\xi}\_{\tau}
> \implies
> d \boldsymbol{X}\_t = - \nabla f(\boldsymbol{X}\_t)dt + \lambda^{(\alpha-1)/\alpha} \sigma d\boldsymbol{L}\_t^{\alpha}
> $$
> , where $[\boldsymbol{\xi}\_{\tau}]_i \sim \mathcal{S}\alpha\mathcal{S}(1)$ and $\boldsymbol{L}\_t^{\alpha} \in \mathbb{R}^d$ denotes $\alpha$-stable Le'vy process.
> Consequently, the conventional analysis of the SDE approximation for the stochastic learning equation depends on the iterative statistical property of the gradient noise.
>
> Whereas, the quantized learning we proposed depends on the quantization error with the axiomatic uniform distribution containing finite variance, not on the gradient error.
> As you note, the random variable with the uniform distribution representing a zero expectation and a finite variance yields a random variable with Gaussian distribution derived by the accumuated quantization error to a unit epoch, through the classical CLT.
> Thus, we claim that the proposed quantization-based scheme can provide the standard Langevin SDE for the stochastic learning equation, regardless of the gradient noise as follows:
> $$
> \boldsymbol{x}\_{\tau+1}^Q = \boldsymbol{x}\_{\tau}^Q - \lambda \nabla f(\boldsymbol{x}\_{\tau}^Q) + Q_p^{-1}(\tau) \boldsymbol{\varepsilon}\_{\tau}^q
> \implies
> d\boldsymbol{X}\_t = -\nabla f(\boldsymbol{X}\_t) + \sigma \sqrt{\lambda} dW\_t
> $$
> We agree with the reviewer's comment on the validity of the assumption that the quantization error follows uniform distribution and other statistical properties.
> However, in our opinion, this work is beyond the scope of the paper.
> In our paper, we propose a novel learning equation and provide the fundamental idea, additional operational conditions to avoid early paralysis, convergence properties of the proposed algorithm, and experimental results.
> The analysis and providing empirical proof for the proposed algorithm would be another topic for us.

---

> ### Author Response · Authors · 2023-11-17
> **Answer for the reviewer JTTw (2)**
>
> #### Motivation and Advantages of Quantization
>
> Generally, engineers have considered quantization as one of the effective schemes providing less computation burden and fast signal processing in the device with low computation power.
> Meanwhile, we consider quantization as a scheme for optimization, inspired by direct search algorithms such as the generating set search(GSS) algorithm.
> Most of all, if we can establish the stochastic learning algorithm as Langevin dynamics rigorously, we can develop a more effective algorithm satisfying global optimization based on not only classical thermodynamics but also quantum mechanics.
> For instance, the analysis based on Le’vy-SDE considers the avoidance of local minima as a jump process.
> Whereas, Langevin-SDE-based analysis can illustrate it as various physical phenomena, such as a statistical property in thermodynamics, tunneling effect in quantum mechanics, and Laplace method accompanied by weak convergence.
>
>
> ### Answer for Questions
>
> According to the recent research on the stochastic learning equation[1][2][3], the gradient noise of ADAM still represents heavy-tailed noise despite the generation of an average gradient driven by momentum operation.
> Hence, as the reviewer states, it would be reasonable to analyze the stochastic differential property of ADAM/ADAMW as the Le'vy-driven SDE.
>
> Whereas, since the quantization-based learning forces the accumulation of quantization error to be a Gaussian random variable by the classical CLT, we can obtain the Langevin SDE even though the search direction is not a gradient descent.
>
> ### Fixing Typos
>
> I correct the typos those the reviewer points out, and submit the revised version of the manuscript.
>
> ## Reference
>
> [1] Pan Zhou, Jiashi Feng, Chao Ma, Caiming Xiong, Steven Hoi, and E. Weinan. "Towards theoretically understanding why SGD generalizes better than ADAM in deep learning", In Proceedings of the 34th International Conference on Neural Information Processing Systems (NIPS'20). Curran Associates Inc., Red Hook, NY, USA, Article 1787, 21285–21296. (2020).
>
> [2] Simsekli, Umut, Levent Sagun and Mert Gürbüzbalaban. “A Tail-Index Analysis of Stochastic Gradient Noise in Deep Neural Networks.” ArXiv abs/1901.06053 (2019)
>
> [3] Nguyen, Thanh Huy, Umut Simsekli, Mert Gürbüzbalaban and Gaël Richard. “First Exit Time Analysis of Stochastic Gradient Descent Under Heavy-Tailed Gradient Noise.” Neural Information Processing Systems (2019).

---

### Official Review · Reviewer_wZWw · 2023-10-31

**Soundness:** 2 fair
**Presentation:** 3 good
**Contribution:** 2 fair
**Rating:** 5
**Confidence:** 3

**Summary:**

This paper introduces a quantization scheme for existing optimization algorithms (SGD, ADAM). The quantization error can be treated as an additive noise thus improves the performance of the algorithm. With increasing resolution, the convergence of the quantized optimization algorithms can be proven. Numerical experiments with CV tasks show that optimization algorithms based using this quantization scheme have better performance and robustness compared with existing algorithms.

**Strengths:**

This paper uses the quantization error, the byproduct of optimization at the implementation level, as an approach of improving the performance of the optimization algorithm at the algorithm design level.  It contains the necessary theoretical analysis of linking the quantization error to additive noise and the convergence of the algorithm, as well as the numerical evidence demonstrating the superior performance of the quantized algorithm. The ideas in the paper are organized well.

**Weaknesses:**

It appears to me that the writing of the paper could be improved:
* Certain notation in the manuscript lacks consistency. Notably, when $\tau$ is first introduced in equation 1 and 2, I thought it would be the index of the batches and that $\tau < B$. However, in equation 10, $\tau$ can be arbitrarily large.
* It appears to me that the ‘transition probability’ in theorem 3.3 should be called ‘transition kernel’, as the probability of transiting from one point to another should be zero.
* The appendix contains a lot of good supplementary information. It is a shame that the main text does not refer to the appendix.
* Should the $Q_p^{-1}(t_e + \tau)$ on top of equation (15) be $Q_p^{-1}(t_e + \tau/B)$?
* Typo on top of equation (9): evalutae
* Typo in table 2: GTXTi

This paper uses a complicated quantization scheme while providing no detail regarding the implementation of the algorithm in the text (data type, time and memory it takes to train the network, etc.). Without this information, it is hard to tell whether the quantized algorithm benefits from quantization in terms of reducing computational burden and simplifying data processing. Consequently, between the injected additive noise (easy to implement) and the quantized algorithms (hard to implement), it is hard to tell which one is better as they are supposed to improve the performance in the same way. This makes assessing the significance of this work hard.

**Questions:**

* Following the weaknesses, I wonder if the authors could provide more information regarding the implementation of the quantized algorithm. I checked the attached program. If I am not mistaken, the quantization is neither implemented via torch API nor casting high-precision tensors into low-precision ones.
* I wonder if the authors checked experimentally that if the performance gain from the quantization can be reproduced by injecting additive noise.
* In the introduction, when the Non-Convex Optimization is introduced, the authors mention that the quantized algorithms have better performance in certain problems than the MCMC algorithms. I wonder if the authors could provide some reference for that.

---

> ### Author Response · Authors · 2023-11-21
> **Answer for the reviewer wZWw (1)**
>
> We appreciate the thoroughness and thoughtfulness of your comments.
>
> ### Answer for the weakness point
> (1) In section A.2 of the Appendix, we denote the $\tau$ as the discrete-time index based on the index of mini-batches, so $\tau$ can increase to infinity as the epoch increases.
> Additionally, we denoted all the notations used in the Appendix of the manuscript, and we established the set for the parameters such as $\forall \tau \in \mathbb{Z}[0, B)$, when the parameters require a range.
> Although our intention, the mini-batch-based time index still confuses readers, as the reviewer comments.
>
> Consequently, we establish the index of mini-batch $\tau_b \in \mathbb{Z}[0, B)$ and the time index based on the mini-batch $\tau \in \mathbb{Z}^+$ such that $\tau_b = \tau \% B$.
>
> We revise all the equations employing $\tau$ carefully in the manuscript to prevent confusion as the reviewer comments and add the definition of $\tau_b$ in section A.1 of the Appendix.
>
> (2) The reviewer's comment that the transition probability is not a correct statement for $p(t, \boldsymbol{x}, t+\bar{\tau}, \boldsymbol{x}^*)$ is right.
> We modify it as the transition probability density since we derive the SDE for the proposed algorithm in Lemma 3.2.
>
> (3) We organized the manuscript providing our fundamental idea on the main pages and analysis and proof of theorems elaborating the idea in the Appendix.
> The reviewer's comments encouraged us to raise our willingness to continue the research. I appreciate the reviewer's comments.
>
> (4) Yes, $Q_p^{-1}(t_e + \tau)$ should be equal to $Q_p^{-1}(t_e + \tau/B)$ for all $\tau \in [0, B)$. In other words, $Q_p(t)^{-1}$ should be constant for a unit epoch and decrease as the epoch increases.
> Based on this condition, we can derive a Gaussian random variable to the epoch time from the uniform distributed quantization error through the central limit theorem(CLT).
>
> (5 and 6) We revise the typo as the reviewer's comments.
>
> ### Answer to the question
> (1) As the reviewer comments, we implemented the presented quantization without PyTorch API, whereas we implemented other algorithms such as ADAM with PyTorch.
> Since our research aims to verify that quantization is another effective optimization scheme, we implemented the quantization with our own Python code for our purpose.
> Many researchers consider the quantization in optimization algorithms for AI as an effective methodology for fast and light computation.
> As a result, the quantization API in PyTorch is not appropriate for our research.
> However, we're going to improve our research and the code more practically using widely used quantization API.
>
> (2) Unfortunately, we didn't compare the proposed algorithm to other models with an injecting additive noise.
>
> However, we guess that comparing the proposed model to the additive noise model is an interesting theme.
> We can write the additive noise model for a mini-batch-based time index $\tau$ such that
> $$
> \begin{aligned}
> \boldsymbol{x}\_{\tau+1} &= \boldsymbol{x}\_{\tau} - \lambda \nabla\_{\tau\_b} \tilde{f}({x}\_{\tau} ) + \sigma(\tau) \boldsymbol{\xi}\_{\tau}\\\\
> &= \boldsymbol{x}\_{\tau} - \lambda (\nabla\_{\tau\_b} \tilde{f}({x}\_{\tau}) + \nabla f(\boldsymbol{x}\_t) - \nabla f(\boldsymbol{x}\_t) )  + \sigma(\tau) \boldsymbol{\xi}\_{\tau}, & \nabla f(\boldsymbol{x}\_t) \triangleq \mathbb{E} \nabla\_{\tau_b} \tilde{f} ({x}\_{\tau} )  \\\\
> &= \boldsymbol{x}\_{\tau} - \lambda \nabla f(\boldsymbol{x}\_t) + \lambda \boldsymbol{\eta}\_{\tau} + \sigma(\tau) \boldsymbol{\xi}\_{\tau}, &\boldsymbol{\eta}\_{\tau} \triangleq \nabla f(\boldsymbol{x}\_t) - \nabla\_{\tau\_b} \tilde{f}({x}\_{\tau}),
> \end{aligned}
> $$
> where $\xi_{\tau} \in \mathbb{R}^d$ denotes an additive noise such as $\xi_{\tau} \sim \mathcal{N}(\boldsymbol{0}, \boldsymbol{I})$, $\tau_b \in \mathbb{Z}[0, B)$ denotes the mini-batch index defined as above, and all other notations in the equation are defined in the manuscript.
>
> If we set the expectation gradient as the average gradient for all samples defined in equation (2) in the manuscript, we can obtain the first moment of the gradient noise such that
> $$
> \mathbb{E}\boldsymbol{\eta}\_{\tau}
> \triangleq \frac{1}{B} \sum\_{{\tau}\_b=0}^{B-1} \boldsymbol{\eta}\_{\tau_b}
> = \frac{1}{B} \sum\_{{\tau}\_b=0}^{B-1} \nabla f(\boldsymbol{x}\_t) - \frac{1}{B} \sum_{{\tau}\_b=0}^{B-1} \nabla_{\tau\_b} \tilde{f}({x}\_{\tau}) = \nabla f(\boldsymbol{x}\_t) \cdot \frac{1}{B} \cdot B - \nabla f(\boldsymbol{x}\_t) = 0, \quad \quad
> \because \frac{1}{B} \sum\_{{\tau}\_b=0}^{B-1} \nabla\_{\tau\_b} \tilde{f}({x}\_{\tau}) = \nabla f(\boldsymbol{x}\_t).
> $$
>
> Equation (1) implies that the expectation of the summation of the gradient noise and additive noise $\bar{\boldsymbol{z}}\_{\tau} \triangleq \lambda \boldsymbol{\eta}\_{\tau} + \sigma(\tau) \boldsymbol{\xi}$ is zero.

---

> ### Author Response · Authors · 2023-11-21
> **Answer for the reviewer wZWw (2)**
>
> However, if we don't assume the independent assumption,  we can obtain the second moment of $\bar{\boldsymbol{z}}\_{\tau}$ ambiguously such that
> $$
> \mathbb{E} \bar{\boldsymbol{z}}\_{\tau} \bar{\boldsymbol{z}}\_{\tau}^T
> = \mathbb{E} (\lambda \boldsymbol{\eta}\_{\tau} - \sigma\boldsymbol{\xi}\_{\tau}) (\lambda \boldsymbol{\eta}\_{\tau} - \sigma\boldsymbol{\xi}\_{\tau})^T
> = \lambda^2 \mathbb{E} \boldsymbol{\eta}\_{\tau}\boldsymbol{\eta}\_{\tau}^T - 2 \lambda \sigma(\tau) \mathbb{E} \boldsymbol{\eta}\_{\tau} \boldsymbol{\xi}\_{\tau}^T + \sigma^2 (\tau) \boldsymbol{I}.
> $$
>
> If we assume that the $\sigma(\tau) \downarrow 0$ as $\tau \uparrow \infty$ and the finite second moment of the gradient noise such that $\| \mathbb{E} \boldsymbol{\eta}_{\tau} \boldsymbol{\eta}_{\tau}^T \| \leq \bar{\sigma} < \infty$, we recognize the approximation of the SGD is a conventional Le'vy-SDE.
>
> Therefore, we conjecture that the additive noise may have little effect on the learning performance.
>
> Meanwhile, for the proposed quantization scheme, we obtain
> $$
> \begin{aligned}
> \boldsymbol{x}\_{\tau+1}^Q
> &= \boldsymbol{x}\_{\tau}^Q - [\lambda \nabla\_{\tau_b} \tilde{f}({x}\_{\tau} )]^Q \\\\
> &= \boldsymbol{x}\_{\tau}^Q - [\lambda \nabla f(\boldsymbol{x}\_t) + \lambda \boldsymbol{\eta}\_{\tau}]^Q\\\\
> &= \boldsymbol{x}\_{\tau}^Q - [\lambda \nabla f(\boldsymbol{x}\_t)]^Q + \overline{[\lambda \boldsymbol{\eta}\_{\tau}]}^Q, \quad &\because [\lambda \nabla f(\boldsymbol{x}\_t) + \lambda \boldsymbol{\eta}\_{\tau}]^Q \equiv [\lambda \nabla f(\boldsymbol{x}\_t)]^Q - \overline{[\lambda \boldsymbol{\eta}\_{\tau}]}^Q \\\\
> &= \boldsymbol{x}\_{\tau}^Q - \lambda \nabla f(\boldsymbol{x}\_t) + Q_p^{-1}(t) \left(\boldsymbol{\varepsilon}\_{\tau}^Q + \boldsymbol{\nu}\_{\tau} \right), \quad &\because \boldsymbol{\nu}\_{\tau} \triangleq \lfloor \bar{\boldsymbol{\eta}} Q\_p^{-1}(t)+ 0.5\rfloor.
> \end{aligned}
> $$
> The equation represents that, even if the quantization error and the normalized gradient error are not independent, we can control the variance using the quantization parameter $Q_p^{-1}(t)$.
>
> Furthermore, supposing that the quantization error and the normalized gradient noise are independent, we can establish the standard Langevin SDE for SGD through the classical or Lindberg CLT.
>
> Since the Langevin SDE can satisfy the Lagrange method for the global convergence on a real-number space, we conjecture the performance of the proposed algorithm would be superior.
>
> (3) We're afraid we cannot provide any information to the some reference for quantization-based non-convex optimization, due to the guidelines of the ICLR committee.
> Briefly, the quantization for the objective function enables to decrease of the level set or search range exponentially, under the numerical number system.
> The dynamics of optimization based on the quantization resembles the quantum annealing.
> However, it does not require a base function for Hamiltonian which the quantum annealing algorithm requires.

---

> > ### Comment · Reviewer_wZWw · 2023-11-23
> >
> > I appreciate the kind and detailed reply. Thank you. Although the idea behind this paper is interesting, further numerical tests and discussions regarding the implementation are still required to support it. I would like to keep my score unchanged.

---

### Official Review · Reviewer_qh6T · 2023-11-01

**Soundness:** 2 fair
**Presentation:** 1 poor
**Contribution:** 2 fair
**Rating:** 3
**Confidence:** 3

**Summary:**

The key argument of the paper is that a variant of stochastic gradient Langevin dynamics can be realized by combining a quantization scheme with standard SGD. In particular, this avoids the explicit injection of noise into the gradient descent scheme.

**Strengths:**

The scheme proposed by the authors which links quantization with SGLD is novel and could be developed in future works. In situations where there is a pre-existing need to quantize the result, this would implicitly lead to benefits.

The method does seem to show improvement empirically, with caveats (see below).

**Weaknesses:**

There are many issues with the writing. In general the clarity of explanations could be improved, and the number of typos has a substantial effect on the readability of the document. See Questions for an exhaustive list. These must be fixed in order for me to recommend acceptance.

The empirical evidence is not entirely convincing for me. The loss curves (Figure 2) do not seem to show clear improvement, and it is difficult to assess Table 1 without standard deviations. Furthermore, it seems a bit strange to me that the SGLD formulation would yield any improvement to SGD-type algorithms, since prior practical performance was not stellar.

The convergence results in Theorems 3.3, 3.4 are quite strange. See questions below.

Due to the above issues, I cannot recommend acceptance for this paper at the moment.

**Questions:**

Why is the metric in Theorem 3.3 chosen to be the overlap of the kernels $p$? Why is this significant in practice, compared to e.g. mean-square convergence or convergence in a probabilistic sense such as KL?

Assumption 4 should be specifically referring to the local optimal point? A definition of local optimum in this case should be given.

In my opinion the results should be stated independently of the “mini-batch” and “epoch” formula, which merely complicates the presentation of the core idea.

Could the authors provide some rough estimation of standard deviations in Table 1?

Why do the training curves performance in Figure 2 not appear to match that in Table 1?

See below for a list of detailed questions regarding the writing and proofs:

**Typos:**

Frame -> framework (page 1)

There should be a space between words if followed by parentheses

Lines should not start with commas, e.g. after Eq 1, Eq 3

Owing to the quantization error as the i.i.d. White noise -> unclear what this means

Increments -> increment (page 5)

What is the point of discussing the transformation approach in such depths if it is not explored further? This point can be made more succinctly.

Appliance -> application (page 6)

In equation 20, what is C_{o1}? There must additionally be some typo, since why do both upper and lower case T appear in the equation?
(22) should be less than or equal to.

Page 15: “We” -> “we”

Definitions of floor/ceiling should not have $\forall x \in \mathbb{R}$ in the set.

Equation 25 is missing a sup on $v$.

Equation 27 seems redundant in light of Equation 28. Likewise Eq. 30 and Eq. 32s, and Eq. 37, 39.

I cannot follow the derivation in the first part of Equation 42, as there appear to be some typos (e.g. the second to last equality cannot be true, and the summation in the first equation does not make sense). Such an argument is not necessary, anyway, and one can simply appeal to symmetry.

Page 22: Indexes -> indices

Lemma: Auxiliary 1 is standard and there is no need to include the proof.

---

> ### Author Response · Authors · 2023-11-20
> **Answer for the reviewer qh6T (1)**
>
> We appreciate the thoroughness and thoughtfulness of your comments.
>
> ## Answer for the questions
>
> We answer **the first and the second questions** in the "weakness" section.
>
> Theorem 3.3  describes that the distribution of a transition probability density in the proposed algorithm converges without any convex assumption, for time increasing.
> The convergence without convex assumption illustrates that the proposed algorithm can escape a local minimum, proportional to the transition probability density.
> From a mathematical standpoint, we conjecture that asymptotic convergence in a strong topology, termed mean-square convergence by the reviewer, does not guarantee global convergence unless a convex assumption is present.
> Although a lot of researchers claim the global convergence of the learning algorithm, most of them require alternative or hidden convex conditions in their proofs.
> We claim that the rigorous proof for the global convergence requires Laplace's method, which is the steepest descent method on the real number space.
> For this purpose, we should weak convergence of the algorithm as described in theorem 3.3.
> Additionally, based on theorem 3.3, we should prove the proposition that the distribution of the transition probability density converges to a stationary distribution.
> However, we couldn't provide the proof in this manuscript.
>
> Additionally, as the reviewer comments, the transition probability is not the correct expression, so we modify it as the transition probability density.
> Since we derive the SDE for the proposed algorithm in Lemma 3.2, the transition probability density is more correct than the transition probability kernel.
>
> This is **the answer to the third question** :
> The reviewer's comment is right.
> We should write equation (130) as the definition of theorem 3.4 in the manuscript.
> However, we removed the equation due to the length limitation of the manuscript.
> We insert equation (130) at the bottom of theorem 3.5 to clarify the theorem 3.5.
>
> This is **the answer to the fourth question** :
> In contrast to conventional research on stochastic learning equations, we employ a double-sided time index system that distinguishes between the unit of a mini-batch and the unit of an epoch.
> Comparing the SDE approximation based on a gradient noise in the conventional research and the proposed quantization error, we'd like to claim that the proposed approximation is based on a more reasonable assumption using the double-sided time system.
> (In the conventional research employing the gradient noise as follows:
> $$
> \boldsymbol{\eta}\_{\tau} \triangleq \frac{1}{B} \sum_{\tau_b=0}^{B-1} \nabla\_{\boldsymbol{x}}\tilde{f}\_{\tau_b}(\boldsymbol{x}\_{t\_e+\tau_b/B}) - \nabla\_{\boldsymbol{x}}\tilde{f}\_{\tau_b} (\boldsymbol{x}\_{t\_e+\tau/B}), \quad \tau\_b \in \mathbb{Z}[0, B)
> $$
> where all notations are defined in the manuscript, specifically in equation  (2).
> Can we ensure that the distribution of $\boldsymbol{\eta}_{\tau}$ is symmetry and the second moment represents a finite value? Recent research indicates that, under the empirical symmetry assumption, the second and higher moments are finite values.
>
> Meanwhile, the distribution of the quantization error is symmetric and involves finite variance under the axiomatic definition. Whereas, the distribution of the quantization error is symmetric and involves finite variance under the axiomatic definition.)
>
> In the proposed double-sided time system, we consider the discrete stochastic learning equation as an infinitesimal for the SDE and derive the SDE applying the (Lindberg-L'evy's) central limit theorem(CLT).
> Herein, we establish the SDE on the continuous time-space based on the epoch unit, not on the mini-batch unit.
> From the practical point of view, we don't stop learning in the middle of the epoch and obtain a learned AI model after the finish of learning at an epoch.
> Consequently, we guess that the proposed double-sided time system is reasonable.
> Furthermore, the proposed double-sided time system is not an original idea but is a conventional time system in research of stochastic approximation.
> The reviewer can see this double-sided time system in Kushner's work[1].
>
> We cannot understand that there exists a discrepancy in the performance curve in training represented in Figure 2.
> Would you misunderstand the algorithm's name in the legend, due to the small picture?

---

> ### Author Response · Authors · 2023-11-20
> **Answer for the reviewer qh6T (2)**
>
> ## Correction of Typos
> 1. I correct the typos the reviewer refers to (1)(2)(3)(5)(7)(9)(10)(11)(14) in the section of Typos, respectively.
>    Please, review the revised paper to verify the correction of typos in the previous version.
>
> 2. For (4), we rewrite the sentence as follows:
>    "~, owing to the quantization error as a uniformly distributed vector-valued random variable. "
>
> 3. For (6), accepting the reviewer's comment, we rewrite the statements for the transformation succinctly.
>
> 4. For (8), $C$ is typo of $C_{o1}$, so we correct it. As the reviewer comments, (20) is less than or equal, so we rewrite it.  The notation $\tau$ is the time index on the unit of mini-batch and the large $T$ denotes a final time index on the unit of epoch. We define those notations in section A.1 of the Appendix. Furthermore, in the same section, we define the relation between the epoch-based continuous time index $t$ and the mini-batch-based discrete time index $\tau$  such that $\tau = t \times B$.
>
> 5. For (12),  as the reviewer comments, those equations are redundant. Those are on purpose. From the brief formulation such as equations (27), (37), and (39), we can obtain the result that some expectations based on the different random variables such as $z$ and $\epsilon^q$ are equivalent. With these equivalents, we prove the theorem for the dithering effect such as equation (48).
>
> 6. For (20), I fully agree with the reviewer's comment. When we check the equation, we recognize the expansion of the equation with absurd logic. Based on the reviewer's comment, we evaluate the result of the equation straightforwardly through the symmetry assumption, which we previously mentioned.
>
> 7. For (15), following the reviewer's comment, we remove the proof of lemma: auxiliary.
>
> ## Reference
> [1] Harold J. Kushner. "On the Weak Convergence of Interpolated Markov Chains to a Diffusion." Ann. Probab. 2 (1) 40 - 50, February, 1974. https://doi.org/10.1214/aop/1176996750

---

### Meta-Review · Area_Chair_BdEE · 2023-12-08

**Metareview:**

The work presents an approach to SGLD based on suitable quantization without the need to explicitly add noise. Theoretical results on the approach as well as empirical evidence on its performance is provided. The reviewers have raised several concerns regarding the work including some of the assumptions for the analysis, the overall motivation, lack of clarity in aspects of the exposition, limited empirical evaluation, among others. The authors have responded to these concerns, but some of the concerns linger on.

**Justification For Why Not Higher Score:**

There were several concerns on the original submission, which does not seem to have been resolved to the satisfaction of the reviewers.

**Justification For Why Not Lower Score:**

N/A

---

### Decision · Program_Chairs · 2024-01-16

Reject